# Orai3 and Orai1 mediate CRAC channel function and metabolic reprogramming in B cells

**Scott M Emrich[1], Ryan E Yoast[1], Xuexin Zhang[1], Adam J Fike[2], Yin-Hu Wang[3], Kristen N Bricker[2], Anthony Y Tao[3], Ping Xin[4,5], Vonn Walter[6], Martin T Johnson[1], Trayambak Pathak[4,5], Adam C Straub[4,5], Stefan Feske[3], Ziaur SM Rahman[2], Mohamed Trebak[1,4,5]\***

[1]Department of Cellular and Molecular Physiology, Pennsylvania State University College of Medicine, Hershey, United States; [2]Department of Microbiology and Immunology, Pennsylvania State University College of Medicine, Hershey, United States; [3]Department of Pathology, New York University School of Medicine, New York, United States; [4]Department of Pharmacology and Chemical Biology, University of Pittsburgh School of Medicine, Pittsburgh, United States; [5]Vascular Medicine Institute, University of Pittsburgh School of Medicine, Pittsburgh, United States; [6]Department of Public Health Sciences, Pennsylvania State University College of Medicine, Hershey, United States

**\*For correspondence:** TREBAKM@PITT.EDU

**Abstract** The essential role of store-operated $Ca^{2+}$ entry (SOCE) through $Ca^{2+}$ release-activated $Ca^{2+}$ (CRAC) channels in T cells is well established. In contrast, the contribution of individual Orai isoforms to SOCE and their downstream signaling functions in B cells are poorly understood. Here, we demonstrate changes in the expression of Orai isoforms in response to B cell activation. We show that both Orai3 and Orai1 mediate native CRAC channels in B cells. The combined loss of Orai1 and Orai3, but not Orai3 alone, impairs SOCE, proliferation and survival, nuclear factor of activated T cells (NFAT) activation, mitochondrial respiration, glycolysis, and the metabolic reprogramming of primary B cells in response to antigenic stimulation. Nevertheless, the combined deletion of Orai1 and Orai3 in B cells did not compromise humoral immunity to influenza A virus infection in mice, suggesting that other in vivo co-stimulatory signals can overcome the requirement of BCR-mediated CRAC channel function in B cells. Our results shed important new light on the physiological roles of Orai1 and Orai3 proteins in SOCE and the effector functions of B lymphocytes.

## Editor's evaluation

This study shows that Orai1 and Orai3 are important elements of the calcium signaling machinery in mouse B cells and regulate downstream responses of proliferation, energetics, and metabolic reprogramming that occur after stimulation of the B cell antigen receptor. This work provides new information of interest to a broad range of readers in the fields of immunology, ion channels, and calcium signaling.

## Introduction

Calcium ($Ca^{2+}$) is an essential regulator of immune cell function (***Prakriya and Lewis, 2015***; ***Trebak and Kinet, 2019***). Crosslinking of immunoreceptors like the T cell receptor (TCR) or B cell receptor (BCR) triggers a robust elevation in intracellular $Ca^{2+}$ concentrations through the release of endoplasmic

reticulum (ER) $Ca^{2+}$ stores and a concomitant influx of $Ca^{2+}$ from the extracellular space (*Baba and Kurosaki, 2016*; *King and Freedman, 2009*). In lymphocytes, $Ca^{2+}$ entry across the plasma membrane (PM) is predominately achieved through CRAC channels, which constitute the ubiquitous SOCE (*Prakriya and Lewis, 2015*; *Trebak and Kinet, 2019*; *Trebak and Putney, 2017*; *Vaeth et al., 2020*). Stimulation of immunoreceptors coupled to phospholipase C isoforms results in the production of the secondary messenger inositol-1,4,5-trisphosphate (IP$_3$), which triggers the release of ER $Ca^{2+}$ through the activation of ER-resident IP$_3$ receptors. A reduction in ER $Ca^{2+}$ concentrations is sensed by stromal interaction molecule 1 (STIM1) and its homolog STIM2, resulting in a conformational change and clustering in ER-PM junctions where STIM molecules interact with and activate PM hexameric Orai channels (Orai1-3) to mediate SOCE (*Lunz et al., 2019*). $Ca^{2+}$ entry through CRAC channels regulates immune cell function through a host of $Ca^{2+}$-sensitive transcription factors including the NFAT, nuclear factor κB (NF-κB), c-Myc, and mTORC1 (*Berry et al., 2018*; *Trebak and Kinet, 2019*; *Vaeth and Feske, 2018*; *Vaeth et al., 2020*; *Vaeth et al., 2017a*). The coordination of these master transcriptional regulators is indispensable for innate and adaptive immune cell effector function including entry into the cell cycle, clonal expansion, cytokine secretion, differentiation, and antibody production (*Shaw and Feske, 2012*; *Trebak and Kinet, 2019*; *Vaeth et al., 2020*).

Orai1 has long been established as a central component of the *native* CRAC channel in all cell types studied. Although all three Orai isoforms are ubiquitously expressed across tissue types and can form functional CRAC channels when ectopically expressed, only recently have Orai2 and Orai3 emerged as regulators of *native* CRAC channel function (*Emrich et al., 2022b*; *Tsvilovskyy et al., 2018*; *Vaeth et al., 2017b*; *Yoast et al., 2020a*). Patients with inherited loss-of-function (LoF) mutations in *Orai1* (e.g. R91W mutation) develop a CRAC channelopathy with symptoms including combined immunodeficiency, ectodermal dysplasia, muscular hypotonia, and autoimmunity (*Feske et al., 2006*; *Lian et al., 2018*; *McCarl et al., 2009*). While these patients display relatively normal frequencies of most immune cell populations, they are highly susceptible to reoccurring viral, bacterial, and fungal infections due to defects in T cell expansion, cytokine secretion, and metabolism (*Vaeth et al., 2020*). There are currently no reported patient mutations in *Orai2* or *Orai3* genes and the role these channels play within the immune system had largely been unclear. Interestingly, recent studies utilizing global Orai2 knockout mice have demonstrated that loss of Orai2 leads to enhanced SOCE and corresponding CRAC currents in bone marrow-derived macrophages, dendritic cells, T cells, enamel cells, and mast cells (*Eckstein et al., 2019*; *Tsvilovskyy et al., 2018*; *Vaeth et al., 2017b*), suggesting that Orai2 is a negative regulator of CRAC channel activity. Combined knockout in mice of both Orai1 and Orai2 in T cells led to a near ablation of SOCE and impaired humoral immunity, while ectopic expression of pore-dead mutants of Orai1 (E106Q) or Orai2 (E80Q) into individual *Orai1$^{-/-}$* or *Orai2$^{-/-}$* T cells blocked native SOCE (*Vaeth et al., 2017b*). Similarly, the generation of HEK293 cell lines lacking each Orai isoform individually and in combination resulted in altered $Ca^{2+}$ oscillation profiles, CRAC currents, and NFAT isoform activation (*Emrich et al., 2022b*; *Emrich et al., 2021*; *Yoast et al., 2020a*; *Yoast et al., 2020b*). These data provide evidence that Orai2 and Orai3 exert negative regulatory effects on native CRAC channels possibly by forming heteromeric CRAC channels with Orai1.

In contrast to T cells, much less is known regarding the role of SOCE in B lymphocytes. Early landmark work investigating B cell-specific STIM knockout mice established that STIM1 mediates the vast majority of SOCE in B cells, while only the combined deletion of both STIM1 and STIM2 resulted in substantial impairments in B cell survival, proliferation, and NFAT-dependent IL-10 secretion (*Matsumoto et al., 2011*). Unexpectedly, mice with STIM1/STIM2-deficient B cells show normal humoral immune responses to immunization with both T cell-dependent and independent antigens. These and other studies suggest that the severe immunodeficiency observed in CRAC channelopathy patients is due to impaired T cell responses (*Matsumoto et al., 2011*; *Vaeth et al., 2016*). Investigation of *Orai1$^{-/-}$* mice on the mixed Institute for Cancer Research (ICR) background and *Orai1$^{R93W}$* knock-in mice (the equivalent of human R91W mutation) demonstrated that SOCE is significantly attenuated in B cells (*Gwack et al., 2008*; *McCarl et al., 2010*). However, SOCE in B cells is only partially reduced with the loss of Orai1, suggesting that Orai2 and/or Orai3 mediate the remaining SOCE in B cell populations. A recent study showed that the deletion of Orai3 in mice does not affect B cell function and humoral immune responses (*Wang et al., 2022*), suggesting other Orai homologs compensate for the loss of Orai3. Thus, the contribution of each Orai isoform to native CRAC channel function and downstream signaling in B cells has remained obscure.

In this study, we investigated the contributions of Orai isoforms to SOCE and its downstream signaling in B cells through multiple CRISPR/Cas9 knockout B cell lines and novel B cell-specific *Orai* knockout mice. Our findings demonstrate that the expression of each Orai isoform is dynamically regulated in response to B cell activation and that the magnitude of SOCE is unique among effector B cell populations. We show that the deletion of Orai1 alone does not alter BCR-evoked cytosolic $Ca^{2+}$ oscillations, proliferation, and development. Unexpectedly, we show that Orai3 is involved in regulating B cell SOCE and downstream signaling functions. Deletion of both Orai1 and Orai3 strongly inhibits SOCE and hampers mitochondrial metabolism in B cells. Transcriptome and metabolomic analysis uncovered key signaling pathways that are regulated by SOCE and the phosphatase calcineurin for the efficient transition of B cells from a quiescent to a metabolically active state. Surprisingly, humoral immunity to influenza A virus infection of mice with B cell-specific deletion of both Orai1 and Orai3 was unaltered, suggesting that alternative $Ca^{2+}$-independent signaling pathways activated in vivo through co-stimulatory receptors can overcome the loss of CRAC channel activity in B cells. Our data elucidate the role of SOCE in B cell functions and show that CRAC channel function in B lymphocytes is mediated by both Orai1 and Orai3. This knowledge is important for the potential targeting of CRAC channels in specific lymphocyte subsets in immune and inflammatory diseases.

## Results

### Orai channel isoforms are dynamically regulated in response to B cell activation

Throughout their lifespan, B lymphocytes must undergo dramatic periods of metabolic adaptation whereby naïve, metabolically quiescent cells prepare to clonally expand and differentiate into effector populations (*Akkaya and Pierce, 2019*; *Boothby and Rickert, 2017*). Induction of these diverse metabolic programs is driven by downstream BCR-mediated $Ca^{2+}$ signals in combination with $Ca^{2+}$-independent costimulatory signals including CD40 activation and/or stimulation with various toll-like receptor (TLR) ligands (*Akkaya et al., 2018*; *Baba and Kurosaki, 2016*; *Berry et al., 2020*). Thus, using Seahorse assays we first evaluated how activation of wild-type mouse primary splenic B cells (isolated as described in methods) regulates oxygen consumption rates (OCR) and extracellular acidification rates (ECAR) as indicators of oxidative phosphorylation (OXPHOS) and glycolysis, respectively (*Figure 1A and B*). Naïve primary B cells from mouse spleens were isolated by negative selection and stimulated for 24 hr under five different conditions as follows: (1) control unstimulated cells (Unstim), (2) stimulation with anti-IgM antibodies (IgM) to activate the BCR, (3) stimulation with anti-CD40 (CD40), (4) stimulation with lipopolysaccharides (LPS), and (5) co-stimulation with anti-IgM and anti-CD40. B cell stimulation for 24 hr led to a robust increase in basal OCR, except for B cells stimulated with anti-CD40 alone where the increase in basal OCR was not statistically significant (*Figure 1A and C*). *Figure 1B* shows that B cells stimulated with either anti-IgM, anti-IgM +anti-CD40, or LPS become highly energetic and upregulate both OXPHOS and glycolytic pathways. However, B cells stimulated with anti-CD40 alone remain metabolically quiescent like control non-stimulated B cells (*Figure 1B*). We used the protonophore trifluoromethoxy carbonylcyanide phenylhydrazone (FCCP) to dissipate the mitochondrial membrane potential and calculate the maximal respiratory capacity of B cells. Consistent with results obtained for basal OCR, stimulated B cells showed a robust increase in maximal respiratory capacity except for cells stimulated with anti-CD40 alone (*Figure 1D*). Both total mitochondrial content and membrane potential measured with MitoTracker Green and TMRE staining, respectively, were substantially increased following anti-IgM, anti-IgM +anti-CD40, or LPS stimulation while both parameters were only slightly increased with stimulation with anti-CD40 alone (*Figure 1E and F*). While previous studies suggested that Orai1 regulates the majority of SOCE in primary B cells (*Gwack et al., 2008*; *McCarl et al., 2010*), the expression of different Orai isoforms both at rest and upon B cell activation is unknown. Utilizing the same five experimental conditions described above, we analyzed the mRNA expression of *Orai1*, *Orai2*, and *Orai3* following 24 hr of stimulation. Stimulation with either anti-IgM or anti-IgM +anti-CD40 dramatically increased *Orai1* expression, while the magnitude of this change following anti-CD40 or LPS stimulation was smaller and did not reach statistical significance (*Figure 1G*). Interestingly, stimulation conditions that strongly increased mitochondrial respiration (anti-IgM, anti-IgM +anti-CD40, LPS) all significantly reduced expression of *Orai2* while stimulation with anti-CD40 alone had no significant effect on *Orai2* expression (*Figure 1H*). The

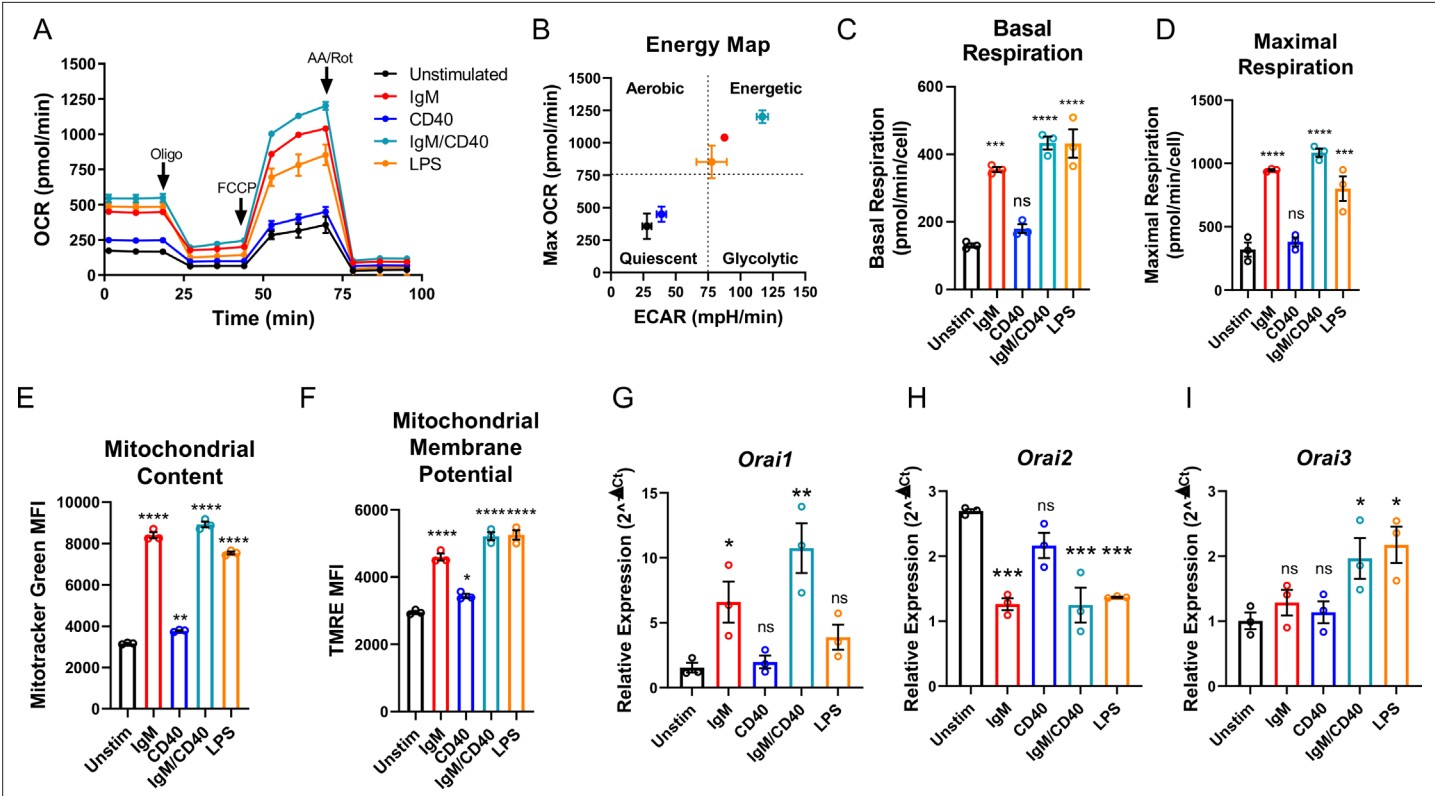

**Figure 1.** B cell activation dynamically regulates Orai channel expression. (**A**) Measurement of oxygen consumption rate (OCR) in primary B lymphocytes following 24 hr stimulation with anti-IgM (20 µg/mL), anti-CD40 (10 µg/mL), anti-IgM +anti-CD40, or LPS (10 µg/mL) using the Seahorse Mito Stress Test (n=3 biological replicates). (**B**) Energy map of maximal OCR and extracellular acidification rate (ECAR) following the addition of the protonophore carbonyl cyanide p-trifluoromethoxyphenylhydrazone (FCCP). (**C, D**) Quantification of basal (**C**) and maximal (**D**) respiration from Seahorse traces in (**A**) (One-way ANOVA with multiple comparisons to Unstimulated). (**E, F**) Measurement of (**E**) total mitochondrial content with the fluorescent dye MitoTracker Green and (**F**) mitochondrial membrane potential with TMRE following 24 hr stimulation (n=3 biological replicates; One-way ANOVA with multiple comparisons to Unstimulated). (**G–I**) Quantitative RT-PCR of (**G**) *Orai1*, (**H**) *Orai2*, and (**I**) *Orai3* mRNA following 24 hr of stimulation with the stimuli indicated (n=3 biological replicates; one-way ANOVA with multiple comparisons to Unstimulated). All scatter plots and Seahorse traces are presented as mean ± SEM. For all figures, *p<0.05; **p<0.01; ***p<0.001; ****p<0.0001; ns, not significant.

The online version of this article includes the following source data for figure 1:

**Source data 1.** Source data for *Figure 1*.

expression of Orai3 has significantly increased only with anti-IgM +anti-CD40 or LPS stimulation, but not with anti-IgM or anti-CD40 alone (*Figure 1I*).

## Both Orai3 and Orai1 mediate SOCE in A20 B lymphoblasts

Robust activation of primary B cells with anti-IgM/anti-CD40 co-stimulation resulted in upregulation of both *Orai1* and *Orai3* mRNA with downregulation of *Orai2* (*Figure 1G–I*). We, therefore, reasoned that B cells may sustain long-term cytosolic Ca$^{2+}$ signals through Orai1 and/or Orai3, while down-regulation of Orai2 might relieve the inhibition of native CRAC channels, as recently demonstrated for activated T cells (*Vaeth et al., 2017b*). To gain insights into the molecular composition of CRAC channels in B cells, we first developed an in vitro system utilizing the mouse A20 B lymphoblast cell line to generate single and double Orai1 and Orai3 knockout cell lines with CRISPR/Cas9 technology. Two guide RNA (gRNA) sequences were utilized to cut at the beginning and end of the coding region of mouse *Orai1* and *Orai3* (*mOrai1* and *mOrai3*), effectively excising the entirety of the genomic DNA for each *Orai* gene (*Figure 2A*). We generated multiple A20 clones that were lacking Orai1 and Orai3 individually and in combination and these knockout clones were validated through genomic DNA sequencing (*Source data 1*) and the absence of *Orai1* or *Orai3* transcripts by qPCR (*Figure 2B–D*). We measured SOCE in A20 cells after passive store depletion with thapsigargin, an inhibitor of

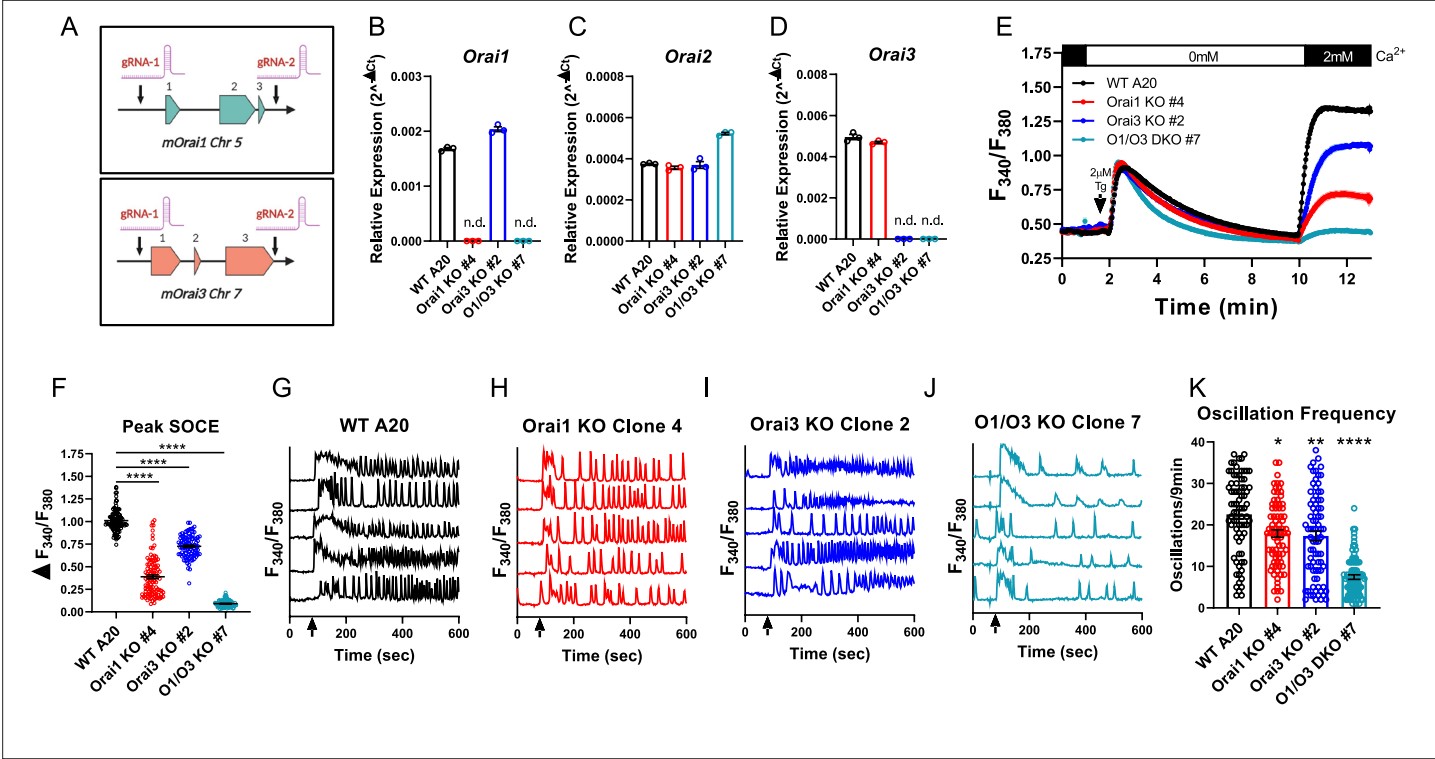

**Figure 2.** Orai1 and Orai3 mediate the bulk of store-operated Ca2+ entry (SOCE) in A20 B lymphoblasts. (**A**) Cartoon schematic of the two gRNA CRISPR strategies we used to excise mouse *Orai1* and *Orai3* genes. (**B–D**) Quantitative RT-PCR of (**B**) *Orai1*, (**C**) *Orai2*, and (**D**) *Orai3* mRNA in A20 Orai CRISPR clones (n=3 biological replicates). (**E**) Measurement of SOCE with Fura2 upon store depletion with 2 μM thapsigargin in 0 mM Ca2+ followed by re-addition of 2 mM Ca2+ to the external bath solution. (**F**) Quantification of peak SOCE in (**E**) (from left to right n=99, 100, 89, and 98 cells; Kruskal-Wallis test with multiple comparisons to WT A20). (**G–J**) Representative Ca2+ oscillation traces from 5 cells/condition measured with Fura2 upon stimulation with 10 μg/mL anti-IgG antibodies at 60 s (indicated by arrows) in the presence of 2 mM external Ca2+. (**K**) Quantification of total oscillations in 9 min from (**G–J**) (from left to right n=76, 79, 79, and 78 cells; Kruskal-Wallis test with multiple comparisons to WT A20). All scatter plots are presented as mean ± SEM. For all figures, *p<0.05; **p<0.01; ****p<0.0001; ns, not significant.

The online version of this article includes the following source data for figure 2:

**Source data 1.** Source data for *Figure 2*.

the sarcoplasmic/endoplasmic reticulum ATPase (SERCA). A20 cells lacking Orai1 demonstrated a large reduction in maximal SOCE by ~62%, while in cells lacking Orai3 SOCE was reduced by ~28% (*Figure 2E and F*). Importantly, the combined deletion of both Orai1 and Orai3 caused a near abrogation of SOCE by ~91% (*Figure 2E and F*). Furthermore, measurements of cytosolic Ca2+ oscillations induced by anti-IgG stimulation demonstrated that the oscillation frequency was substantially reduced with combined Orai1/Orai3 knockout, while only slightly reduced with the loss of either Orai1 or Orai3 individually (*Figure 2G–K*). These data suggest that both Orai3 and Orai1 were involved in optimal CRAC channel function in A20 B lymphoblasts. These data are consistent with the function of Orai channels in other cell types. Indeed, we recently utilized a series of single, double, and triple Orai CRISPR/Cas9 knockout HEK293 cell lines and showed that SOCE, but not Orai1, is required for agonist-induced Ca2+ oscillations (*Yoast et al., 2020b*). Under conditions of physiological agonist stimulation that causes modest ER store depletion while eliciting Ca2+ oscillations, Orai2 and Orai3 were sufficient to maintain cytosolic Ca2+ oscillations, while having relatively minor contributions to SOCE induced by maximal store depletion (*Yoast et al., 2020b*).

## Orai1 is dispensable for cytosolic Ca2+ oscillations in primary B cells

To determine the contribution of Orai1-mediated Ca2+ signals to primary B cell function, we generated B cell-specific Orai1 knockout (*Orai1*fl/fl Mb1-Cre/+) mice (*Figure 3*). Compared to *Orai1*fl/fl controls, the average surface expression of Orai1 was significantly reduced on B220+ B cells from *Orai1*fl/fl Mb1-Cre/+ mice by~70% (*Figure 3A and B*). The residual signal of ~30% above FMO (*Figure 3A*) is likely

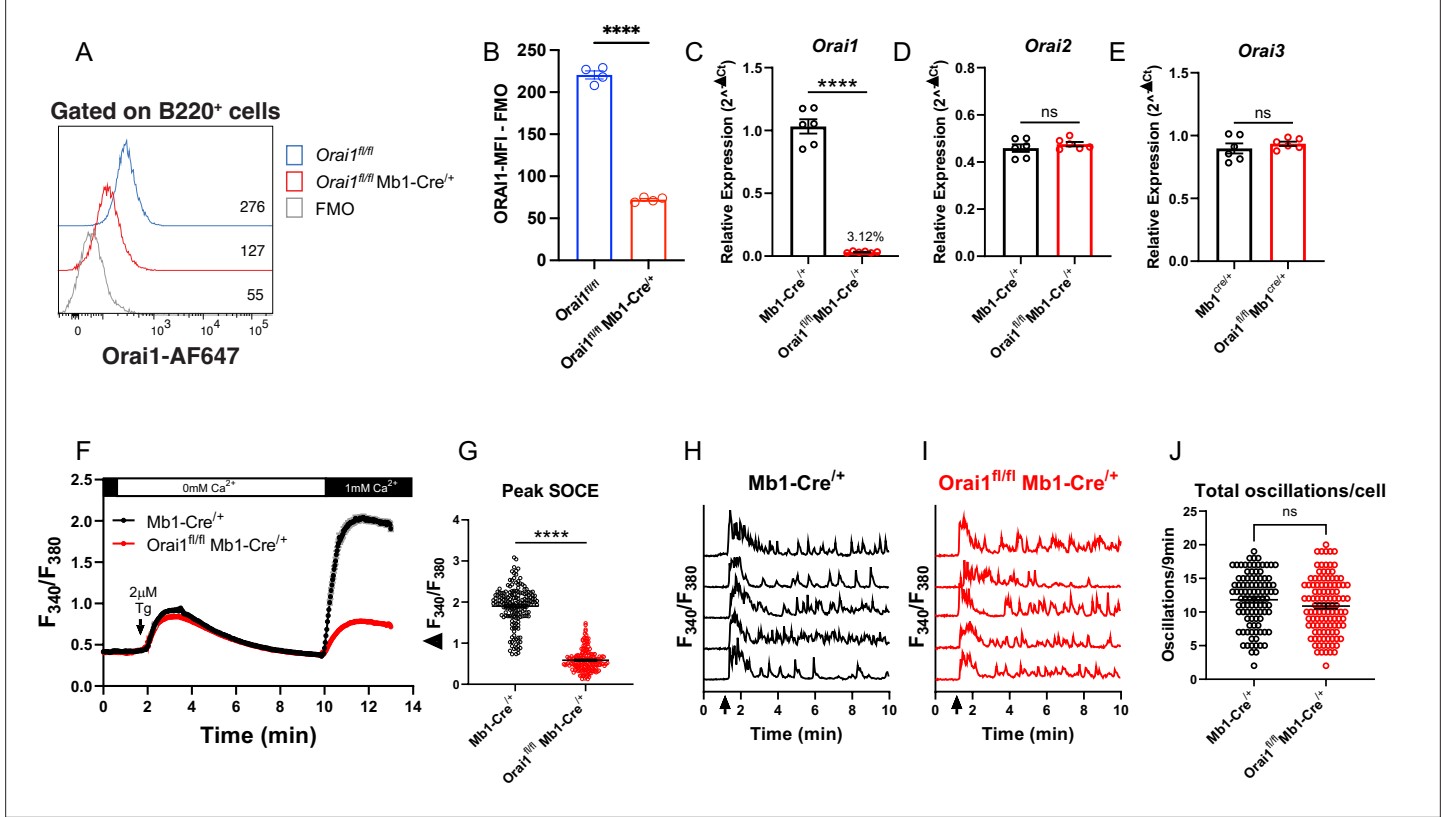

**Figure 3.** Orai1 is dispensable for BCR-induced Ca²⁺ oscillations in primary B cells. (**A**) Representative flow cytometry histogram of B cells isolated from *Orai1^fl/fl* and *Orai1^fl/fl* Mb1-Cre^/+ mice. Splenocytes from naïve *Orai1^fl/fl* and *Orai1^fl/fl* Mb1-Cre^/+ mice were fixed, permeabilized, and stained with rabbit anti-Orai1 polyclonal antibody (YZ6856, epitope: human ORAI1#275–291 intra-cellular C-terminal, cross-reacts with the mouse). The numbers inside the panel represent the mean fluorescence intensity (MFI) for the Orai1 antibody staining for each sample. (**B**) Quantification of Orai1 MFI minus fluorescence minus one (FMO) in B cells is shown. (n=4 biological replicates; unpaired T-test). (**C–E**) Quantitative RT-PCR of (**C**) *Orai1*, (**D**) *Orai2*, and (**E**) *Orai3* mRNA in isolated B cells (n=6 biological replicates for each; Mann-Whitney test). (**F**) Measurement of SOCE with Fura2 upon store depletion with 2 µM thapsigargin in 0 mM Ca²⁺ followed by re-addition of 1 mM Ca²⁺ to the external bath solution. (**G**) Quantification of peak store-operated Ca2+ entry (SOCE) in (**F**) n=169 and 178 cells; Mann-Whitney test. (**H–I**) Representative Ca²⁺ oscillation traces from 5 cells/condition measured with Fura2 upon stimulation with 20 µg/mL anti-IgM antibodies at 1 min (indicated by arrows) in the presence of 1 mM external Ca²⁺. (**J**) Quantification of total oscillations in 9 min from (**I, J**) (n=97 and 111 cells; Mann-Whitney test). All scatter plots are presented as mean ± SEM. For all figures, \*\*p<0.01; \*\*\*\*p<0.0001; ns, not significant.

The online version of this article includes the following source data and figure supplement(s) for figure 3:

**Source data 1.** Source Data for *Figure 3*.

**Figure supplement 1.** Loss of Orai1 does not overtly affect B cell activation.

**Figure supplement 1—source data 1.** Source data for *Figure 3—figure supplement 1*.

the result of the non-specific binding of the Orai1 antibody by its variable region to other Orai homologs (Orai2, Orai3), which partially share the peptide epitope recognized by the anti-Orai1 antibody. This residual Orai1 antibody signal is consistent with previous studies showing that Cre-mediated deletion of Orai1^fl/fl alleles is very efficient in other cell types including T cells (*Kaufmann et al., 2016*), and neuronal progenitor cells (*Somasundaram et al., 2014*). Indeed, other lines of evidence indicate that Orai1 is deleted in B cells of *Orai1^fl/fl* Mb1-Cre^/+ mice: our qPCR data show that B cells from *Orai1^fl/fl* Mb1-Cre^/+ mice showed a near abrogation of *Orai1* mRNA (to ~3% of control) compared to Mb1-Cre^/+ control mice (*Figure 3C*) without any significant change in *Orai2* (*Figure 3D*) or *Orai3* (*Figure 3E*) mRNA expression. Further, Ca²⁺ measurements demonstrated that SOCE is reduced by ~69% in B cells isolated from *Orai1^fl/fl* Mb1-Cre^/+ mice (*Figure 3F and G*). We measured cytosolic Ca²⁺ oscillations in response to anti-IgM stimulation in the presence of 1 mM extracellular Ca²⁺. In agreement with previous reports and consistent with our data with A20 B lymphoblasts (*Figure 2G and H*), Orai1-deficient B cells display a comparable frequency of agonist-induced Ca²⁺ oscillations

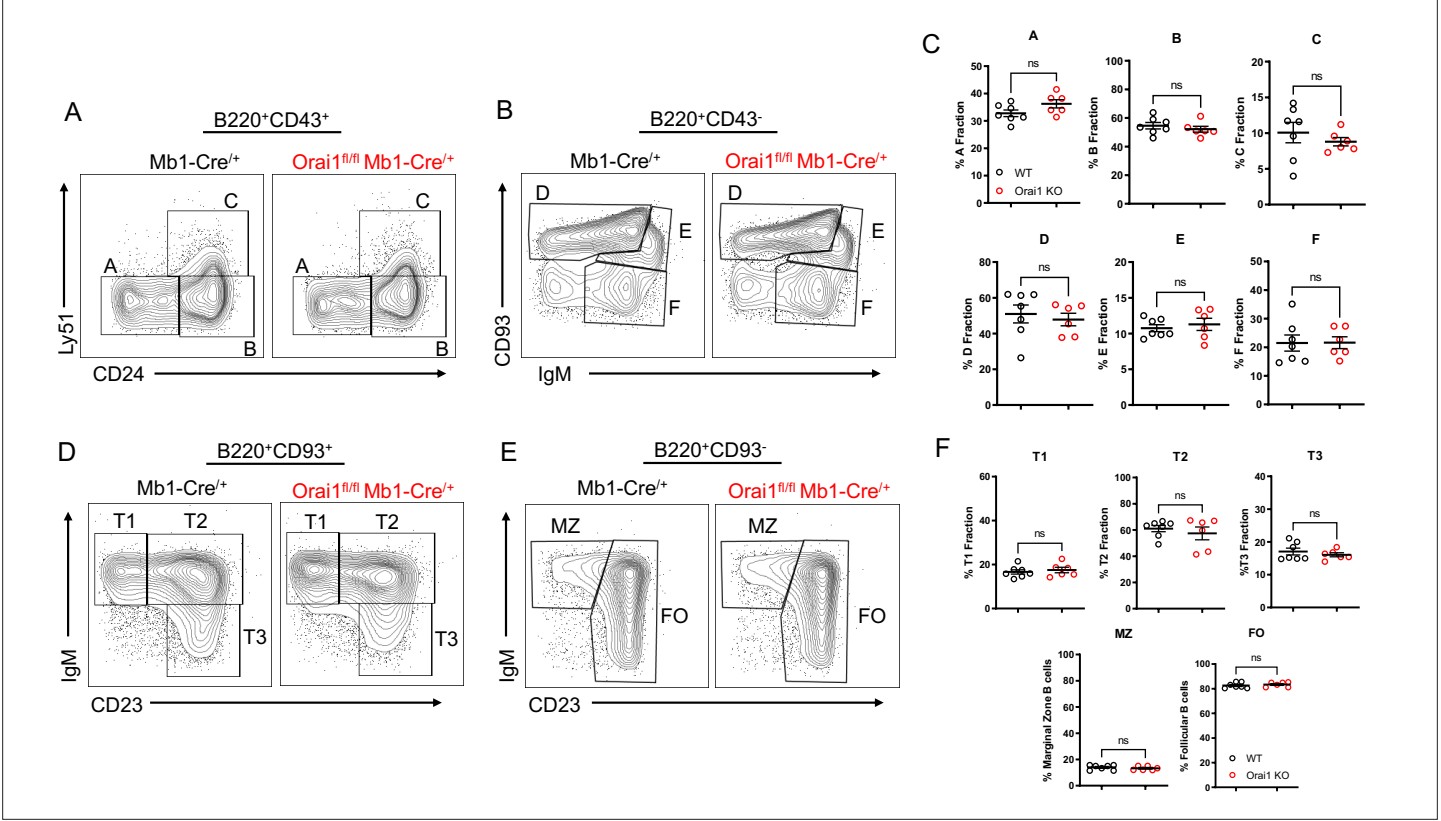

**Figure 4.** Orai1 is dispensable for B cell development. (**A–B**) Flow cytometric analysis of bone marrow populations for B cell fractions A (B220⁺CD43⁺HSA⁻BP-1⁻), B (B220⁺CD43⁺HSA⁺BP-1⁻), C (B220⁺CD43⁺HSA⁺BP-1⁺), D (B220⁺CD43⁻IgM⁻CD93⁺), E (B220⁺CD43⁻IgM⁺CD93⁺), and F (B220⁺CD43⁻IgM⁺CD93⁻). (**C**) Quantification of bone marrow populations in (**A, B**) (n=7 and six biological replicates; Mann-Whitney test). (**D**) Flow cytometric analysis of isolated populations in the spleen for B cell developmental stages T1 (B220⁺AA4.1⁺CD23⁻IgM⁺), T2 (B220⁺AA4.1⁺CD23⁺IgM⁺), and T3 (B220⁺AA4.1⁺CD23⁺IgM⁻). (**E**) Flow cytometric analysis of isolated populations in the spleen for marginal zone (MZ) B cells (B220⁺CD93⁻CD23⁻IgM⁺) and follicular (FO) B cells (B220⁺CD93⁻CD23⁺IgM⁺). (**F**) Quantification of splenic populations in (**D, E**) (n=7 and six biological replicates; Mann-Whitney test). All scatter plots are presented as mean ± SEM.

The online version of this article includes the following source data for figure 4:

**Source data 1.** Source data for *Figure 4*.

as control B cells (*Figure 3H–J*). Thus, while Orai1 knockout significantly decreased SOCE in primary B cells, it is dispensable for maintaining Ca²⁺ oscillations in response to physiological agonist stimulation. In control Mb1-Cre/⁺ mice, Orai1 expression was comparable between B220⁺ B cells and CD8⁺ T cells, with CD4⁺ T cells showing a slightly lower signal (*Figure 3—figure supplement 1A, B*). After 24 hour stimulation of control B cells under the five conditions described above, Orai1-deficient B cells showed no apparent defects in activation as the expression of MHC-II was comparable to control B cells (*Figure 3—figure supplement 1C–E*), while expression of CD86 was slightly reduced with anti-IgM stimulation or anti-IgM +anti-CD40 co-stimulation (*Figure 3—figure supplement 1F–H*).

## Loss of Orai1 does not alter B cell development

While patients and mouse models with loss of function (LoF) mutations in *Orai1* present with severe immunodeficiency due to impaired T cell function, the development of most immune cell populations is largely unaltered, suggesting SOCE is dispensable for initial immune cell selection and development (*Gwack et al., 2008*; *Lacruz and Feske, 2015*; *McCarl et al., 2010*). We evaluated B cell development within the bone marrow and spleen of *Orai1ᶠˡ/ᶠˡ* Mb1-Cre/⁺ mice (*Figure 4*). In agreement with previous reports that investigated B cell development in global *Orai1* deficient mice, the development of early B cell progenitors in fractions A-F within the bone marrow were unaltered in *Orai1ᶠˡ/ᶠˡ* Mb1-Cre/⁺ mice (*Figure 4A–C*). Similarly, we observed no significant differences in peripheral transitional type 1 (T1), T2, and T3 immature B cells in the spleen (*Figure 4D and F*). Mature B cell populations

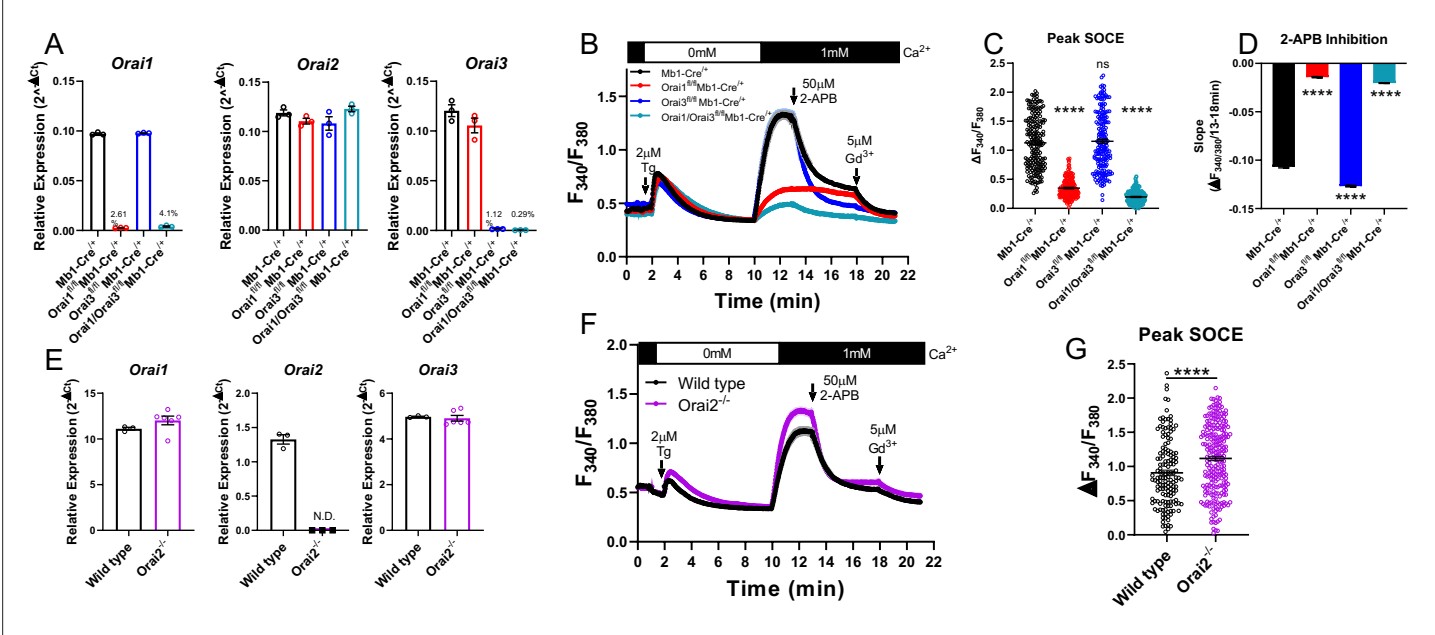

**Figure 5.** Orai1 and Orai3 synergistically mediate store-operated Ca2+ entry (SOCE) in primary B cells. (**A**) Quantitative RT-PCR of *Orai1*, *Orai2*, and *Orai3* mRNA in negatively isolated B cells from B cell-specific Orai knockout mice (n=3 biological replicates per genotype). (**B**) Measurement of SOCE in naïve B cells with Fura2 upon store depletion with 2 µM thapsigargin in 0 mM Ca$^{2+}$ followed by re-addition of 1 mM Ca$^{2+}$ to the external bath solution. Subsequently, SOCE was inhibited with the addition of 50 µM 2-APB at 13 min followed by 5 µM Gd$^{3+}$ at 18 min. (**C**) Quantification of peak SOCE in (**B**) (from left to right n=200, 200, 199, and 149 cells; Kruskal-Wallis test with multiple comparisons to Mb1-Cre$^{/+}$). (**D**) Quantification of the rate of 2-APB inhibition from 13 to 18 min. (**E**) Quantitative RT-PCR of *Orai1*, *Orai2*, and *Orai3* mRNA in negatively isolated B cells from wild-type and Orai2$^{-/-}$ mice (n=3 and six biological replicates). (**F**) Measurement of SOCE in naïve B cells with Fura2 from wild-type and *Orai2$^{-/-}$* mice. (**G**) Quantification of peak SOCE in (**F**) (n=147 and 240 cells; Mann-Whitney test). All scatter plots are presented as mean ± SEM. For all figures, ***p<0.001; ****p<0.0001; ns, not significant.

The online version of this article includes the following source data and figure supplement(s) for figure 5:

**Source data 1.** Source data for *Figure 5*.

**Figure supplement 1.** Store-operated Ca2+ entry (SOCE) in B cells activated for 48 hr with Anti-IgM +Anti CD40.

**Figure supplement 1—source data 1.** Source data for *Figure 5—figure supplement 1*.

in the spleen are predominately follicular B cells with a smaller fraction of marginal zone B cells, and these ratios were largely comparable between control and *Orai1$^{fl/fl}$* Mb1-Cre$^{/+}$ mice (*Figure 4E and F*).

## Both Orai3 and Orai1 synergistically contribute to SOCE in primary B cells

To determine whether Orai3 regulates Ca$^{2+}$ signals and function of primary B cells, we generated B cell-specific Orai3 knockout mice (*Orai3$^{fl/fl}$* Mb1-Cre$^{/+}$) and B cell-specific double Orai1/Orai3 knockout mice (*Orai1/Orai3$^{fl/fl}$* Mb1-Cre$^{/+}$). Primary B cells isolated from spleens of Orai1, Orai3, and Orai1/Orai3 knockout mice showed near complete ablation of their respective *Orai* isoform mRNA (to ~0.3–4% of control) with no compensatory changes in *Orai2* mRNA expression (*Figure 5A*). As documented above (*Figure 3F and G*), depletion of ER Ca$^{2+}$ stores with thapsigargin demonstrated that SOCE was significantly reduced in Orai1 knockout B cells (by ~69%), and this remaining Ca$^{2+}$ entry was further reduced in Orai1/Orai3 double knockout B cells (by ~83%; *Figure 5B and C*). However, SOCE in single Orai3 knockout B cells was ~102% of control, which was not significantly different from control B cells.

Orai channel isoforms demonstrate distinct pharmacological profiles and sensitivities to various CRAC channel modifiers (*Bird and Putney, 2018*; *Zhang et al., 2020*). One of the most extensively utilized CRAC channel modifiers is 2-aminoethyl diphenyl borate (2-APB) (*Prakriya and Lewis, 2001*), which at high (25–50 µM) concentrations strongly inhibits Orai1, partially inhibits Orai2, and potentiates Orai3 channel activity (*DeHaven et al., 2008*; *Zhang et al., 2008*; *Zhang et al., 2020*). After allowing Ca$^{2+}$ entry to the plateau, 50 µM 2-APB was added, followed by the addition of 5 µM gadolinium

(Gd$^{3+}$) which potently blocks all Orai isoforms (*Yoast et al., 2020b*; *Zhang et al., 2020*). In wild-type B cells, 2-APB led to a gradual inhibition of SOCE over the course of 5 min, which completely returned to baseline following the addition of Gd$^{3+}$. Interestingly, the remaining SOCE in Orai1 knockout B cells showed essentially no inhibition by 2-APB but was strongly inhibited by Gd$^{3+}$. The lack of effect of 2-APB is likely due to the residual SOCE mediated by the remaining Orai3 isoform, which is resistant to inhibition by 2-APB. Furthermore, SOCE in Orai3 knockout B cells was inhibited at a significantly faster rate by 2-APB compared to wild-type B cells and was not further inhibited by Gd$^{3+}$ (*Figure 5B and D*). The small amount of SOCE remaining in Orai1/Orai3 double knockout cells was reduced to baseline following the addition of 2-APB. This remaining SOCE in Orai1/Orai3 double knockout B cells prompted us to measure SOCE in B cells isolated from global Orai2 knockout mice (*Orai2$^{-/-}$*). B cells from *Orai2$^{-/-}$* mice showed complete loss of *Orai2* mRNA with no apparent compensation in *Orai1* or *Orai3* mRNA (*Figure 5E*). SOCE stimulated by thapsigargin in B cells from *Orai2$^{-/-}$* mice was enhanced by comparison to B cells from wild-type littermate controls (*Figure 5F and G*), suggesting that, as was shown in T cells (*Vaeth et al., 2017b*), Orai2 is also a negative regulator of SOCE in B cells. Thus, these experiments indicate that the remaining SOCE in B cells from Orai3/Orai1 double knockout mice is most likely mediated by Orai2.

Given the modulation of Orai channel isoforms in response to B cell activation, we performed Ca$^{2+}$ imaging recordings, similar to those in *Figure 5*, on primary B cells that were first activated for 48 hr with anti-IgM +anti-CD40 (*Figure 5—figure supplement 1A–C*). While similar trends were observed in experiments with naïve primary B cells, several exceptions were notable. SOCE in Orai3 knockout B cells was reduced compared to control cells, like data in A20 Orai knockout cell lines (*Figure 5—figure supplement 1A, B*). Differences in inhibition by 50 µM 2-APB also became less pronounced between control and Orai3 knockout cells (*Figure 5—figure supplement 1A–C*). Collectively, these data demonstrate that Orai3 contributes to SOCE in activated B cells.

## SOCE is an essential regulator of NFAT activation in B cells

Ca$^{2+}$ entry through CRAC channels is critical for the activation of multiple NFAT isoforms (*Prakriya and Lewis, 2015*; *Trebak and Kinet, 2019*; *Vaeth and Feske, 2018*). NFAT nuclear translocation mostly requires Ca$^{2+}$ entry through native Orai1, while native Orai2 and Orai3 caused marginal NFAT activation in cells lacking Orai1 despite mediating a Ca$^{2+}$ signal (*Yoast et al., 2020b*). To determine the role of Orai1 in regulating NFAT1 activation in B cells, primary B cells from control Mb1-Cre$^{/+}$ and *Orai1$^{fl/fl}$* Mb1-Cre$^{/+}$ mice were stimulated with anti-IgM antibodies and native NFAT1 nuclear translocation was evaluated using ImageStream analysis (*Figure 6A–C*). In unstimulated B cells, colocalization of endogenous NFAT1 with nuclear DAPI staining was relatively low, and this colocalization increased three-fold following anti-IgM stimulation (*Figure 6A–C*). However, this anti-IgM mediated NFAT1 translocation was significantly reduced in B cells from *Orai1$^{fl/fl}$* Mb1-Cre$^{/+}$ mice (*Figure 6B and C*).

We used another complimentary biochemical protocol to assess the nuclear translocation of native NFAT1 in response to stimulation with thapsigargin. After B cell stimulation, proteins were harvested and processed for Western blotting using a specific NFAT1 antibody (*Figure 6D–G*). In unstimulated samples, NFAT1 appears as a single band corresponding to its highly phosphorylated, non-activated state (*Figure 6D and F*). Stimulation of naïve B cells with thapsigargin, which completely empties ER stores and maximally activates SOCE, led to a complete shift of the phosphorylated single band into lower molecular weight bands (*Figure 6D and E*, *left*) and this shift in NFAT1 molecular weight was inhibited by 43.6% in Orai1-deficient B cells (*Figure 6D and E*, *right*). Given the dynamic expression of each Orai isoform following B cell activation (*Figure 1G–I*), we also evaluated NFAT1 activation in B cells stimulated for 48 hr with anti-IgM +anti-CD40. While thapsigargin stimulation also led to a complete shift in NFAT1 molecular weight in activated control B cells, this shift was reduced by 78% in activated B cells from *Orai1$^{fl/fl}$* Mb1-Cre$^{/+}$ mice (*Figure 6F and G*). Similar results were obtained when we analyzed NFAT2 isoform dephosphorylation in naïve B cells (*Figure 6—figure supplement 1A*). Furthermore, when naïve B cells were stimulated with anti-IgM (instead of thapsigargin), we observed similar trends of NFAT1 and NFAT2 dephosphorylation, although this dephosphorylation was not as robust as with thapsigargin (*Figure 6—figure supplement 1A*). Nevertheless, this anti-IgM mediated dephosphorylation of NFAT1 and NFAT2 was inhibited in B cells from *Orai1$^{fl/fl}$* Mb1-Cre$^{/+}$ mice (*Figure 6—figure supplement 1A*). Our findings that B cells from *Orai1$^{fl/fl}$* Mb1-Cre$^{/+}$ mice have maintained BCR-induced Ca$^{2+}$ oscillations with

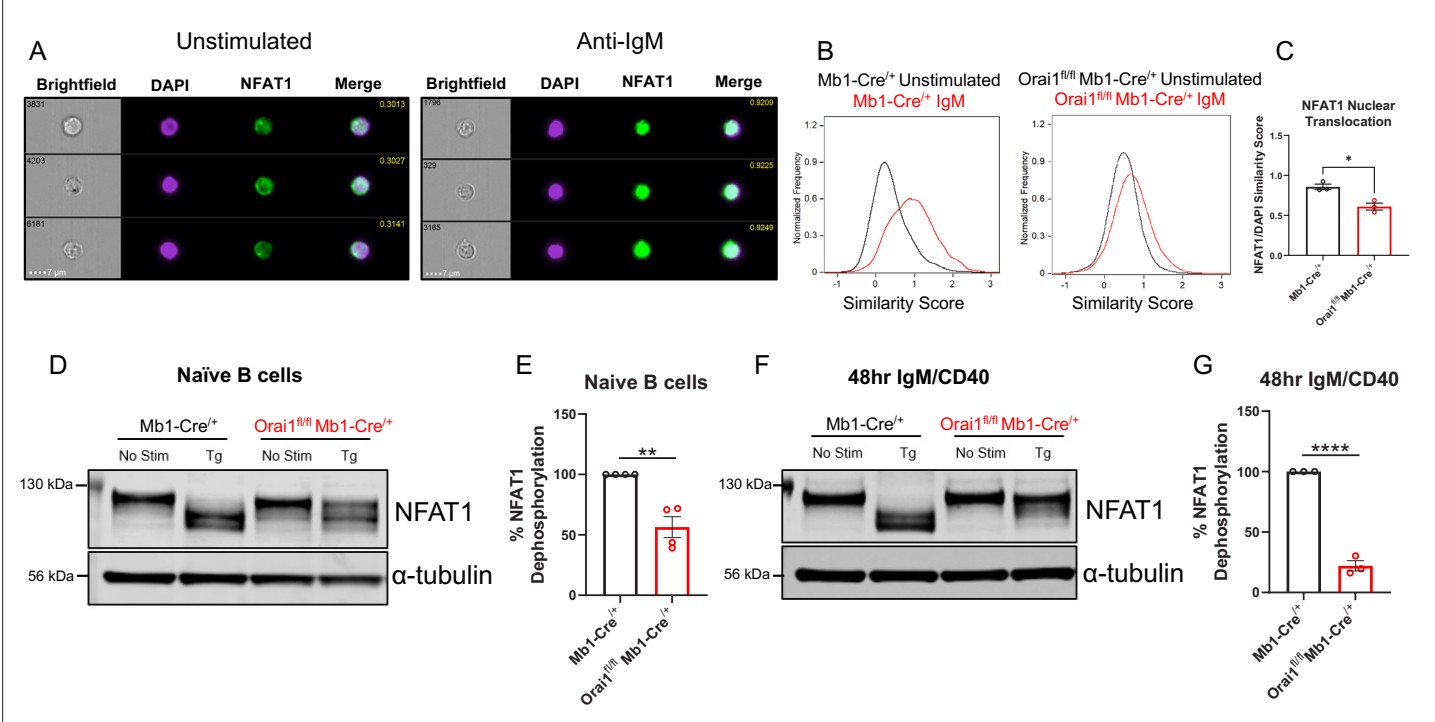

**Figure 6.** Orai1 is a regulator of nuclear factor of activated T cells (NFAT) activation in naïve and activated B cells. (**A**) Representative Imagestream images following intracellular staining for NFAT1 and DAPI in naïve B cells from Mb1-Cre[/+] mice before and after 20 µg/mL anti-IgM stimulation for 15 min. Merge image indicates similarity score co-localization between NFAT1/DAPI. (**B**) Histograms of NFAT1/DAPI similarity scores before (black trace) and after (red trace) anti-IgM stimulation in naïve B cells from Mb1-Cre[/+] and *Orai1*[fl/fl] Mb1-Cre[/+] mice. (**C**) Quantification of similarity scores following anti-IgM stimulation in (**B**) (n=3 biological replicates for each; unpaired T-test). (**D**) Western blot analysis of NFAT1 and α-tubulin in naïve B cells isolated from Mb1-Cre[/+] and *Orai1*[fl/fl] Mb1-Cre[/+] mice. B cells were left unstimulated or treated with 2 µM thapsigargin (Tg) for 15 min before harvesting. (**E**) Quantification of NFAT1 dephosphorylation in (**D**) (n=4 biological replicates for each; Mann-Whitney test). (**F**) Western blot analysis of NFAT1 and α-tubulin in B cells stimulated for 48 hr with anti-IgM +anti-CD40. (**G**) Quantification of NFAT1 dephosphorylation in (**F**) (n=3 biological replicates for each; Mann-Whitney test). All scatter plots are presented as mean ± SEM. For all figures, *p<0.05; **p<0.01; ****p<0.0001.

The online version of this article includes the following source data and figure supplement(s) for figure 6:

**Source data 1.** Source data for *Figure 6* including labeled blots for NFAT1 and α-tubulin from *Figure 6D* and *Figure 6F*.

**Source data 2.** Panel D and F-Raw unedited uncropped blots for NFAT1 and α-tubulin from *Figure 6*.

**Figure supplement 1.** Nuclear factor of activated T cells (NFAT) activation in primary Orai1/Orai3-deficient B cells.

**Figure supplement 1—source data 1.** Labeled western blots for NFAT1, NFAT2, and α-tubulin from *Figure 6—figure supplement 1*.

**Figure supplement 1—source data 2.** Raw unedited uncropped blots for NFAT1, NFAT2, and α-tubulin from *Figure 6—figure supplement 1*.

defects in NFAT activation is in agreement with previous studies in HEK293 cells showing that NFAT activation depends mostly on native Orai1 (*Yoast et al., 2020b*). In these studies, intact Ca²⁺ oscillations in Orai1 knockout cells were mediated by Orai2/3, yet NFAT nuclear translocation was largely reduced (*Yoast et al., 2020b*).

We also evaluated NFAT1 activation in response to thapsigargin in both naïve B cells (*Figure 6—figure supplement 1B*) and activated B cells (48 hr with anti-IgM +anti-CD40; *Figure 6—figure supplement 1C*) isolated from either *Orai3*[fl/fl] Mb1-Cre[/+] or *Orai1/Orai3*[fl/fl] Mb1-Cre[/+] mice. Loss of Orai3 alone from naïve B cells had no effect on NFAT1 activation, while naïve B cells isolated from double Orai1/Orai3 knockout cells showed reductions in NFAT1 activation comparable to B cell isolated from single Orai1 knockout mice (*Figure 6—figure supplement 1B*). Importantly, this impairment of NFAT1 activation was further exacerbated in activated B cells isolated from Orai1 knockout and Orai1/Orai3 double knockout mice (*Figure 6—figure supplement 1C*). Collectively, these data reveal that Orai1 plays a more prominent role in the activation of NFAT isoforms in activated B cells compared to naïve cells.

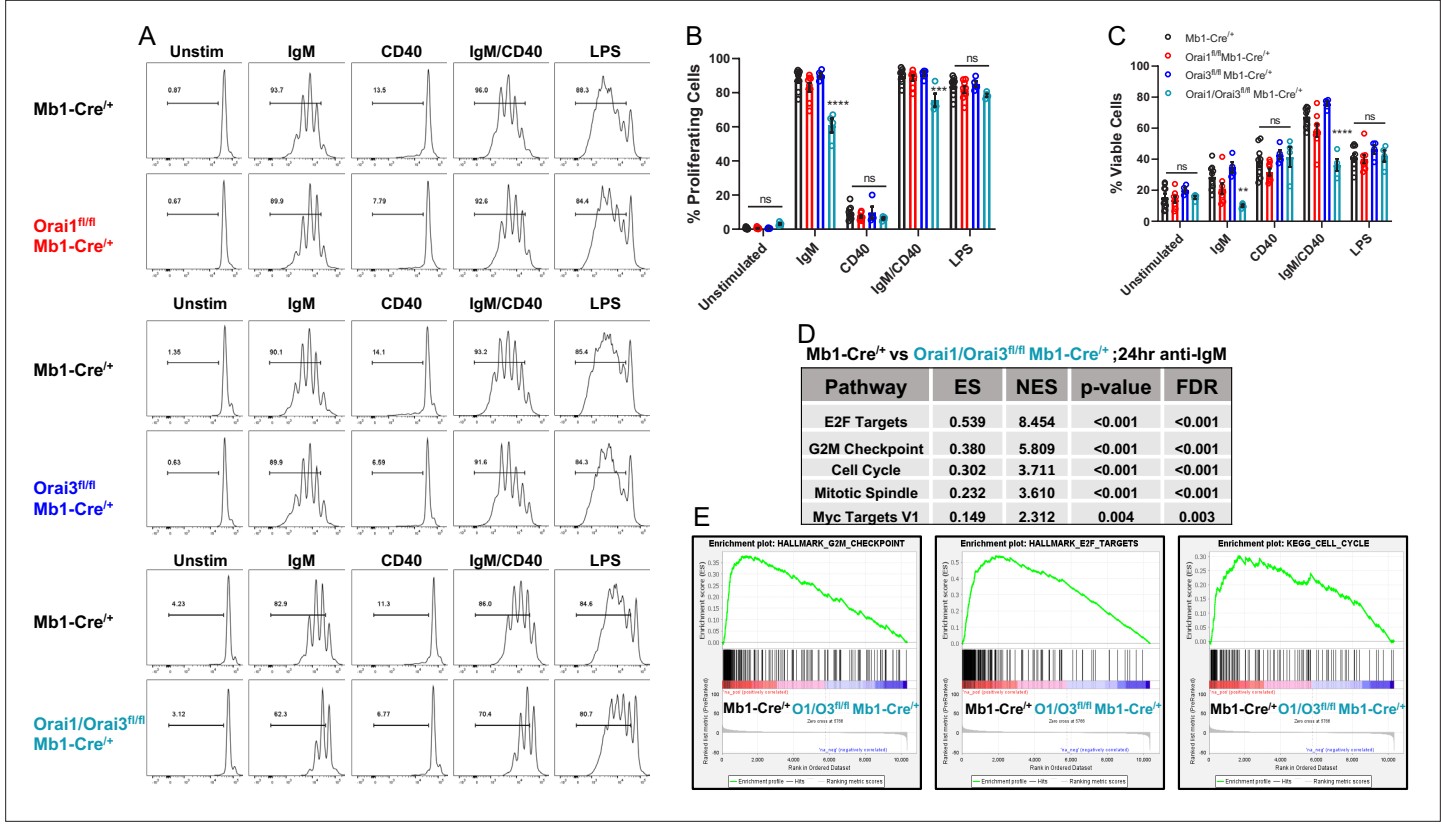

**Figure 7.** Orai1 and Orai3 regulate B cell proliferation and survival. (**A**) Measurement of B cell proliferation by tracking cabroxyfluorescein diacetate succinimidyl ester (CFSE) dilution. B lymphocytes from control, Orai1, Orai3, and Orai1/Orai3 knockout mice were loaded with CFSE (3 μM) and stimulated with anti-IgM (20 μg/mL), anti-CD40 (10 μg/mL), anti-IgM +anti-CD40, or LPS (10 μg/mL). CFSE dilution was determined 72 hr after stimulation for all conditions. (**B**) Quantification of the percentage of proliferating cells for each condition in (**A**) (from left to right n=9, 8, 4, and 4 biological replicates; one-way ANOVA with multiple comparisons to Mb1-Cre[/+]). CFSE dilution gate is drawn relative to unstimulated controls. (**C**) Quantification of the percentage of viable cells for each condition in (**A**) as determined by a Live/Dead viability dye (from left to right n=9, 8, 4, and 4 biological replicates; one-way ANOVA with multiple comparisons to Mb1-Cre[/+]). (**D**) Top KEGG pathways showing differential expression from RNA-sequencing analysis of B cells from Mb1-Cre[/+] vs *Orai1/Orai3[fl/fl]* Mb1-Cre[/+] mice stimulated for 24 hr with anti-IgM (20 μg/mL) (n=3 biological replicates for each). (**E**) Gene set enrichment analysis (GSEA) plots show enrichment statistics (ES) in the Hallmark G2M Checkpoint, E2F Targets, and Cell Cycle gene sets. Large positive ES values suggest activation of these pathways. Normalized enrichment score (NES) values are used to assess statistical significance, and the results for these gene sets are highly significant. All scatter plots are presented as mean ± SEM. For all figures, **p<0.01; ***p<0.001; ****p<0.0001; ns, not significant.

The online version of this article includes the following source data and figure supplement(s) for figure 7:

**Source data 1.** Source data for *Figure 7*.

**Figure supplement 1.** Loss of Orai2 does not alter primary B cell proliferation or viability.

**Figure supplement 1—source data 1.** Source data for *Figure 7—figure supplement 1*.

## The combined deletion of Orai3 and Orai1 inhibits B cell proliferation

BCR-induced Ca[2+] signals that are sustained through SOCE are critical for the activation of gene programs that regulate proliferation and apoptosis (*Berry et al., 2020*; *Matsumoto et al., 2011*). Interestingly, suboptimal proliferation and survival of B cells in response to BCR stimulation can largely be rescued through the addition of secondary co-stimulatory signals (e.g. CD40 or TLR stimulation) (*Berry et al., 2020*; *Matsumoto et al., 2011*; *Tang et al., 2017*). To understand how Ca[2+] signals downstream of Orai isoforms regulate B cell expansion, primary B cells were labeled with carboxyfluorescein diacetate succinimidyl ester (CFSE) and cell divisions tracked in response to multiple conditions of stimulation (*Figure 7A*). BCR stimulation with anti-IgM alone resulted in multiple rounds of cell division in most wild-type cells (87.7%) with 28.6% of viable cells at 72 hr post-stimulation (*Figure 7B and C*, *black*). B cell viability at 72 hr was dramatically increased to 67.4% when cells were co-stimulated with anti-IgM +anti-CD40, with an increase in the number of cells undergoing several cycles of cell

division. Following anti-IgM stimulation Orai1-deficient B cells showed a similar percentage of proliferating cells (83.1%) as controls, with a moderate reduction in cell viability to 20.9% (*Figure 7B and C*, *red*). While anti-IgM mediated proliferation of Orai3-deficient B cells was comparable to controls (90.3%), their viability was higher, at 34.8% (*Figure 7B and C*, *blue*). The combined deletion of Orai1 and Orai3 resulted in a substantial reduction in the percentage of both viable (10%) and proliferating (60.8%) cells in response to anti-IgM stimulation (*Figure 7B and C*, *teal*). These defects in survival and proliferation of Orai1/Orai3-deficient B cells were partially rescued when cells were co-stimulated with anti-IgM +anti-CD40, albeit to a lesser extent than control and single Orai knockout B cells. Similar percentages of viability and proliferation were observed in all experimental groups when B cells were stimulated with either anti-CD40 or LPS, suggesting that activation of the toll-like receptor 4 (TLR4) signaling pathway is able to compensate for the loss of Orai1/Orai3. We also analyzed the proliferation and survival of B cells isolated from Orai2$^{-/-}$ mice and their wildtype littermates exposed to the same stimuli. We found only marginal or no effects of Orai2 knockout on B cell proliferation and survival (*Figure 7—figure supplement 1A, B*). Furthermore, we performed RNA sequencing on B cells isolated from control Mb1-Cre$^{/+}$ and *Orai3/Orai1*$^{fl/fl}$ Mb1-Cre$^{/+}$ mice after stimulation with anti-IgM for 24 hr (*Source data 2*) and differential expression analysis was performed with edgeR (*Source data 3*). Based on the edgeR output, GSEA software was applied to perform, we performed pathway analyses using gene set enrichment analysis (GSEA) (*Source data 4*). In agreement with our proliferation and survival data, GSEA showed significant downregulation of pathways that govern survival and cell cycle progression in Orai1/Orai3-deficient B cells (*Figure 7D and E*). Together, these findings reveal that only loss of both Orai1 and Orai3 affects B cell survival and proliferation in response to anti-IgM stimulation while single knockout of either Orai1 or Orai3 had marginal effects.

## The combined deletion of Orai3 and Orai1 inhibits B cell metabolism

Metabolic reprogramming results in dramatically enhanced OXPHOS and remodeling of the mitochondrial network (*Akkaya et al., 2018*; *Waters et al., 2018*). Indeed, GSEA of B cells stimulated for 24 hr with anti-IgM identified the Oxidative Phosphorylation gene set as one of the most highly enriched pathways relative to unstimulated controls (*Figure 8A*). Additionally, anti-IgM stimulated B cells to demonstrate a significant increase in the total number of mitochondria per cell compared to naïve B cells as determined by transmission electron microscopy (TEM; *Figure 8B and C*). However, this increase in mitochondrial mass triggered by B cell activation was not affected by the depletion of Orai1 from B cells, as determined by Mitotracker Green staining (*Figure 8D*). We measured OCR and ECAR in B cells isolated from control and B cell-specific Orai knockout mice after B cell stimulation with anti-IgM for 24 hr (*Figure 8E*). Anti-IgM stimulation of wild-type B cells led to a robust increase in both OCR and ECAR (*Figure 8F*). Furthermore, Anti-IgM stimulation led to an increase in both basal and maximal respiration of wild-type B cells (*Figure 8E–H*). The increase in OCR and ECAR following BCR activation was significantly blunted in Orai1-deficient B cells, and further reduced in Orai1/Orai3-deficient B cells (*Figure 8E–H*). Interestingly, GSEA analysis showed that anti-IgM mediated enrichment of the Oxidative Phosphorylation gene set and Myc targets gene set were specifically inhibited in B cells from *Orai1/Orai3*$^{fl/fl}$ Mb1-Cre$^{/+}$ mice, but not in B cells from *Orai1*$^{fl/fl}$ Mb1-Cre$^{/+}$ mice (*Figure 8—source data 1*). However, the glycolysis gene set was inhibited in B cells from both *Orai1/Orai3*$^{fl/fl}$ Mb1-Cre$^{/+}$ mice and *Orai1*$^{fl/fl}$ Mb1-Cre$^{/+}$ mice (*Figure 8—source data 1*). Loss of Orai3 alone only partially reduced glycolytic flux and basal respiration but did not affect maximal respiration (*Figure 8E–H*). Previous research has shown that loss of SOCE in the chicken DT40 B cell line impaired mitochondrial metabolism by reducing CREB-mediated expression of the mitochondrial calcium uniporter (MCU) (*Shanmughapriya et al., 2015*). We observed no differences in CREB phosphorylation on Ser133 (a surrogate for CREB activation) or MCU expression in primary Orai1-deficient B cells from *Orai1*$^{fl/fl}$ Mb1-Cre$^{/+}$ mice (*Figure 8—figure supplement 1A*). Likewise, we observed no differences in MCU expression in mouse A20 B cells lacking either Orai1, Orai3, or both Orai1 and Orai3 (*Figure 8—figure supplement 1B*). Both naïve and activated (24 hr with anti-IgM) B cells from *Orai1*$^{fl/fl}$ Mb1-Cre$^{/+}$ mice showed no obvious changes in the expression of different components of the electron transport chain (*Figure 8—figure supplement 1C, D*).

To further investigate how Orai-mediated Ca$^{2+}$ signaling regulates B cell metabolism, we profiled polar metabolites utilizing liquid chromatography followed by mass spectrometry in control and Orai1/Orai3-deficient B cells (*Figure 9*). Isolated B cells from each genotype were either unstimulated

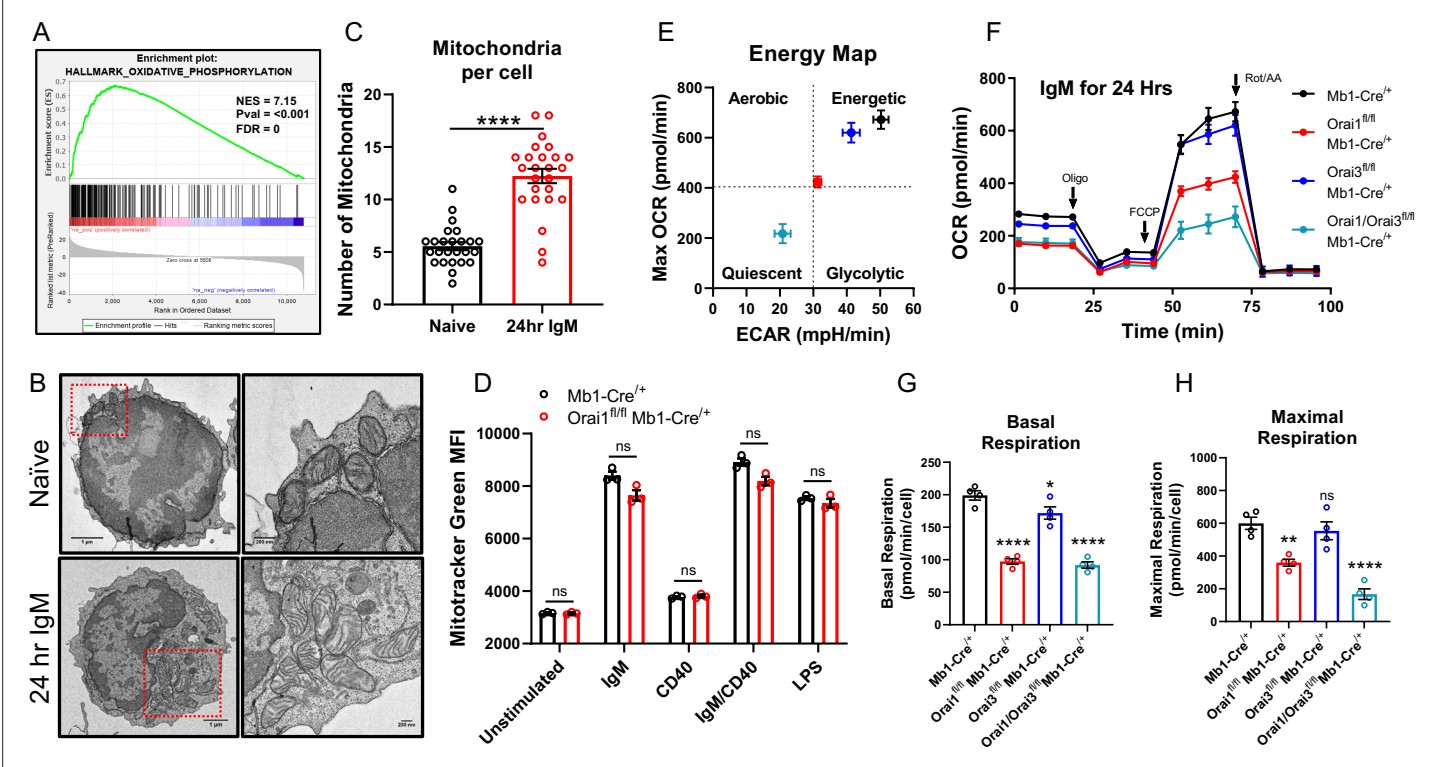

**Figure 8.** Orai1 and Orai3 regulate B cell mitochondrial respiration. (**A**) Gene set enrichment analysis (GSEA) of the KEGG Oxidative Phosphorylation gene set in B cells stimulated for 24 hr with anti-IgM (20 μg/mL) relative to unstimulated controls. (**B**) Representative transmission electron microscopy (TEM) images of B cells from Mb1-Cre[/+] mice. Shown are naïve, unstimulated B cells (top) and B cells stimulated for 24 hr with anti-IgM (bottom). (**C**) Quantification of total mitochondria per cell in unstimulated B cells and B cells stimulated for 24 hr with anti-IgM (n=25 for each; Mann-Whitney test). (**D**) Measurement of total mitochondrial content with MitoTracker Green in B cells from Mb1-Cre[/+] and *Orai1[fl/fl]* Mb1-Cre[/+] mice following 24 hr stimulation (n=3 biological replicates for each; Mann-Whitney test). (**E**) Measurement of oxygen consumption rate (OCR) in primary B lymphocytes following 24 hr stimulation with anti-IgM (20 μg/mL) using the Seahorse Mito Stress Test. (**F**) Energy map of maximal oxygen consumption rates (OCR) and extracellular acidification rates (ECAR) following carbonyl cyanide p-trifluoromethoxyphenylhydrazone (FCCP) addition. (**G, H**) Quantification of basal (**G**) and maximal (**H**) respiration from Seahorse traces in (**E**) (n=4 biological replicates for each genotype; one-way ANOVA with multiple comparisons to Mb1-Cre[/+]). All scatter plots and Seahorse traces are presented as mean ± SEM. For all figures, **p<0.01; ****p<0.0001; ns, not significant.

The online version of this article includes the following source data and figure supplement(s) for figure 8:

**Source data 1.** Source data for GSEA summary statistics for *Figure 8*.

**Source data 2.** Source data for *Figure 8*.

**Figure supplement 1.** Loss of Orai1 does not alter CREB phosphorylation, MCU expression, or expression of electron transport chain proteins.

**Figure supplement 1—source data 1.** Labeled western blots for *Figure 8—figure supplement 1*.

**Figure supplement 1—source data 2.** Raw unedited uncropped blots for GAPDH and MCU from *Figure 8—figure supplement 1A*.

**Figure supplement 1—source data 3.** Raw unedited uncropped blots for Phospho-CREB (pCREB) from *Figure 8—figure supplement 1A*.

**Figure supplement 1—source data 4.** Raw unedited uncropped blots for MCU and GAPDH from *Figure 8—figure supplement 1B*.

**Figure supplement 1—source data 5.** Raw unedited uncropped blots for electron transport chain (ETC) proteins from *Figure 8—figure supplement 1C*.

**Figure supplement 1—source data 6.** Raw unedited uncropped blots for GAPDH from *Figure 8—figure supplement 1C*.

**Figure supplement 1—source data 7.** Raw unedited uncropped blots for electron transport chain (ETC) proteins and GAPDH from *Figure 8—figure supplement 1D*.

or stimulated for 24 hr with anti-IgM alone, anti-IgM +anti-CD40, anti-IgM +the calcineurin inhibitor FK506, or anti-IgM +the CRAC channel inhibitor GSK-7975A. We utilized GSK-7975A as we recently demonstrated that this compound inhibited all Orai isoforms compared to differential effects on Orai isoforms with other common SOCE inhibitors like Synta66 (*Zhang et al., 2020*). Stimulation

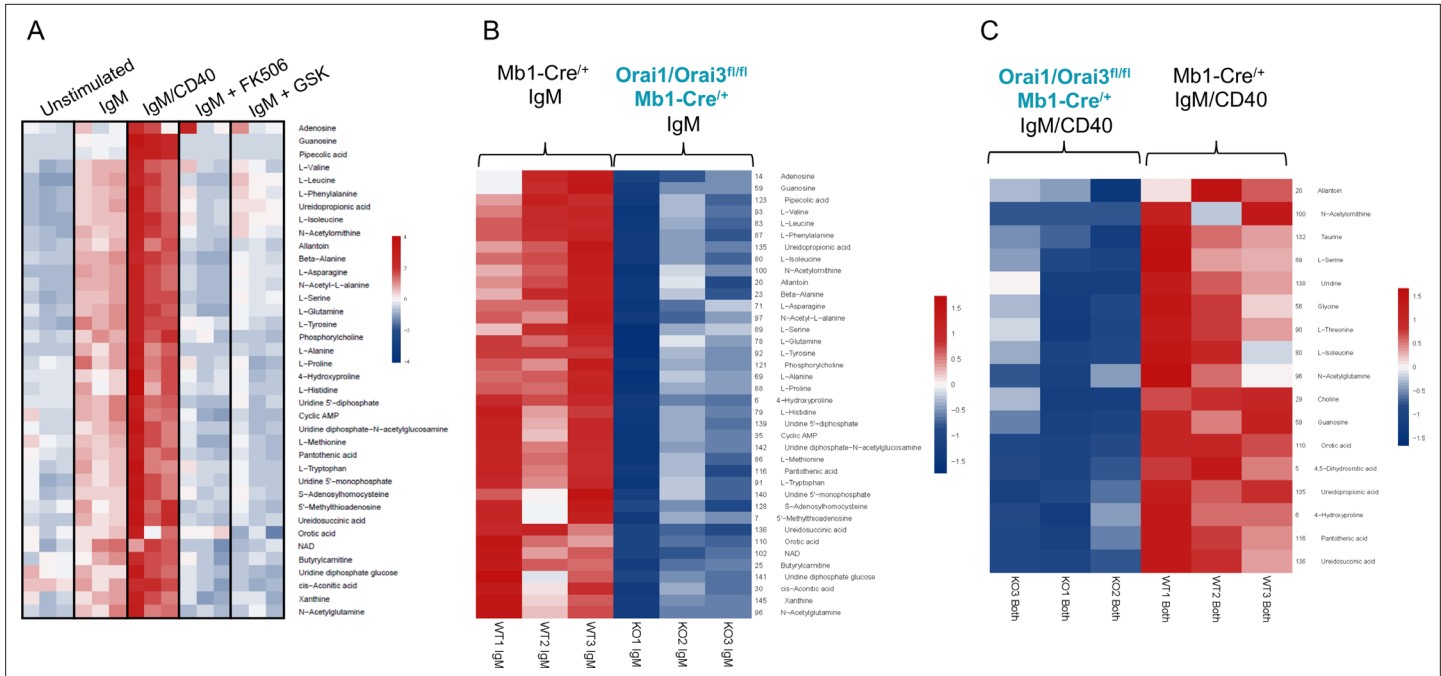

**Figure 9.** Orai1/Orai3-mediated SOCE-calcineurin-NFAT pathway regulates B cell metabolism. (**A**) Analysis of polar metabolites in B cells from Mb1-Cre$^{/+}$ mice utilizing liquid chromatography followed by mass spectrometry. B cells were either unstimulated or stimulated for 24 hr with anti-IgM, anti-IgM +anti-CD40, anti-IgM with 1 μM FK506, or anti-IgM with 10 μM GSK-7975A. (**B, C**). Heat maps of statistically significant polar metabolites in B cells from Mb1-Cre$^{/+}$ and *Orai1/Orai3*$^{fl/fl}$ Mb1-Cre$^{/+}$ mice following (**B**) 24 hr anti-IgM stimulation or (**C**) 24 hr anti-IgM +anti-CD40 stimulation. (n=3 biological replicates for each condition).

The online version of this article includes the following source data and figure supplement(s) for figure 9:

**Source data 1.** Source data for *Figure 9*.

**Figure supplement 1.** Polar metabolite analysis in Orai1/Orai3 deficient-B cells.

**Figure supplement 1—source data 1.** Source data for *Figure 9—figure supplement 1*.

---

of control B cells with anti-IgM led to a significant increase of glycolytic and TCA cycle metabolites along with most non-essential amino acids. This effect was further enhanced with anti-IgM +anti-CD40 co-stimulation (*Figure 9A*; *Figure 9—figure supplement 1A*). Inclusion of FK506 or GSK-7975A strongly blunted the effects of anti-IgM activation as overall metabolic profiles remained like those of unstimulated cells (*Figure 10A*). Importantly, this upregulation in polar metabolites upon B cell activation was significantly reduced in Orai1/Orai3-deficient B cells with either anti-IgM stimulation or anti-IgM +anti-CD40 co-stimulation (*Figure 9B and C*; *Figure 9—figure supplement 1B*). Of note, upregulation of most polar metabolites from Orai1/Orai3 knockout B cells co-stimulated with anti-IgM +anti-CD40 typically reached levels comparable to those of control B cells stimulated with anti-IgM alone. The inhibitory effects of FK506 and GSK-7975A on the metabolite status of wild-type B cells were comparable to those of Orai1/Orai3-deficient B cells (*Figure 9A*, *Figure 9—figure supplement 1A*). Collectively, these results reveal that SOCE mediated through Orai1/Orai3 channels contributes to the metabolic reprogramming of B lymphocytes.

## Immunity to influenza A virus is intact in mice with B cell-specific deletion of Orai1 and Orai3

Given the important role of Orai1 and Orai3 for multiple B cell functions in vitro, we hypothesized that deletion of *Orai1* and *Orai3* affects the B cell cytokine profile and compromises immune responses to influenza A virus (IAV) infection. Profiling of B cell cytokines and cytokine receptors by RNA-sequencing of B cells from *Orai1*$^{fl/fl}$*Orai3*$^{fl/fl}$ Mb1-Cre$^{/+}$ and littermate control mice stimulated ex vivo with anti-IgM showed no obvious differences between the two groups (*Figure 10—figure supplement 1C–D*). To evaluate the role of Orai1/Orai3 for B cell function and humoral immunity in vivo, we infected *Orai1*$^{fl/}$

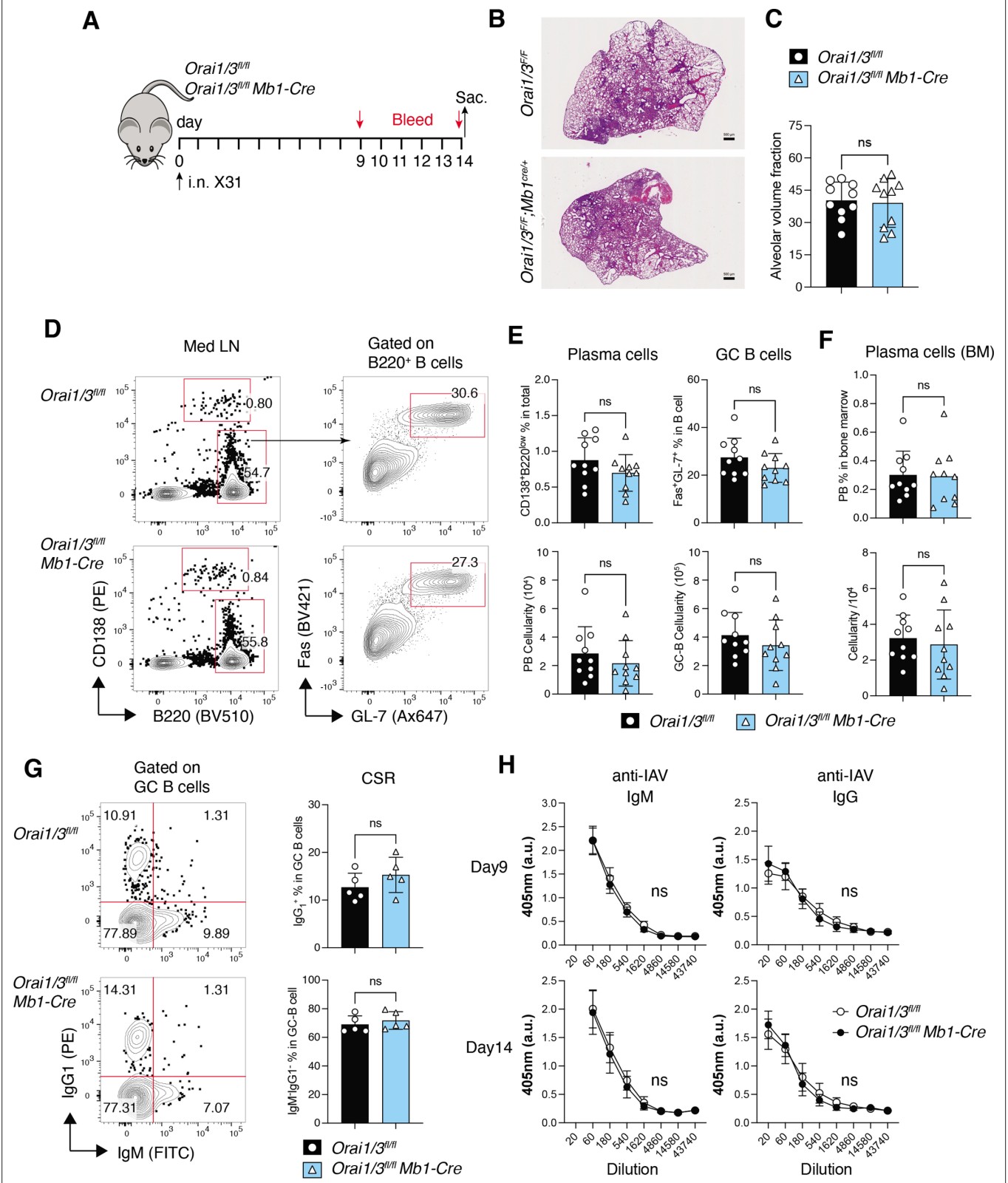

**Figure 10.** Deletion of *Orai1* and *Orai3* in B cells does not compromise immunity to influenza A virus (IAV). (**A**) Experimental outline. Littermate controls and *Orai1^{fl/fl}Orai3^{fl/fl}* Mb1-Cre^{/+} mice have infected intranasally with 1×10^5 TCID_{50} of the x31 H3N2 strain of influenza A virus. Serum was collected on days 9 and 14, and mice were sacrificed on day 14 for analysis. (**B**) Representative H&E stains of lung sections. Scale bar: 500 μm. (**C**) Alveolar volume fraction of 10 mice per cohort. (**D**) Representative flow cytometry plots of B cells isolated from mediastinal lymph nodes (med LN). (**E**) Summary

*Figure 10 continued on next page*

*Figure 10 continued*

of the frequencies (%) and total cell numbers of B220⁻CD138⁺ plasma cells and Fas⁺GL-7⁺ GC B cells shown in panel D from 10 mice per cohort. (**F**) Representative flow cytometry plots of plasma cells isolated from the bone marrow of five IAV-infected mice per cohort. (**G**) Representative flow cytometry plots (left) and summary (right) of the frequencies of class-switched IgG1⁺ GC B cells in med LN of five IAV-infected mice per cohort. (**H**) IAV-specific IgM and IgG levels in the serum of 10 mice per cohort were measured on days 9 and 14. Panels B-H show the results of two independent experiments. Statistical analysis by unpaired Student's t-test: ***$p<0.001$, **$p<0.01$, *$p<0.05$.

The online version of this article includes the following source data and figure supplement(s) for figure 10:

**Source data 1.** Source data for *Figure 10*.

**Figure supplement 1.** Cytokine profiling of B cells from *Orai1/Orai3*$^{fl/fl}$ Mb1-Cre$^{/+}$ mice.

---

$^{fl}$*Orai3*$^{fl/fl}$ Mb1-Cre$^{/+}$ mice and *Orai1*$^{fl/fl}$*Orai3*$^{fl/fl}$ littermate controls intranasally with a single dose of the laboratory strain A/HK/x31 (Hkx31, H3N2) (*Thomas et al., 2006*) and analyzed immune responses 14 days later (*Figure 10A*). All infected WT and *Orai1/3*-deficient mice survived and experienced a similar reduction in body weight over the two week-course of infection (not shown). Lung histology at day 14 post-infection (p.i.) showed comparable pulmonary inflammation and alveolar volume in *Orai1/3*-deficient and littermate control mice (*Figure 10B and C*). To directly investigate the effects of Orai1 and Orai3 on humoral immunity during IAV infection, we analyzed total B cells, germinal center (GC) B cells, and plasma cells in mice at day 14 p.i. The frequencies and numbers of B220⁻CD138⁺ plasma cells and B220⁺Fas⁺GL-7⁺ GC B cells were similar in the mediastinal lymph nodes of WT and *Orai1/3*-deficient mice (*Figure 10D and E*). Moreover, the numbers of CD138⁺ plasma cells in the bone marrow of infected *Orai1*$^{fl/fl}$*Orai3*$^{fl/fl}$ Mb1-Cre$^{/+}$ mice were comparable to those in control littermate mice (*Figure 10F*). We next investigated if deletion of *Orai1* and *Orai3* in B cells affects their ability to induce class switch recombination. The percentages of class-switched IgG1⁺ cells (and those of IgM⁻IgG⁻ non-switched B cells) among GC B cells, however, were comparable in control and *Orai1/3*-deficient mice (*Figure 10G*). Consistent with these results, we found normal levels of IAV-specific IgM and class-switched IgG antibodies in the serum (*Figure 10H*). Taken together, our data show that B cell-specific deletion of *Orai1* and *Orai3* does not significantly impair immune responses to influenza A virus infection.

## Discussion

Given the essential role of B lymphocytes in driving humoral immunity against foreign pathogens, a comprehensive understanding of the molecular pathways that govern their development, differentiation, and effector functions is critical for future targeted therapies. One of the earliest signaling events upon crosslinking of the BCR is a biphasic increase in intracellular Ca²⁺ concentrations (*Baba and Kurosaki, 2016*). Early landmark studies established that multiple Ca²⁺ dependent transcription factors display unique activation requirements by relying on either ER Ca²⁺ release (e.g. JNK, NF-κB) or sustained Ca²⁺ signals driven by SOCE (NFAT) (*Dolmetsch et al., 1997*; *Healy et al., 1998*). While the SOCE-calcineurin-NFAT pathway is well established in the context of B cell effector function, recent reports have shed light on the mechanisms by which SOCE also regulates NF-κB activation and its downstream target genes (*Berry et al., 2020*; *Berry et al., 2018*). These two BCR-activated signaling pathways are sustained by Ca²⁺ entry through CRAC channels and synergize with one another to activate a series of Ca²⁺-regulated checkpoints that determine B cell survival, entry into the cell cycle, and proliferation (*Akkaya et al., 2018*; *Berry et al., 2020*). Unlike recent findings that have established Orai1 and Orai2 as the major Orai isoforms mediating SOCE in T cells (*Vaeth et al., 2017b*), the relative contributions of Orai isoforms to the native CRAC channel in B cells remained, until now, unclear.

Our results herein establish that B cell activation through BCR stimulation alone or co-stimulation with secondary signals like CD40 or TLR ligands significantly enhanced metabolic activity. Concurrently, these stimulation conditions drive dynamic changes in the expression of each Orai isoform. Interestingly, we observe that robust B cell activation with BCR and CD40 co-stimulation results in upregulation of both *Orai1* and *Orai3*, and downregulation of *Orai2*. However, we do not see any apparent increase in SOCE in activated B cells versus naive B cells, suggesting that Orai1 and Orai3 upregulation in activated B cells helps maintain the magnitude of SOCE, commensurate with the enhanced volume of activated B cells. While Orai1 has previously been shown to contribute to the majority of SOCE in B cells under conditions of maximal ER Ca²⁺ depletion (*Gwack et al., 2008*;

*McCarl et al., 2010*), we show using B-cell specific Orai1 knockout mice that Orai1 is dispensable for maintaining cytosolic $Ca^{2+}$ oscillations in response to BCR crosslinking. In agreement with our previous results in HEK293 cells (*Yoast et al., 2020b*), we also show that Orai1 is an essential regulator of NFAT1 and NFAT2 isoforms in B cells and that its role becomes more prominent for NFAT induction in activated B cells compared to naïve unstimulated B cells. By generating CRISPR/Cas9 B cell lines and B-cell specific knockout mice lacking Orai1 or Orai3 individually and in combination, we found that both Orai1 and Orai3 contribute to the native CRAC channels in B cells. However, the oligomeric state of native CRAC channels in B cells and whether Orai1 and Orai3 form homomeric or heteromeric assemblies or both in primary B cells remains an open question.

The combined loss of Orai1 and Orai3, but not either isoform alone, led to a significant reduction in both B cell proliferation and survival. The original study by Gwack et al showed a partial inhibition of proliferation in B cells from *Orai1*[-/-] mice. While we observe only a marginal inhibition of proliferation in B cells from *Orai1*[fl/fl] Mb1-Cre[/+] mice and a moderate reduction in the viability of these cells. The reason for the difference between our results on B cell proliferation and those of *Gwack et al., 2008* is unknown. However, one potential explanation could be that in B cells from *Orai1*[-/-] mice in *Gwack et al., 2008* Orai1 is deleted from early B cell development in mice on a mixed ICR background (note that *Orai1*[-/-] mice on the C57BL/6 background show perinatal lethality) whereas in our *Orai1*[fl/fl] Mb1-Cre[/+] mice, Orai1 is deleted later in B cell development in mice that are on a pure C57Bl/6 background. Another potential explanation could be the complete deletion of the Orai1 gene in the studies of *Gwack et al., 2008* vs the 97% reduction in mRNA expression we observe in B cells from *Orai1*[fl/fl] Mb1-Cre[/+] mice.

We show that SOCE is important for the metabolic reprogramming of B cells by regulating mitochondrial metabolism and the flux of polar metabolites in response to B cell activation. This shift in metabolic profiles in response to B cell activation could be neutralized by inhibition of either calcineurin or CRAC channels with FK506 or GSK-7975A, respectively, suggesting that this metabolic flux is driven through SOCE and NFAT-dependent mechanisms. Our data demonstrate that both Orai1 and Orai3 contribute to CRAC channel activity and to shaping cytosolic $Ca^{2+}$ signaling in B cells. While our data has provided evidence for the contribution of Orai3 to CRAC channels in naïve B cells, this contribution was only apparent within the context of double Orai1 and Orai3 knockout. Our data with B cell-specific Orai3 knockout is consistent with a recent study reporting a lack of SOCE inhibition in B cells, T cells, and macrophages from Orai3 global knockout mice (*Wang et al., 2022*). Our data and the study of *Wang et al., 2022*, showed that Orai3 is highly expressed in B cells compared to other immune cells like T cells. Nevertheless, Orai3 knockout, on its own, has no measurable contribution to SOCE in naïve B cells while moderately reducing SOCE in activated B cells. This reduced SOCE in activated B cells from Orai3 knockout mice is consistent with the reduced basal respiration of B cells stimulated with anti-IgM for 24 hr. This could also explain why the double knockout of Orai1 and Orai3, which inhibits SOCE more dramatically, has a larger effect on mitochondrial respiration and metabolomic reprogramming of B cells. Alternatively, it is possible that Orai3 couples to alternative cytosolic signaling pathways that synergize with SOCE-mediated $Ca^{2+}$ influx. Orai3 could mediate its effects on mitochondrial respiration through the store-independent $Ca^{2+}$ influx pathway activated by arachidonic acid or its metabolite, leukotriene$C_4$ (*Thompson et al., 2013*; *Zhang et al., 2015*; *Zhang et al., 2013*; *Zhang et al., 2018*; *Zhang et al., 2014*). Earlier studies investigating human effector T cells demonstrated that Orai3 becomes upregulated in response to oxidative stress, which may act as a potential mechanism to maintain a threshold of $Ca^{2+}$ influx and T cell function in various inflammatory conditions (*Bogeski et al., 2010*). In support of this model, new findings have demonstrated that Orai3 expression is increased in CD4[+] T cells from patients with rheumatoid arthritis and psoriatic arthritis and that silencing of Orai3 reduces tissue inflammation in a human synovium adoptive transfer model (*Ye et al., 2021*). Interestingly, these effects were proposed to be through the store-independent function of Orai3 (*Thompson et al., 2013*; *Zhang et al., 2015*; *Zhang et al., 2013*; *Zhang et al., 2018*; *Zhang et al., 2014*). However, a recent study showed that Orai3 deficient mice have no altered susceptibility in a model of collagen-induced arthritis (*Wang et al., 2022*), challenging a role for Orai3, on its own, in the immune system.

The situation in T cells is different as the combined loss of Orai1 and Orai2 in murine T cells led to a near complete ablation of SOCE and substantially impaired T cell function (*Vaeth et al., 2017b*). Whether Orai3 contributes to the small, residual amount of SOCE in Orai1/Orai2-deficient T cells and/

or mediates store-independent signaling functions in murine T and B cells is unknown. Furthermore, differences in Orai channel contributions are apparent between mice and humans, as SOCE and CRAC currents are completely inhibited in T cells from patients with LoF mutations in *Orai1*, while SOCE is only partially inhibited in Orai1-deficient murine T cells (**Feske et al., 2006**; **Lian et al., 2018**; **McCarl et al., 2009**; **Vig et al., 2008**). Curiously, B cells from *Orai1* LoF mutation patients still retain residual SOCE, suggesting that another Orai isoform may contribute to SOCE in human B cells than in T cells (**Feske et al., 2001**), in agreement with our data herein on mouse B cells. Thus, the composition of native CRAC channels among lymphocytes appears cell-type and context-specific.

While SOCE is substantially reduced in Orai1/Orai3-deficient B cells, defects in their ability to proliferate and survive in response to antigenic stimulation are not as severe as the phenotype observed in STIM1/STIM2-deficient B cells (**Berry et al., 2020**; **Matsumoto et al., 2011**). Our data establish additivity between Orai1 and Orai3 channels in controlling B cell survival and proliferation, but we cannot exclude a potential role for Orai2 in the regulation of SOCE in B cells. Quite the opposite, we show that Orai2 is functional in B cells by documenting that B cells from Orai2$^{-/-}$ mice have enhanced SOCE. This suggests that Orai2 acts as a negative regulator of SOCE in B cells as was shown for T cells (**Vaeth et al., 2017b**), mast cells (**Tsvilovskyy et al., 2018**), and HEK293 cells (**Yoast et al., 2020b**), and that the remaining SOCE activity in B cells from Orai1/Orai3 double knockout mice is likely mediated by Orai2. This is consistent with our results showing that the SOCE inhibitor GSK-7975A inhibits the upregulation of polar metabolites by B cells, in response to anti-IgM +anti-CD40 stimulation, to a greater extent than Orai1/Orai3 double knockout. Therefore, the development of Orai triple knockout mice and additional Orai knockout pairs is needed to further clarify this issue. Similarly, how SOCE and the composition of the native CRAC channel vary among different B cell subsets and during pathological conditions is completely unknown. Indeed, earlier studies have demonstrated that effector populations like germinal center B cells display unique metabolic and Ca$^{2+}$ signaling requirements (**Khalil et al., 2012**; **Luo et al., 2018**). Our data also suggest that the expression of each Orai isoform is dynamic in response to B cell activation, similar to the case of naïve vs effector T cells (**Vaeth et al., 2017b**).

Considering the clear effects of B cell-specific deletion of Orai1 and Orai3 on SOCE, NFAT activation, metabolic reprogramming, survival, and proliferation, it was surprising that humoral immunity to influenza A virus (IAV) infection was unaffected in *Orai1$^{fl/fl}$Orai3$^{fl/fl}$* Mb1-Cre$^{/+}$ mice. This raises the question regarding the role of Orai1, Orai3, and SOCE in general for B cell function and humoral immunity in vivo. Our results are, however, consistent with those from B cell-specific deletion of *Stim1* and *Stim2* genes, which was shown to result in abolished SOCE causing impaired expression of two key anti-apoptotic genes and blunted activation of the mTORC1 and c-Myc metabolic signaling pathways, resulting in decreased B cell survival and proliferation (**Berry et al., 2020**). While this study did not investigate the effects of *Stim1/Stim2* deletion on B cell function in vivo, Baba and colleagues had reported normal B cell development as well as T-dependent and T-independent antibody responses following immunization of mice with B cell-specific deletion of *Stim1* and *Stim2* genes (**Matsumoto et al., 2011**). This was surprising, because *Stim1/Stim2* deletion almost completely abolished thapsigargin and anti-IgM induced SOCE, and strongly impaired B cell survival and proliferation in vitro (**Matsumoto et al., 2011**). A potential explanation for the normal humoral immune responses in *Orai1$^{fl/fl}$Orai3$^{fl/fl}$* Mb1-Cre$^{/+}$ and *Stim1$^{fl/fl}$Stim2$^{fl/fl}$* Mb1-Cre$^{/+}$ mice is provided by the fact that, in contrast to anti-IgM stimulation, stimulation of *Stim1/Stim2*-deficient B cells *via* CD40 or TLR4 results in normal B cell survival and proliferation (**Matsumoto et al., 2011**). When combined with anti-IgM stimulation, CD40 was able to rescue the proliferation defect of *Stim1/Stim2*-deficient B cells. A similar rescue of defective B cell proliferation by CD40 and TLR9 agonists in *Stim1/Stim2*-deficient B cells was reported by Berry at al. (**Berry et al., 2020**). Our data with *Orai1/Orai3*-deficient B cells show only a partial rescue of B cell proliferation by anti-IgM and anti-CD40 co-stimulation compared to anti-IgM alone. We did, however, observe normal B cell survival and proliferation in *Orai1/Orai3*-deficient B cells stimulated with the TLR4 agonist LPS. Collectively, these data demonstrate that co-stimulatory pathways such as CD40, TLR4, and TLR9 can bypass the requirement for BCR-induced SOCE to induce B cell activation. IAV is a dsRNA virus that is detected by TLR7 in plasmacytoid dendritic cells and B cells (**Iwasaki and Pillai, 2014**). TLR7 is widely expressed in B cells and antibody responses to IAV were shown to depend on TLR7 (**Geeraedts et al., 2008**). Although we did not investigate if TLR7 stimulation with synthetic agonists such as imiquimod can rescue the proliferation and survival defects of

*Orai1/Orai3*-deficient B cells in vitro, we speculate that it would, given the fact that TLR7 and TLR4, whose activation by LPS rescues the function of *Stim1/Stim2* and *Orai1/Orai3*-deficient B cells, signal through the same Myd88-IRAK4-TRAF6 pathway to activate NF-κB and other transcription factors (*Duan et al., 2022*). Activation of TLR7 and potentially other costimulatory pathways, in *Orai1/Orai3*-deficient B cells is, therefore, a likely explanation for their normal humoral immune response to IAV in vivo. Although $Ca^{2+}$ and SOCE are clearly important for B cell function, B cells have developed other $Ca^{2+}$-independent means to ensure their proper activation. Future studies with *Orai1*[fl/fl]*Orai3*[fl/fl] Mb1-Cre[/+] mice might reveal indispensable in vivo contributions of SOCE to B cell populations such as memory B cells or to immune disease in other in vivo models such as the experimental autoimmune encephalomyelitis (EAE) mouse model. Taken together, our results uncover additive functions of Orai3 and Orai1 in B cell $Ca^{2+}$ signaling that regulates NFAT activation, metabolism, survival, and proliferation of B cells.

## Methods

### Mice

All animal experiments were carried out in compliance with the Institutional Animal Care and use Committee (IACUC) guidelines of the Pennsylvania State University College of Medicine. All mice were housed under specific pathogen-free conditions and experiments were performed in accordance with protocols approved by the IACUC of the Pennsylvania State University College of Medicine. Mb1-Cre mice (*Hobeika et al., 2006*) were obtained from The Jackson Laboratory (strain#: 020505; https://www.jax.org/strain/020505). *Orai1*[fl/fl] mice (*Ahuja et al., 2017*) were obtained from Dr. Paul Worley (Johns Hopkins University). *Orai3*[fl/fl] mice were generated by our laboratory through the MMRC at the University of California Davis. A trapping cassette was generated including 'SA-βgeo-pA' (splice acceptor-beta-geo-polyA) flanked by Flp-recombinase target 'FRT' sites, followed by a critical *Orai3* coding exon flanked by Cre-recombinase target 'loxP' sites. This cassette was inserted within an intron upstream of the *Orai3* critical exon, where it tags the *Orai3* gene with the lacZ reporter. This creates a constitutive null *Orai3* mutation in the target *Orai3* gene through efficient splicing to the reporter cassette resulting in the truncation of the endogenous transcript. Mice carrying this allele were bred with FLP deleter C57BL/6 N mice to generate the *Orai3*[fl/fl] mouse. All experiments were performed with 8–12 week-old age and sex-matched mice.

### Cell culture

Parental A20 cells were purchased from ATCC (Catalog # TIB-208) and cultured in RPMI 1640 with L-glutamine supplemented with 10% fetal bovine serum and 1 X Antibiotic-Antimycotic and were routinely tested for lack of mycoplasma contamination (*Emrich et al., 2022a*). Naïve splenic B lymphocytes were purified by negative selection using the EasySep Mouse B Cell Isolation Kit (STEMCELL Technologies). Primary B cells from each transgenic mouse line were cultured in RPMI 1640 with L-glutamine supplemented with 10% fetal bovine serum, 1 X GlutaMAX, 1 X Penicillin-Streptomycin Solution, sodium pyruvate (1 mM), 2-ME (5 uM), and HEPES (10 mM). Primary lymphocytes were stimulated with F(ab')$_2$ Fragment Goat Anti-Mouse IgM antibody (Jackson ImmunoResearch, 20 µg/mL), anti-mouse CD40 antibody (BioXCell, 10 µg/mL), or LPS (Sigma-Aldrich, 10 µg/mL). All cell lines and primary lymphocytes were cultured at 37 °C in a humidified incubator with 5% $CO_2$.

### Generation of A20 Orai CRISPR/Cas9 knockout cells

For the generation of A20 Orai knockout clones, we used a similar strategy to our previous studies (*Emrich et al., 2021*; *Yoast et al., 2021*; *Yoast et al., 2020b*; *Zhang et al., 2019*). Briefly, two gRNAs targeting the beginning and end of the mouse *Orai1* or *Orai3* coding region were cloned into two fluorescent vectors (pSpCas9(BB)–2A-GFP and pU6-(BbsI)_CBh-Cas9-T2AmCherry; Addgene). For *mOrai1* knockout, the gRNA sequences are the following: mOrai1n: 5'-GCCTTCGGATCCGGTGC GTC-3'; mOrai1c: 5'-CACAGGCCGTCCTCCGGACT-3'. For *mOrai3* knockout, the gRNA sequences are the following: mOrai3n: 5'- GCGTCCGTAACTGTTCCCGC-3'; mOrai3c: 5- GAAGGAGGTCTGT CGATCCC-3'. A20 cells were electroporated with both N- and C-terminal gRNA combinations with an Amaxa Nucleofector II and single cells with high GFP and mCherry expression were sorted at one cell per well into 96-well plates 24 hr after transfection using a FACS Aria SORP Cell Sorter. Individual

clones were obtained and genomic DNA was tested using primers targeting the N- and C-terminal gRNA cut sites to resolve wild-type and knockout PCR products. Knockout PCR products were cloned into pSC-B-amp/kan with the StrataClone Blunt PCR Cloning Kit (Agilent) and Sanger sequenced to determine the exact deletion. Knockout clones were also confirmed for the absence of mRNA with qRT-PCR and functionally through $Ca^{2+}$ imaging experiments.

## Fluorescence imaging

A20 cell lines and primary B cells were seeded onto poly-L-lysine (Sigma-Aldrich) coated coverslips. Coverslips were mounted in Attofluor cell chambers (Thermo Scientific) and loaded with 2 µM Fura-2AM (Molecular Probes) in a HEPES-buffered saline solution (HBSS) containing 120 mM NaCl, 5.4 mM KCl, 0.8 mM $MgCl_2$, 1 mM $CaCl_2$, 20 mM HEPES, 10 mM D-glucose, at pH 7.4 for 30 min at room temperature. Following Fura-2 loading, cells were washed three times with HBSS and mounted on a Leica DMi8 fluorescence microscope. Fura-2 fluorescence was measured every 2 s by excitation at 340 nm and 380 nm using a fast shutter wheel and the emission at 510 nm was collected through a 40 X fluorescence objective. Fluorescence data was collected from individual cells on a pixel-by-pixel basis and processed using Leica Application Suite X. All cytosolic $Ca^{2+}$ concentrations are presented as the ratio of $F_{340}/F_{380}$.

## Flow cytometry

Spleens were processed into single-cell suspensions and stained using the following antibodies: B220-BV605 (RA3-6B2, Biolegend), CD86-PE/Cy5 (GL-1, Biolegend), MHC-II-PE/Cy7 (M5/114.15.2, Biolegend), Orai1 rabbit polyclonal (*Gwack et al., 2008*; *McCarl et al., 2009*), CD3e-PE (145–2 C11, BD Biosciences), CD8a-V500 (53–6.7, BD Biosciences), CD4-BB700 (RM4-5, BD Biosciences), APC–anti-CD24 (HSA; M1/69), FITC–anti-CD23 (B3B4), PE–anti-IgM (eB121-15F9, eBioscience), APC–anti-CD93 (AA4.1, eBioscience), PE–Cy5-streptavidin (Biolegend), and Pacific blue–anti-B220 (RA3-6B2, Biolegend). All staining was performed in FACS buffer (DPBS, 2% FBS, 1 mM EDTA) for 30 min at 4 °C. Prior to surface staining, all cells were stained with eBioscience Fixable Viability Dye eFluor 780 (Thermo Fisher) and anti-CD16/32 Fc block (2.4G2, Tonbo Biosciences). Measurement of mitochondrial content was performed using MitoTracker Green (Molecular Probes) and mitochondrial membrane potential was performed using TMRE (Molecular Probes). For all CFSE dilution experiments, primary B lymphocytes were stained with 3 µM CFSE (C1157, Thermo Fischer) in PBS with 5% FBS for 5 min at room temperature, followed by the addition of FBS and two washes in PBS before resuspension in complete RPMI media. CFSE dilution of labeled cells was assessed 72 hr after plating and stimulation. All flow cytometry data were collected on a BD LSR II flow cytometer using FACSDiva software (BD Biosciences) and analyzed with FlowJo 9.9.6 software (Tree Star).

## Analysis of NFAT nuclear translocation

Primary B cells were purified using the EasySep Mouse B Cell Isolation Kit (STEMCELL Technologies) and stimulated with anti-IgM (20 µg/mL) for 15 min at room temperature in complete RPMI media. Cells were fixed and permeabilized using the Foxp3 /Transcription Factor Staining Buffer Set (eBioscience) following the manufacturer's instructions. Cells were stained in 1 X permeabilization buffer for 1 hr at room temperature with NFAT1-Alexa Fluor 488 (D43B1, Cell Signaling Technology). Nuclei were stained with DAPI prior to acquisition on an Amnis ImageStream X Mark II Imaging Flow Cytometer and analyzed with IDEAS software using the "Nuclear Localization" feature (EMD Millipore).

## Western blot analysis

A20 and primary B lymphocytes were harvested from the culture and lysed for 10 min in RIPA buffer (150 mM NaCl, 1.0% IGEPAL CA-630, 0.5% sodium deoxycholate, 0.1% SDS, 50 mM Tris, pH 8.0; Sigma) containing 1 X Halt protease/phosphatase inhibitors (Thermo Scientific). Following lysis, samples were clarified by centrifugation at 15,000 × g for 10 min at 4°C. Supernatants were collected and protein concentration was determined using the Pierce Rapid Gold BCA Protein Assay Kit (Thermo Scientific). Equal concentrations of protein extract were loaded into 4–12% NuPAGE BisTris gels (Life Technologies) and transferred to PVDF membranes utilizing the Transblot Turbo Transfer System (Bio-Rad). Membranes were blocked for 1 hr at room temperature in Odyssey Blocking Buffer in TBS (LI-COR) and incubated overnight at 4°C with primary antibody. The following antibodies and dilutions

were used: MCU (1:2000; 14997 S, Cell Signaling Technology), GAPDH (1:5000; MAB374, Sigma), Total OXPHOS Rodent Antibody Cocktail (1:1000, ab110413, Abcam), NFAT1 (1:1000, 4389 S, Cell Signaling Technology), NFAT2 (1:1000, 8032 S, Cell Signaling Technology), α-Tubulin (1:5000, 3873 S, Cell Signaling Technology), phospho-CREB (Ser133, 1:1000, 9198 S, Cell Signaling Technology), and CREB (1:1000, 9104 S, Cell Signaling Technology). Membranes were washed with TBST and incubated for 1 hr at room temperature with the following secondary antibodies: IRDye 680RD goat anti-mouse (1:10,000 LI-COR) or IRDye 800RD donkey anti-rabbit (1:10,000 LI-COR). Membranes were imaged on an Odyssey CLx Imaging System (LI-COR) and analysis was performed in Image Studio Lite version 5.2 (LI-COR) and ImageJ.

## Quantitative RT-PCR

Total mRNA was isolated from A20 cells and primary B lymphocytes using an RNeasy Mini Kit (Qiagen) following the manufacturer's instructions. RNA concentrations were measured using a NanoDrop 2000 (Thermo Scientific) and 1 μg of DNAse I treated RNA was used with the High-Capacity cDNA Reverse Transcription Kit (Applied Biosystems). cDNA was amplified on a QuantStudio 3 Real-Time PCR System (Applied Biosystems) using PowerUp SYBR Green Master Mix (Applied Biosystems). PCR amplification was performed by initial activation for 2 min at 50°C, followed by a 95°C 2 min melt step. The initial melt steps were then followed by 40 cycles of 95°C for 15 s, 60°C for 15 s, and 72°C for 30 s. Data were analyzed with the instrument software v1.3.1 (Applied Biosystems) and analysis of each target was carried out using the comparative Ct method.

## Seahorse extracellular flux analysis

Oxygen consumption rates (OCR) and extracellular acidification rates (ECAR) were measured using an XFe24 Extracellular Flux Analyzer (Seahorse Bioscience). A20 cells ($0.8 \times 10^6$ per well) and primary B lymphocytes ($2 \times 10^6$ per well) were resuspended in XF DMEM pH 7.4 media supplemented with 1 mM pyruvate, 2 mM glutamine, and 10 mM glucose (Mito Stress Test) and plated in poly-L-lysine coated microchamber wells. For the Mito Stress Test, 1.5 μM oligomycin, 2 μM FCCP, and 0.5 μM antimycin/rotenone were utilized. Data were analyzed using the Agilent Seahorse Wave Software and normalized to total protein context per well using the Pierce Rapid Gold BCA Protein Assay Kit (Thermo Scientific).

## Transmission electron microscopy

Primary B cells (unstimulated or 24 hr IgM stimulated) were seeded onto poly-L-lysine coated cell culture dishes and fixed with 1% glutaraldehyde in 0.1 M sodium phosphate buffer, pH 7.3. After fixation, the cells were washed with 100 mM Tris (pH 7.2) and 160 mM sucrose for 30 mins. The cells were washed twice with phosphate buffer (150 mM NaCl, 5 mM KCl, 10 mM Na3PO4, pH 7.3) for 30 min, followed by treatment with 1% OsO4 in 140 mM Na3PO4 (pH 7.3) for 1 hr. The cells were washed twice with water and stained with saturated uranyl acetate for 1 hr, dehydrated in ethanol, and embedded in Epon (Electron Microscopy Sciences, Hatfield, PA). Roughly 60 nm sections were cut and stained with uranyl acetate and lead nitrate. The stained grids were analyzed using a Philips CM-12 electron microscope (FEI; Eindhoven, The Netherlands) and photographed with a Gatan Erlangshen ES1000W digital camera (Model 785, 4 k 3 2.7 k; Gatan, Pleasanton, CA).

## RNA sequencing and differential expression analysis

Total mRNA was isolated from primary B lymphocytes using an RNeasy Mini Kit (Qiagen) following the manufacturer's instructions. RNA concentrations were quantitated using a NanoDrop 2000 (Thermo Scientific) and library preparation was performed by Novogene. A total amount of 1 μg RNA per sample was used as input material for the RNA sample preparations. Sequencing libraries were generated using NEBNext Ultra TM RNA Library Prep Kit for Illumina (NEB, USA) following the manufacturer's recommendations, and index codes were added to attribute sequences to each sample. Briefly, mRNA was purified from total RNA using poly-T oligo-attached magnetic beads. Fragmentation was carried out using divalent cations under elevated temperature in NEBNext First-strand Synthesis Reaction Buffer (5 X). First-strand cDNA was synthesized using random hexamer primer and M-MuLV Reverse Transcriptase (RNase H-). Second-strand cDNA synthesis was subsequently performed using DNA Polymerase I and RNase H. Remaining overhangs were converted into blunt ends via exonuclease/

polymerase activities. After adenylation of 3' ends of DNA fragments, NEBNext Adaptor with hairpin loop structure was ligated to prepare for hybridization. To select cDNA fragments of preferentially 150~200 bp in length, the library fragments were purified with AMPure XP system (Beckman Coulter, Beverly, USA). Then 3 µl USER Enzyme (NEB, USA) was used with size-selected, adaptor-ligated cDNA at 37°C for 15 min followed by 5 min at 95°C before PCR. Then PCR was performed with Phusion High-Fidelity DNA polymerase, Universal PCR primers, and Index (X) Primer. Lastly, PCR products were purified (AMPure XP system) and library quality was assessed on the Agilent Bioanalyzer 2100 system. The clustering of the index-coded samples was performed on a cBot Cluster Generation System using PE Cluster Kit cBot-HS (Illumina) according to the manufacturer's instructions. After cluster generation, the library preparations were sequenced on an Illumina NovaSeq 6000 Platform (Illumina, San Diego, CA, USA) using a paired-end 150 run (2 × 150 bases).

## Differential expression analysis

Ensembl gene identifiers were converted to gene symbols, and identifiers with duplicated or missing gene symbols were removed from the analysis. Exploratory analyses were performed on the gene-level read count data, and lowly expressed genes were removed from the data set. The EDASeq R package (*Risso et al., 2011*) was used to create a SeqExpressionSet object based on the read counts, then upper quantile normalization was applied. The RUVSeq R package (*Risso et al., 2011*) was then applied using k=1 and a set of 14 mouse housekeeping genes identified by *Ho and Patrizi, 2021* to identify factors of unwanted variation that were included as a covariate in the differential expression analysis performed with edgeR (*McCarthy et al., 2012*; *Robinson et al., 2010*). Differentially expressed genes were chosen based on a false discovery rate threshold of q<0.05. R 4.0.5 was used for all analyses (https://www.R-project.org).

## Gene set enrichment analysis (GSEA)

Based on the edgeR output, GSEA software was applied to perform pathway analyses (*Mootha et al., 2003*; *Subramanian et al., 2005*). The edgeR output includes likelihood ratio (LR) test statistics, p-values, and false discovery rate q-values, as well as log fold change (logFC) values based on the expression values of each gene in the comparison groups of interest. Because the LR statistics are non-negative, their values alone cannot distinguish up- and down-regulated genes, as is required for a GSEA pre-ranked analysis. Thus, we computed a signed version of the LR statistic that was defined to be the product of LR statistic times and the sign of the logFC, thereby enabling us to rank the genes according to both the statistical significance and the direction of the expression differences. A GSEA pre-ranked analysis was performed using the ranked genes and gene sets available at the Molecular Signature Database (https://www.gsea-msigdb.org/gsea/msigdb/index.jsp). After the biomaRt R package (*Durinck et al., 2005*; *Durinck et al., 2009*) was used to convert mouse gene symbols to human gene symbols. Statistical significance was assessed at the q<0.05 level.

## Metabolite profiling

### Metabolite extraction

A metabolite extraction was carried out on each sample based on a previously described method (*Pacold et al., 2016*). An approximate cell count (5 × 10$^6$ cells for all samples) of the samples was used to scale the metabolite extraction to a ratio of 1 × 10$^6$ cells/1 mL extraction solution [80% LCMS grade methanol (aq) with 500 nM labeled amino acid internal standard (Cambridge Isotope Laboratories, Inc, Cat No. MSK-A2-1.2)]. The lysis was carried out in two steps as follows. First, 1 mL of freezing extraction solution was added to the tubes containing each pellet. That suspension was transferred to tubes along with zirconium disruption beads (0.5 mm, RPI) and homogenized for 5 min at 4 °C in a BeadBlaster with a 30 s on/30 s off pattern. The resulting lysate was then diluted to a fixed concentration of 1 × 10$^6$ cells/1 mL in a new tube with disruption beads in that same buffer and then re-homogenized in the same way as above. The homogenate was centrifuged at 21,000 × g for 3 min, and 450 µL of the supernatant volume was transferred to a 1.5 mL microfuge tube for speed vacuum concentration, no heating. The dry extracts were resolubilized in 50 µL of LCMS grade water, sonicated in a water bath for 2 min, centrifuged as above, and transferred to a glass insert for analysis.

## LC-MS/MS methodology

Samples were subjected to an LC-MS/MS analysis to detect and quantify known peaks. The LC column was a Millipore ZIC-pHILIC (2.1 × 150 mm, 5 µm) coupled to a Dionex Ultimate 3000 system and the column oven temperature was set to 25 °C for the gradient elution. A flow rate of 100 µL/min was used with the following buffers; (A) 10 mM ammonium carbonate in water, pH 9.0, and (B) neat acetonitrile. The gradient profile was as follows; 80–20% B (0–30 min), 20–80% B (30–31 min), 80–80% B (31–42 min). Injection volume was set to 2 µL for all analyses (42 min total run time per injection). MS analyses were carried out by coupling the LC system to a Thermo Q Exactive HF mass spectrometer operating in heated electrospray ionization mode (HESI). Method duration was 30 min with a polarity switching data-dependent Top five method for both positive and negative modes. Spray voltage for both positive and negative modes was 3.5 kV and the capillary temperature was set to 320 °C with a sheath gas rate of 35, aux gas of 10, and max spray current of 100 µA. The full MS scan for both polarities utilized 120,000 resolution with an AGC target of $3 \times 10^6$ and a maximum IT of 100ms, and the scan range was from 67 to 1000 $m/z$. Tandem MS spectra for both positive and negative modes used a resolution of 15,000, AGC target of $1 \times 10^5$, maximum IT of 50 ms, isolation window of 0.4 m/z, isolation offset of 0.1 m/z, fixed first mass of 50 m/z, and 3-way multiplexed normalized collision energies (nCE) of 10, 35, 80. The minimum AGC target was $1 \times 10^4$ with an intensity threshold of $2 \times 10^5$. All data were acquired in profile mode.

## Data analysis

Metabolomics data were processed with an in-house pipeline for statistical analyses and plots were generated using a variety of custom Python code and R libraries including: heatmap, MetaboAnalystR, and manhattanly. Peak height intensities were extracted based on the established accurate mass and retention time for each metabolite as adapted from the Whitehead Institute and verified with authentic standards and/or high-resolution MS/MS manually curated against the NIST14MS/MS and METLIN spectral libraries. The theoretical m/z of the metabolite molecular ion was used with a±10 ppm mass tolerance window, and a±0.2 min peak apex retention time tolerance within the expected elution window (1–2 min). To account for sample-to-sample variance in the estimated cell counts, a sum-normalization step was carried out on a per-column (sample) basis. Detected metabolite intensities in a given sample were summed, and a percentage intensity was calculated for each metabolite (custom Rscript available from NYU Metabolomics Laboratory at https://med.nyu.edu/research/scientific-cores-shared-resources/metabolomics-laboratory, contact the Laboratory Director Dr. Drew Jones at: drew.jones@nyulangone.org). The median mass accuracy vs the theoretical m/z for the library was −0.7 ppm (n=90 detected metabolites). Median retention time range (time between earliest and latest eluting sample for a given metabolite) was 0.23 min (30 min LCMS method). A signal-to-noise ratio (S/N) of 3 X was used compared to blank controls throughout the sequence to report detection, with a floor of 10,000 (arbitrary units). Labeled amino acid internal standards in each sample were used to assess instrument performance (median CV=5%).

## Influenza A virus (IAV) infection

*Orai1/Orai3*[fl/fl] Mb1-Cre[/+] and *Orai1/Orai3*[fl/fl] (control) mice were anesthetized with isoflurane and infected intranasally (i.n.) with $10^5$ TCID$_{50}$ of the laboratory strain A/HK/x31 (x31-IAV) of the influenza A virus subtype H3N2. Lungs were isolated for histology. Mediastinal lymph nodes and bone marrow were used to prepare single-cell suspensions followed by flow cytometric analysis. Serum was harvested for analyzing virus-specific antibody titers.

## Histology

Lungs were fixed with 4% paraformaldehyde in PBS, embedded in paraffin, and cut at 5 µm. Sample slides were stained with hematoxylin and eosin (H&E) using standard methods. Images were acquired using an SCN400 slide scanner (Leica), viewed with Omero Slidepath (Open Microscope Environment). The total alveolar volume fraction was determined using the following procedure: (i) regions representing empty space were acquired by setting a color threshold in ImageJ (**Schneider et al., 2012**); (ii) to calculate the total alveolar volume fraction, the area of empty space was divided by the total lung area using MATLAB (v2018a).

## ELISA

To measure X31-specific antibodies, half-area ELISA plates (Corning, Cat#3690) were coated with heat-inactivated x31-IAV overnight at 4 °C. ELISA plates were blocked with 20% FBS in PBS at 37 °C for 1 hr followed by washing three times with wash buffer (0.05% v/v Tween-20 in PBS). Plates were incubated with gradient dilutions of serum from control and *Orai1/Orai3*$^{fl/fl}$ Mb1-Cre$^{/+}$ mice for 1 hr at 37 °C followed by incubation with alkaline phosphatase (AP) goat anti-rabbit IgG secondary antibody (SouthernBiotech, Cat#1030–04), goat anti-rabbit IgM (SouthernBiotech, Cat#1021–04) at room temperature for 2 hr. After the addition of substrate solution, absorption was measured at 405 nm using a Flexstation three plate reader (Molecular Devices).

## Cell preparation of mediastinal lymph nodes and bone marrow

Single-cell suspensions of mediastinal lymph nodes (medLNs) and bone marrow were grounded and passed through 70 µm cell strainer (BD, 22-363-548). Cells were treated with ACK buffer for 3 min and then washed and spun at 800 g for 5 min, resuspended in RPMI medium plus 2% fetal bovine serum, and stained with antibodies as described below. Cells isolated from medLNs and bone marrow were counted with trypan blue, washed, and prepared for flow cytometry analysis in PBS containing 2% FBS and 2 mM EDTA. For surface staining, cells were stained with fluorescently labeled antibodies at 4°C for 15 min in the dark, followed by Live/Dead Blue (Invitrogen, Cat# L23105) staining following the manufacturer's instructions. Samples were acquired on an LSR Fortessa (BD Biosciences) and analyzed using FlowJo software (TreeStar, versions 9.3.2 and 10.5.3.). The list of antibodies used is as follows: B220- BV510 (RA3-6B2, Biolegend), CD138-PE (281–2, Biolegend), CD95-BV421 (Jo2, BD Bioscience), GL-7-Alexa-Fluor647 (GL7, Biolegend), CD4-APC-Cy7 (GK1.5, Biolegend), and CD38-PE-Cy7 (T10, Biolegend).

## Statistics

All statistical tests were conducted using GraphPad Prism 9 and data was presented as mean ± SEM. When comparing two groups the Student's t-test was used. If greater than two groups were compared, then a One-way analysis of variance was used. For all results with normally distributed data, parametric statistical tests were used. When data were not normally distributed, non-parametric statistical tests were used. Data normality was determined using Prism 9.0. Biological replicates were defined as primary cells isolated from an individual mouse from their respective genotype. Statistically significant differences between groups were identified within the figures where *, **, ***, and **** indicating p-values of <0.05, <0.01, <0.001, and <0.0001, respectively.

## Materials availability statement

All materials generated from this study including mice strains are available upon request from the lead PI (email: trebakm@pitt.edu).

# Acknowledgements

We thank Dr. Han Chen from The Pennsylvania State University College of Medicine EM facility for assistance with TEM imaging and Dr. Drew Jones and Mr. Leonard Ash from New York University Langone Health Metabolomics Core Resource Laboratory for assistance with metabolomics experiments. We are grateful to the Flow Cytometry and Informatic and Data Analysis Core facilities from The Pennsylvania State University College of Medicine. This work was supported by National Institutes of Health (MIH) grants R35-HL150778 (to M.T.) and in part by R35-HL161177 (to A.C.S), R01-AI162971 (to ZSMR) and R01-AI097302 and R01-AI130143 (to SF), and training grant F30-AI164803-01 (to AYT).

# Additional information

### Competing interests

---

Adam C Straub: owns stock options and is a consultant for Creegh Pharmaceuticals. Stefan Feske: is scientific co-founder of Calcimedica. Mohamed Trebak: Reviewing editor, *eLife*. The other authors declare that no competing interests exist.

## Funding

| Funder | Grant reference number | Author |
|---|---|---|
| National Heart, Lung, and Blood Institute | R35-HL150778 | Mohamed Trebak |
| National Institute of Allergy and Infectious Diseases | R01-AI162971 | Ziaur SM Rahman |
| National Institute of Allergy and Infectious Diseases | R01-AI097302 and R01-AI130143 | Stefan Feske |
| National Institute of Allergy and Infectious Diseases | F30-AI164803-01 | Anthony Y Tao |

The funders had no role in study design, data collection and interpretation, or the decision to submit the work for publication.

## Author contributions

Scott M Emrich, Conceptualization, Formal analysis, Validation, Investigation, Visualization, Methodology, Writing – original draft; Ryan E Yoast, Investigation, Methodology; Xuexin Zhang, Adam J Fike, Yin-Hu Wang, Kristen N Bricker, Anthony Y Tao, Ping Xin, Martin T Johnson, Trayambak Pathak, Investigation; Vonn Walter, Resources, Formal analysis; Adam C Straub, Resources; Stefan Feske, Ziaur SM Rahman, Resources, Writing – review and editing; Mohamed Trebak, Conceptualization, Resources, Supervision, Funding acquisition, Visualization, Project administration, Writing – review and editing

## Author ORCIDs

Scott M Emrich http://orcid.org/0000-0002-4804-7436
Kristen N Bricker http://orcid.org/0000-0001-8963-9780
Vonn Walter http://orcid.org/0000-0001-6114-6714
Ziaur SM Rahman http://orcid.org/0000-0001-8431-9681
Mohamed Trebak http://orcid.org/0000-0001-6759-864X

## Ethics

This study was performed in strict accordance with the recommendations in the Guide for the Care and Use of Laboratory Animals of the National Institutes of Health. All of the animals were handled according to approved institutional animal care and use committee (IACUC) protocols of Penn State University: Protocols #: 46290, 47477, and 47350.

## Decision letter and Author response

Decision letter https://doi.org/10.7554/eLife.84708.sa1
Author response https://doi.org/10.7554/eLife.84708.sa2

---

# Additional files

## Supplementary files

- MDAR checklist
- Source data 1. Genomic sequencing data of A20 Orai CRISPR clones.
- Source data 2. Raw RNA sequencing read count data from primary Orai KO B cells.
- Source data 3. RNA sequencing analysis of primary Orai KO B cells.
- Source data 4. GSEA analysis of RNA sequencing data from primary Orai KO B cells.

## Data availability

All antibodies, cell lines, chemicals, mice strains, and sequences of primers and gRNA used in the study are listed in the key resources table. Source data files have been provided.

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

# Appendix 1

## Appendix 1—key resources table

| Reagent type (species) or resource | Designation | Source or reference | Identifiers | Additional information |
|---|---|---|---|---|
| Antibody | Total OXPHOS Rodent WB Antibody Cocktail (Mouse monoclonal) | Abcam | Cat# ab110413, RRID:AB_2629281 | (1:1000) |
| Antibody | anti-GAPDH (Mouse monoclonal) | Millipore Sigma | Cat# MAB374, RRID:AB_2107445 | (1:5000) |
| Antibody | MCU (D2Z3B) (Rabbit monoclonal) | Cell Signaling Technologies | Cat# 14997, RRID:AB_2721812 | (1:2000) |
| Antibody | α-Tubulin (DM1A) (Mouse monoclonal) | Cell Signaling Technologies | Cat# 3873, RRID:AB_1904178 | (1:5000) |
| Antibody | NFAT1 Antibody (Rabbit polyclonal) | Cell Signaling Technologies | Cat# 4389, RRID:AB_1950418 | (1:1000) |
| Antibody | NFAT1 (D43B1) XP (Rabbit monoclonal) (Alexa Fluor 488 Conjugate) | Cell Signaling Technologies | Cat# 14324, RRID:AB_2798450 | (1:50) |
| Antibody | NFAT2 (D15F1) (Rabbit monoclonal) | Cell Signaling Technologies | Cat# 8032, RRID:AB_10829466 | (1:1000) |
| Antibody | Phospho-CREB (Ser133) (87G3) (Rabbit monoclonal) | Cell Signaling Technologies | Cat# 9198, RRID:AB_2561044 | (1:1000) |
| Antibody | CREB (86B10) (Mouse monoclonal) | Cell Signaling Technologies | Cat# 9104, RRID:AB_490881 | (1:1000) |
| Antibody | IRDye 680RD Goat anti-Mouse IgG antibody | LI-COR Biosciences | Cat# 925–68070, RRID:AB_2651128 | (1:10,000) |
| Antibody | IRDye 800CW Donkey anti-Rabbit IgG antibody | LI-COR Biosciences | Cat# 926–32213, RRID:AB_621848 | (1:5000) |
| Antibody | anti-Orai1 (Rabbit polyclonal) | Feske Lab | | (1:200) |
| Antibody | Goat anti-Rabbit IgG (H+L) Cross-Adsorbed Secondary Antibody, Alexa Fluor 647 | Thermo Fisher Scientific | Cat# A-21244, RRID:AB_2535812 | (1:1000) |
| Antibody | AffiniPure F(ab')$_2$ Fragment Goat Anti-Mouse IgM, μ chain specific (Goat polyclonal) | Jackson ImmunoResearch | Cat# 115-006-020 | (20 ug/mL) |
| Antibody | Rabbit anti-Mouse IgG (H&L) - Affinity Pure | Tonbo Biosciences | Cat# 70–8076 M002 | (10 ug/mL) |
| Antibody | InVivoMAb anti-mouse CD40 FGK4.5/FGK45 (Rat monoclonal) | Bio X Cell | Cat# BE0016-2 | (10 ug/mL) |
| Antibody | Brilliant Violet 605 anti-mouse/human CD45R/B220 Antibody (Rat monoclonal) | Biolegend | Cat# 103243 | (1:200) |
| Antibody | PE/Cyanine5 anti-mouse CD86 Antibody (Rat monoclonal) | Biolegend | Cat# 105015 | (1:100) |
| Antibody | PE/Cyanine7 anti-mouse I-A/I-E Antibody (Rat monoclonal) | Biolegend | Cat# 107629 | (1:800) |
| Antibody | CD3e Monoclonal Antibody (145–2 C11), PE (Hamster monoclonal) | Thermo Fisher Scientific | Cat# 12-0031-82 | (1:200) |

*Appendix 1 Continued on next page*

*Appendix 1 Continued*

| Reagent type (species) or resource | Designation | Source or reference | Identifiers | Additional information |
|---|---|---|---|---|
| Antibody | V500 Rat anti-Mouse CD8a (Rat monoclonal) | BD Biosciences | Cat# 560776 | (1:200) |
| Antibody | BB700 Rat Anti-Mouse CD4 (Rat monoclonal) | BD Biosciences | Cat# 566408 | (1:200) |
| Antibody | APC Rat Anti-Mouse CD24 (Rat monoclonal) | BD Biosciences | Cat# 562349 | (1:200) |
| Antibody | FITC Rat Anti-Mouse CD23 (Rat monoclonal) | BD Biosciences | Cat# 553138 | (1:200) |
| Antibody | IgM Monoclonal Antibody (eB121-15F9) (Rat monoclonal) | Thermo Fisher Scientific | Cat# 12-5890-82 | (1:200) |
| Antibody | CD93 (AA4.1) Monoclonal Antibody (AA4.1), APC (Rat monoclonal) | Thermo Fisher Scientific | Cat# 17-5892-82 | (1:200) |
| Antibody | PE/Cyanine5 Streptavidin | Biolegend | Cat# 405205 | (1:200) |
| Antibody | Pacific Blue anti-mouse/human CD45R/B220 Antibody (Rat monoclonal) | Biolegend | Cat# 103230 | (1:200) |
| Cell line (*Mus musculus*) | A20 | ATCC | ATCC Cat# TIB-208, RRID:CVCL_1940 | |
| Chemical compound, drug | FK506 | STEMCELL Technologies | Cat# 74152 | |
| Chemical compound, drug | CRAC Channel Inhibitor IV, GSK-7975A | Sigma Aldrich | Cat# 5343510001 | |
| Chemical compound, drug | 2-APB | Tocris Bioscience | Cat# 1224 | |
| Chemical compound, drug | Thapsigargin | Thermo Fisher Scientific | Cat# T7458 | |
| Chemical compound, drug | 5 (6)-CFDA, SE; CFSE (5-(and-6)-Carboxyfluorescein Diacetate, Succinimidyl Ester), mixed isomers | Thermo Fisher Scientific | Cat# C1157 | |
| Chemical compound, drug | Gadolinium(III) Chloride | ACROS Organics | Cat# AC383560050 | |
| Commercial assay or kit | cDNA Reverse Transcription Kit | Applied Biosystems | Cat# 4368814 | |
| Commercial assay or kit | Seahorse XF Cell Mito Stress Test Kit | Agilent Technologies | Cat# 103015–100 | |
| Commercial assay or kit | Pierce Rapid Gold BCA Protein Assay Kit | Thermo Fisher Scientific | Cat# A53225 | |
| Commercial assay or kit | Tetramethylrhodamine, Ethyl Ester, Perchlorate (TMRE) | Thermo Fisher Scientific | Cat# T669 | |
| Commercial assay or kit | MitoTracker Green FM - Special Packaging | Thermo Fisher Scientific | Cat# M7514 | |
| Commercial assay or kit | Cell Line Nucleofector Kit V | Lonza | Cat# VCA-1003 | |
| Commercial assay or kit | StrataClone Blunt PCR Cloning Kit | Agilent Technologies | Cat# 240207 | |
| Commercial assay or kit | EasySep Mouse B Cell Isolation Kit | STEMCELL Technologies | Cat# 19854 | |
| Commercial assay or kit | eBioscience Foxp3 / Transcription Factor Staining Buffer Set | Thermo Fisher Scientific | Cat# 00-5523-00 | |

*Appendix 1 Continued on next page*

*Appendix 1 Continued*

| Reagent type (species) or resource | Designation | Source or reference | Identifiers | Additional information |
|---|---|---|---|---|
| Commercial assay or kit | RNeasy Mini Kit | Qiagen | Cat# 74106 | |
| Commercial assay or kit | LIVE/DEAD Fixable Near-IR Dead Cell Stain Kit | Thermo Fisher Scientific | Cat# L34975 | |
| Recombinant DNA reagent | pSpCas9(BB)–2A-GFP (PX458) | Addgene | Cat# 48138 | |
| Recombinant DNA reagent | pU6-(BbsI)_CBh-Cas9-T2A-mCherry | Addgene | Cat# 64324 | |
| Genetic Reagent (*Mus musculus*) | B6.C(Cg)-Cd79atm1(cre)Reth/EhobJ | Jackson Laboratory | Strain #:020505 RRID:IMSR_JAX:020505 | Mb1-Cre on C57BL/6 |
| Genetic Reagent (*Mus musculus*) | Orai1fl/fl | *Ahuja et al., 2017* PMID:28273482 | | |
| Genetic Reagent (*Mus musculus*) | Orai3fl/fl | *Gammons et al., 2021* PMID:33849280 | | |
| Genetic Reagent (*Mus musculus*) | Orai1fl/fl Mb1cre/+ | This paper | | Orai1fl/fl mice crossed with Mb1-Cre on C57BL/6 |
| Genetic Reagent (*Mus musculus*) | Orai3fl/fl Mb1cre/+ | This paper | | Orai3fl/fl mice crossed with Mb1-Cre on C57BL/6 |
| Genetic Reagent (*Mus musculus*) | Orai1/Orai3fl/fl Mb1cre/+ | This paper | | Orai1/Orai3fl/fl mice crossed with Mb1-Cre on C57BL/6 |
| Chemical compound, drug | Fura-2, AM, cell permeant | Thermo Fisher Scientific | Cat# F1221 | |
| Other | SYBR Select Master Mix | Thermo Fisher Scientific | Cat# 4472920 | Master mix for qRT-PCR |
| Other | Lipopolysaccharides from *Escherichia coli* O111:B4 | Sigma Aldrich | Cat# L2630-10MG | Stimulation of primary B lymphocytes |
| Other | Intercept (TBS) Blocking Buffer | LI-COR Biosciences | Cat# 927–60001 | Western blot blocking buffer |
| Other | Halt Protease and Phosphatase Inhibitor | Thermo Fisher Scientific | Cat# PI78443 | Western blot lysis buffer component |
| Other | RIPA Buffer | Sigma Aldrich | Cat# R0278-50ML | Western blot lysis buffer component |
| Other | DAPI | Sigma Aldrich | Cat# D9542 | Nuclear localization staining for ImageStream |
| Other | EasySep Buffer | STEMCELL Technologies | Cat# 20144 | B cell isolation kit buffer |
| Other | NuPAGE 4 to 12%, Bis-Tris, 1.0–1.5 mm, Mini Protein Gels | Thermo Fisher Scientific | Cat# NP0321BOX | Western blot denaturing gels |
| Other | Poly-L-lysine solution | Sigma Aldrich | Cat# P4832-50ML | Attachment of suspension cells to coverslips |
| Sequence-based Reagent | mOrai1n | This paper | gRNA primers for mouse Orai1 CRISPR knockout | GCCTTCGGATCCGGTGCGTC |
| Sequence-based Reagent | mOrai1c | This paper | gRNA primers for mouse Orai1 CRISPR knockout | CACAGGCCGTCCTCCGGACT |
| Sequence-based Reagent | mOrai3n | This paper | gRNA primers for mouse Orai3 CRISPR knockout | GCGTCCGTAACTGTTCCCGC |
| Sequence-based Reagent | mOrai3c | This paper | gRNA primers for mouse Orai3 CRISPR knockout | GAAGGAGGTCTGTCGATCCC |
| Sequence-based Reagent | Mb1 Cre Common Primer | This paper | PCR primers for genotyping Mb1 Cre | ACT GAG GCA GGA GGA TTG G |
| Sequence-based Reagent | Wild Type Forward Primer | This paper | PCR primers for genotyping Mb1 Cre | CTC TTT ACC TTC CAA GCA CTG A |

*Appendix 1 Continued on next page*

*Appendix 1 Continued*

| Reagent type (species) or resource | Designation | Source or reference | Identifiers | Additional information |
|---|---|---|---|---|
| Sequence-based Reagent | Mutant Forward Primer | This paper | PCR primers for genotyping Mb1 Cre | CAT TTT CGA GGG AGC TTC A |
| Sequence-based Reagent | Orai1 fl/fl Forward | This paper | PCR primers for genotyping Orai1 flox | ACC CAT GTG GTG GAA AGA AA |
| Sequence-based Reagent | Orai1 fl/fl Reverse | This paper | PCR primers for genotyping Orai1 flox | TGC AGG CAC TAA AGA CGA TG |
| Sequence-based Reagent | Orai3 fl/fl Forward | This paper | PCR primers for genotyping Orai3 flox | GAG CTG GGA TTA AAG GTG TAT GCC |
| Sequence-based Reagent | Orai3 fl/fl Reverse | This paper | PCR primers for genotyping Orai3 flox | TGA CTT CAC CTC AGT CTC AAA GGG G |
| Sequence-based Reagent | mOrai1 F | This paper | RT-PCR primers | CCA AGC TCA AAG CTT CCA GC |
| Sequence-based Reagent | mOrai1 R | This paper | RT-PCR primers | GCA CTA AAG ACG ATG AGC AAC C |
| Sequence-based Reagent | mOrai2 F | This paper | RT-PCR primers | GCAGCTACCTGGAACTCGTC |
| Sequence-based Reagent | mOrai2 R | This paper | RT-PCR primers | GTTGTGGATGTTGCTCACCG |
| Sequence-based Reagent | mOrai3 F | This paper | RT-PCR primers | ACC AAC GAC TGC ACA GAT AC |
| Sequence-based Reagent | mOrai3 R | This paper | RT-PCR primers | CCA ATG GGC ACA AAC TTG AC |
| Sequence-based Reagent | mGAPDH F | This paper | RT-PCR primers | GTG GCA AAG TGG AGA TTG TTG |
| Sequence-based Reagent | mGAPDH R | This paper | RT-PCR primers | CGT TGA ATT TGC CGT GAG TG |
| Software, algorithm | Image J | https://imagej.net/ | RRID:SCR_003070 | |
| Software, algorithm | Graphpad Prism | http://www.graphpad.com/ | RRID:SCR_002798 | |
| Software, algorithm | Leica Application Suite X | https://www.leica-microsystems.com/ | RRID:SCR_016555 | |
| Software, algorithm | Image Studio Lite | https://www.licor.com/bio/image-studio-lite/download | RRID:SCR_013715 | |
| Software, algorithm | FlowJo 9.9.6 | https://www.flowjo.com/solutions/flowjo | RRID:SCR_008520 | |

