## [Editor Report]

This study shows that Orai1 and Orai3 are important elements of the calcium signaling machinery in mouse B cells and regulate downstream responses of proliferation, energetics, and metabolic reprogramming that occur after stimulation of the B cell antigen receptor. This work provides new information of interest to a broad range of readers in the fields of immunology, ion channels, and calcium signaling.

---

## [Decision Letter]

**Decision letter after peer review:**

[Editors’ note: the authors submitted for reconsideration following the decision after peer review. What follows is the decision letter after the first round of review.]

Thank you for submitting the paper "Orai3 and Orai1 are essential for CRAC channel function and metabolic reprogramming in B cells" for consideration by *eLife*. Your article has been reviewed by 3 peer reviewers, one of whom is a member of our Board of Reviewing Editors, and the evaluation has been overseen by a Senior Editor. The following individual involved in the review of your submission has agreed to reveal their identity: Yousang Gwack (Reviewer #2).

Comments to the Authors:

While all three reviewers expressed interest in the work including its conceptual framework, they identified major weaknesses and essential revisions that will be needed to support the main conclusions. In particular, the reviewers raised strong concerns regarding the effects of the Orai1 KO in native B cells and about the data in Figure 3, showing only a 25% reduction in Orai1 protein at the cell surface. Additional concerns dealing with the rather limited range of B cell functions examined reduced enthusiasm for the study. These gaps will likely require more time than *eLife* policy allows for inviting revisions (~two months). Thus, we are sorry to say that we cannot consider this manuscript for publication by *eLife* at this time. However, given the enthusiasm of the reviewers, we would be willing to consider a revised manuscript that addresses these concerns as a new submission. To assist you with the revision, we list below the most significant concerns.

Essential revisions:

1) Verify the true extent of cell surface Orai1 expression in Orai1 KO B cells: The data in Figure 3A, B clearly show that expression in the MB1-Cre cells is ~75% of the wildtype controls implying that most Orai1 protein is in fact still retained at the cell surface. This makes it very difficult to draw any realistic conclusions about the role of Orai1, and by extension, Orai3 using the Orai1/Orai3 DKOs. The Orai1 antibody should be validated, and if it is not specific, a better one should be used with appropriate validation.

2) Related to the above point, the strong residual NFAT activation in Orai1 KO cells (surprising!), and lack of effect on B cell proliferation (which contradicts Gwack et al., 2008) seems consistent with the persistence of functional Orai1 protein at the cell surface.

3) The paper seems to imply that CRAC channels in B cells are heteromultimers of Orai1/Orai3 proteins. Without electrophysiological analysis, this claim realistically cannot be made and the study needs to demonstrate this point more definitively than simply looking at increases or decreases in cellular calcium concentrations.

4) The difference in phenotypes between Orai1 and Orai1/3 DKO B cells needs to be presented side by side to allow direct comparison. This should be done for Ca oscillations, NFAT activation and kinetics, proliferation, OCR, and respiration. See reviewer 2's comments on this matter. Experiments pertaining to Orai1 KO B cells in Figures 3 and 4 should also include those from Orai1/3 DKO B cells.

5) The functional significance of the study in its current form is very limited. Additional functional analysis to include cytokine production or antibody production is needed to increase the significance of the study.

6) B cell development. Check whether B cell development is affected in Orai1/3 Mb1cre KO mice. As noted by reviewer 2, Orai3 KO alone is unlikely to affect B cell development, however, does co-deletion of Orai1 and Orai3 affect B cell development or mature B cell populations in the spleen and peritoneum (Figure 4)?

7) What is the reason for reduced respiration in the Orai1/3 KO B cells which seems more than that observed in Orai1 KO cells? Use RNA-seq to examine this question.

*Reviewer #1 (Recommendations for the authors):*

In this study, Emrich and coworkers examine the consequences of ablating Orai3 and Orai1 on store-operated calcium entry and B cell functions. It is known that SOCE is important for T cell function including T cell activation, proliferation, and cytokine synthesis. But less is known about Orai channel contributions to B cell function. Using conditional deletion of Orai1 and Orai3 in B cells, the authors examine several functional endpoints including B cell proliferation, development, and metabolism. These endpoints are diminished by Orai1 deletion. Deletion of Orai3 evokes no phenotype for most endpoints. The results indicating a lack of developmental effects on B cells with Orai1 deletion are not surprising and replicate previous work on these topics both in B cells and in T cells (relevant papers from Feske lab). Orai1 deletion also prevents NFAT activation, a well-known consequence of Orai1 inhibition in many cells. However, in several cases, the double Orai1/Orai3 knockout shows phenotypes that are slightly larger than those seen in Orai1-alone deficient B cells, suggesting that Orai3 plays a modest role in residual calcium entry and cellular functions.

While the experiments are carefully conducted, I think the paper exaggerates the role of Orai3 in B cell function. Many statements such as "Orai3 and Orai1 coordinate for efficient B cell proliferation" are not meaningful and vague and overstate the role of Orai3. In fact, Orai1 or Orai3 deletion has no effect on B cell proliferation. The lack of effect of Orai1 deletion is somewhat surprising and contradicts earlier findings (Gwack et al., 2008) indicating that Orai1 is critical for B cell proliferation and this contradiction is not explained. The double Orai3/Orai1 KO evokes a modest reduction of B cell proliferation but it is unclear whether this is a specific effect or downstream of more general suppression of metabolism since it is well known that rapidly proliferating cells require a high level of metabolism and glycolysis which is suppressed in the double KO. Perhaps the most interesting part of the paper is that the double KO shows a greater suppression of oxygen consumption and maximal respiration than Orai1 KO cells alone. But no effect is seen with Orai3 KO ablation and none on basal respiration even though deletion of Orai1 alone evokes the strongest effects. Thus, statements indicating that Orai1/Orai3 channels are crucial for metabolic reprogramming do not easily follow from the data shown, which seem to indicate very minor contributions of Orai3 for these endpoints.

Overall, the role of Orai3 is greatly over-emphasized and is not supported by the results. One could easily conclude based on the results that Orai3 is largely dispensable/redundant for the B cell responses that were studied here. But it is also worth noting that only a small subset of physiologically relevant B cell responses was assessed and notably cytokine and antibody production were not studied at all. These issues reduce the significance of the study.

Additional comments:

1) What is the effect of deleting Orai3 on NFAT1 activation? A lack of effect on cell proliferation in vitro is not surprising and is reminiscent of the lack of effects on proliferation in T cells with non-function Orai1 channels (McCarl, Feske et al., JI 2010).

2) Are Orai channels in a complex with Orai1 and Orai3 subunits in the same channel or are they expressed separately to form distinct channels in B cells? 2-APB appears to inhibit the residual SOCE in Orai3 deficient cells but not in Orai1 deficient cells. What are the implications of this result? The 2-APB experiment is uninterpreted in terms of the molecular composition of the SOCE channels.

3) In order to really assess the contributions of Orai3 to B cell physiological responses, it is important to directly compare the outcomes side-by-side for the different endpoints. Thus, the metrics for NFAT activation, proliferation, OCR, respiration, etc. should be directly compared as in Figure 2.

4) Figure 6: the fact that Orai1 deletion in primary B cells causes a substantial decrease in NFAT1 activation but has no effects on Ca oscillations (Figure 2/3) implies that the Ca oscillations have no direct relevance for NFAT activation in B cells. This result seems to contradict previous literature on this issue and should be discussed. Is NFAT1 activation more dependent on the plateau Ca responses instead of oscillations?

5) There is a large discrepancy between mRNA and protein levels in Figure 3. What is the true efficiency of deletion?

6) Two key functions of B cells are antibody production and secretion of cytokines that fosters T cell activation. It is hard to assess the importance of Orai channels for B cell physiology without at least some measurement of these functions. As such, the narrow focus on the study on B cell proliferation and metabolism falls short of truly understanding the role of SOCE in B cell physiology.

*Reviewer #2 (Recommendations for the authors):*

This manuscript examines the roles of Orai channels, the predominant ca^2+^ channels in T cells, in the development and activation of B cells. This manuscript evaluates the detailed role of the members of the Orai family, Orai1, Orai2, and Orai3 channels, in ca^2+^ signaling, proliferation, and metabolism in B cells. The authors find a positive contribution of Orai1 and Orai3 in B cell activation and metabolic functions using knockout mouse studies. This study has significant novelty and impact.

Overall, the manuscript has novelty in terms of examining the role of the Orai channels in B lymphocytes, which is currently understudied. While the role of Orai1 in B cell activation and proliferation has been described before, that of Orai3 is not studied. The major weakness of the current work is that the authors need to emphasize experiments showing comparative studies between Orai1 KO, Orai3 KO, and Orai1/3 DKO B cells. Ideally, these results must be compared to Orai1/2 DKO B cells. To strengthen their conclusions, I have the following recommendations:

1. To emphasize the role of Orai3 in B cells, the authors should check whether B cell development is affected in Orai1/3 Mb1cre KO mice. Orai3 KO alone is unlikely to affect B cell development, however, does co-deletion of Orai1 and Orai3 affect B cell development or mature B cell populations in the spleen and peritoneum (Figure 4)?

2. The difference in phenotypes between Orai1 and Orai1/3 DKO B cells (which is the novelty of this paper) needs to be presented side by side. Experiments pertaining to Orai1 KO B cells in Figures 3 and 4 should also include those from Orai1/3 DKO B cells. For example, does the loss of both Orai1 and Orai3 affect ca^2+^ oscillations in DKO primary B cells (Figure 4), similar to their observation with cell line in Figure 3? The NFAT immunoblots from Figure 6 supplement should move to the main Figure 6. And it will be useful to compare NFAT translocation kinetics shown in Figure 6A among Orai1 KO and Orai1/3 DKO B cells.

3. What is the physiological output of reduced proliferation in Orai1/3 DKO B cells seen in Figure 7? As shown in the STIM1/2 DKO B cell paper, the authors can check IL-10 production by all 4 genotypes by ELISA to get a quantitative comparison between Orai1 and Orai1/3 DKO cells. The authors should also show a heat map of the B cell cytokine profile (after IgM stimulation) from the RNA-seq results (Figure 7D and E).

4. In Figure 8, are the mitochondria numbers affected in Orai1/3 DKO (Figure 8D)? What is the cause of reduced respiration (more reduction than that observed in Orai1 KO cells) in the Orai1/3 KO B cells? Using RNA-seq data can the authors uncover the mechanisms linking loss of mitochondrial metabolism to reduced SOCE via Orai1/3?

5. Ideally, the metabolomic analysis in Figure 9 should be compared between Orai1 KO and Orai1/3 DKO B cells.

*Reviewer #3 (Recommendations for the authors):*

Orai channels mediate calcium influx which is essential for immune cell activation. While the role of Orai1 and Orai2 are well established in T lymphocytes, the particular Orai isoforms that function in B cells are not well established. This study uses CRISPR-mediated gene deletion in the A20 B-cell line and primary B cells in mice to examine the functional roles of Orai1, 2, and 3. The authors show evidence that Orai1 and Orai3 are critical components of store-operated Ca channels in B cells and help drive downstream events in B cell activation. New findings include:

Orai1 and Orai3 are upregulated and Orai2 is downregulated during B cell activation by different stimuli in vitro, including anti-IgM, anti-IgM + anti-CD40, and LPS. The significance of this result is undetermined, as the impact of Orai regulation on B cell responses and Ca signal generation was not studied.

Based on the effects of CRISPR-mediated deletion, Orai1 and Orai3 both contribute to the magnitude of store-operated Ca entry evoked by full ER Ca depletion in A20 B cells and primary B cells. The combination of Orai1 and Orai3 was critical for the generation of sustained Ca oscillations after BCR stimulation, while deletion of one or the other channel had little effect. However, surface staining of the Orai1 knockout B cells with polyclonal anti-Orai1 antibodies indicated that despite complete suppression of mRNA, surface protein levels were ~75% of wild-type cells. The persistence of Orai1 protein weakens many of the conclusions about the functions of Orai1 based on the knockout cells, e.g. that Orai1 is dispensable for B cell differentiation in vivo, proliferation, and survival after stimulation in vitro, and has only a modest effect on NFAT activation in vitro.

The double knockout of Orai1 and Orai3 has greater effects on proliferation and survival than Orai1 knockout alone, and greatly inhibits metabolic reprogramming of B cells after in vitro stimulation. However, due to the persistence of Orai1 protein the magnitude of these effects may underestimate the true contribution of Orai1.

1. Orai1 and Orai3 are shown to be upregulated, while Orai2 is downregulated after stimulation of B cells in vitro (Figure 1), and the effects are dependent on the type of stimulating agent. These same stimulation conditions are used later in experiments on NFAT activation, proliferation and survival, and metabolic reprogramming. However, an essential link between upregulation and functional effects is how are the Ca signals affected? SOCE (with TG) and oscillations (with anti-IgG) should be measured.

2. A serious problem that affects many of the conclusions of the paper is that CRISPR-mediated deletion of Orai1 in mouse B cells causes only a ~25% decrease in Orai1 protein as detected by the polyclonal antibody. The characteristics of this antibody are not described, and there are questions about its specificity. Has it been validated against cells known to lack Orai1? Was it directly labeled or was a secondary Ab used for FACS? Surface labeling should also be assessed in the A20 Orai1 KO lines.

Assuming the antibody is specific, the persistence of 75% of the surface Orai1 undermines or changes the interpretations of many of the experiments in the paper:

a) Cannot say that Orai1 is dispensable for ca^2+^ oscillations (line 206-7), B cell development (Figure 4), B cell proliferation and survival (Figure 7), or increased mitochondrial mass in activated B cells (Figure 8).

b) The enhancement of SOCE in Orai2 KO cells does not actually indicate that residual SOCE in Orai1/Orai3 DKO is due to Orai2 (Figure 5; lines 251-2). An alternative explanation is that the residual SOCE is due to Orai1, and the very low SOCE seen is from Orai2 suppressing the already reduced function of Orai1.

c) Cannot conclude that Orai3 is an "essential" component of the native SOCE pathway in B cells if Orai1 was not totally eliminated in the Orai1 KO (and therefore had a minimal effect) (line 413).

It is also worth noting that because Orai1 is the only isoform that supports NFAT activation, the persistence of NFAT translocation in the Orai1 KO cells is another sign that Orai1 protein is still present and functional at significant levels (Figure 6). Moreover, the persistence of residual Orai1 may explain why the phenotype of the Orai1/Orai3 DKO is not as severe as the STIM1/STIM2 DKO studied by other groups (line 446-8).

3. At several points in the paper, the authors imply that the CRAC channel in B cells is a heteromultimer of Orai1 and Orai3 (e.g., lines 259-60). Orai3 contributes to SOCE, but no arguments are made to rule out the alternative that Orai1 and Orai3 make separate channels. Pharmacology, kinetic profiling (CDI), or permeation properties of CRAC currents in whole-cell recordings may be an approach to distinguish between these possibilities. Otherwise, it remains an open question.

[Editors’ note: further revisions were suggested prior to acceptance, as described below.]

Thank you for resubmitting your work entitled "Orai3 and Orai1 are essential for CRAC channel function and metabolic reprogramming in B cells" for further consideration by *eLife*. Your revised article has been evaluated by Kenton Swartz (Senior Editor) and a Reviewing Editor.

The manuscript has been improved but there are some remaining key issues that need to be addressed in the reviewers' comments. The essential points are:

1) The patch-clamp data are not credible and the currents do not show the classic inward rectification that is the hallmark of CRAC. Given that no other features are shown, these data should be removed as they are not critical for the study.

2) There continue to be claims indicating that Orai3 and Orai1 form heteromeric channels. There is no evidence in the study to support this conclusion and the results can be fully explained considering that Orai1 and Orai3 form functional but separate channels. Since the conclusions don't depend on this finding, these claims should be removed.

3) Reviewers had a problem with language that make claims about Orai3 being essential (or "critical") for CRAC channel function, or other things such as proliferation (in the Title, Abstract, Discussion, and Results headers). These appear to be left over from the previous version and were not revised. Since the SOCE measurements clearly show that Orai3 is not critical or essential for CRAC channel activity nor "essential" for function including proliferation the language should be changed to "involved" in or something analogous.

4) Discuss why the Orai3 cKO cells have a respiration defect as SOCE measurements do not show any change in SOCE (Figure 5).

5) Kudos to the authors for evaluating several in vivo metrics of B cells lacking both Orai1 and Orai3 for B cell cytokine production, IgG antibodies, receptors, and overall immunity for infection (Figure 11). The lack of any effects is extremely surprising. This raises a key concern regarding the in vivo significance of Orai1 and Orai3 for B cell function. In particular, what do all the in vitro data mean if knocking out Orai1 and Orai3 has no effect on B cell development or function? This should be tackled in the Discussion.

*Reviewer #1 (Recommendations for the authors):*

This revised manuscript by Emrich et al. has addressed most of the concerns brought up in the previous round of reviews. The authors have done a nice job and appropriately revised the text and discussion, but a couple of key points remain. Please see the detailed comments below.

– The new patch-clamp data in Figure 6 are confusing; the currents do not show the classic inward rectification of CRAC currents, and the ramps shown in A-C in fact seem to have the wrong curvature. Since these data do not support the idea of heteromeric Orai1/Orai3 channels and are of unclear origin, I suggest removing these data as they are not needed to support the main conclusions.

– Supplementary Figure 10: An interesting result here is that upregulation of most polar metabolites co-stimulated with anti-IgM + anti-CD40 is similar to that of control cells stimulated with IgM alone and much larger than the levels seen in WT cells exposed to the GSK compound. This difference means that the effects of the GSK compound are not equivalent to that of the compound. This should be discussed in light of the fact that the DKO eliminates SOCE almost entirely.

– On page 6, the heading "Both Orai3 and Orai1 are essential components of SOCE in B cells". The word essential here is incorrect as the data shows that removing Orai3 has no effects on SOCE (Figure 5B, C, and Figure 6E). Please remove the word "essential" in this heading.

Similarly, in the abstract, the sentence "We show that Orai3 and Orai1 are essential components of native CRAC channels in B cells …" should be modified to remove the word essential for the reason above.

– Finally, the title remains unchanged from the previous version. The title should be amended since the data clearly shows that Orai3 is not essential for CRAC channel function. Please revise.

*Reviewer #3 (Recommendations for the authors):*

Comments on authors' rebuttal:

1. The effects of Orai protein up- or down-regulation in response to stimulation in vitro need to be related to their effects on NFAT activation, proliferation and survival, and metabolic reprogramming. The authors compare SOCE in naïve vs. stimulated cells after full store depletion (TG), but a comparison of Ca oscillations evoked by IgM stimulation is the relevant one and should be done since these are presumably the kind of Ca signals that are involved in regulating downstream behavior under more physiological conditions.

2. The updated data on Orai1 expression in the Orai1fl/fl Mb1cre/+ cells makes more sense than the original dataset. It is still not totally clear why there is only a 70% reduction in Orai1 staining. The argument that the antibody for Orai1 crossreacts with Orai2 and Orai3 is not documented here, but the fact that qPCR data show an almost complete absence of Orai1 mRNA is sufficient here to conclude that Orai1 protein is effectively suppressed.

3. The new patch clamp data are problematic (see comment #2 below) and cannot be used to support the idea that CRAC channels are heteromers of Orai1 and Orai3 as the authors state. One cannot have it both ways – to conclude that there are heteromers, but concede that the evidence does not prove this.

Comments on the revised manuscript:

1. A major conclusion of the paper is that Orai3 and Orai1 are "essential" or "crucial" components of native CRAC channels in B cells. This is based on findings that deletion of either protein alone partially reduces SOCE, while the double KO almost completely eliminates it. The authors argue at several points in the paper that the CRAC channel may be a heteromultimer of Orai1 and Orai3, but the data fall short of demonstrating the molecular composition of CRAC channels. In regard to Figure 5 (and Figure S2), "The slower rate of SOCE inhibition by 2-APB in wildtype B cells compared to Orai1 knockout B cells suggests that the native CRAC channel in B cells is a heteromer of Orai1 and Orai3." I disagree – the data could just as well be explained by separate populations of channels containing Orai1 or Orai3. In that case, 2-APB would inhibit Orai1 but enhance Ca influx through Orai3 (Yamashita et al., JBC 2011); thus, knocking out Orai3 would be expected to increase the inhibitory effect of 2-APB, as seen in Figure 5B. Likewise, the fact that double KO is required to inhibit oscillations could be explained simply by oscillations not being sensitive to a partial loss of SOCE, and only cease when SOCE is largely eliminated. The subunit composition of the channel could be complex, with multiple populations of homomers and heteromers, and unfortunately, these experiments do not answer this question.

2. The patch-clamp recordings of whole-cell CRAC current are not convincing. The I/V plots are not inwardly rectifying as would be expected from normal CRAC currents, and there is no mention of correction for leak current. This throws into question the conclusion that Orai3 is "a moderate negative regulator of CRAC channel activity," especially as the KO had no effect on SOCE of naïve cells, and actually decreased SOCE in pre-activated cells or A20 B cells. A careful analysis of current characteristics in the various wild-type and knockout B cells might provide evidence for heteromultimers, but these data do not.

3. The data from the Orai1 KO cells do not support the conclusion that NFAT activation depends specifically on native Orai1. Figure S3A shows that in the Orai1 KO cells, IgM and TG still dephosphorylate NFAT, just not as much as in control – this is to be expected, as SOCE is partially reduced by O1 KO. If Orai1 were the only isoform that supports NFAT activation, IgM or TG would not dephosphorylate NFAT in the KO cells.

4. The authors state that Orai1 and Orai3 are critical for primary B cell proliferation and survival (Figure 8). This conclusion needs to be toned down, as KO of either protein alone had little or no effect, and even the double KO had only a partial effect on proliferation. At any rate, the relevance of measuring viability effects after 72 hrs in vitro seems questionable, given that these conditions differ significantly from the environment of the cells in vivo (where there are no discernable functional effects of the DKO).

5. It is surprising (and disappointing!) that the deletion of Orai1 and Orai3 in B cells did not affect humoral immunity to influenza A infection in mice. In my view, this result seriously undermines the significance of the paper. If Orai channels are not required for B cell development or function in vivo, it is not clear how the in vitro results of the many different assays in this paper relate to the humoral immune response.

---

## [Author Response]

[Editors’ note: the authors resubmitted a revised version of the paper for consideration. What follows is the authors’ response to the first round of review.]

Essential revisions:1) Verify the true extent of cell surface Orai1 expression in Orai1 KO B cells: The data in Figure 3A, B clearly show that expression in the MB1-Cre cells is ~75% of the wildtype controls implying that most Orai1 protein is in fact still retained at the cell surface. This makes it very difficult to draw any realistic conclusions about the role of Orai1, and by extension, Orai3 using the Orai1/Orai3 DKOs. The Orai1 antibody should be validated, and if it is not specific, a better one should be used with appropriate validation.

We sincerely apologize for mistakenly including the data in Figure 3A, B, which resulted in an unfortunate misunderstanding. Data in Figure 3A, B were preliminary and based on earlier experiments conducted without Fc block, resulting in non-specific binding of the anti-Orai1 antibody (via its Fc region) to B cells (via Fc receptors expressed on B cells). In this revision, we now show data from experiments that were performed with an Fc block. These data show that the surface expression of Orai1 in B cells from Orai1^fl/fl^ Mb1^cre/+^ mice is reduced by ~ 70%, as determined by flow cytometry with an updated intracellular staining protocol. The vast majority of the residual signal of ~ 30% above FMO (fluorescence minus one, i.e. the anti-Orai1 antibody signal) is likely due to non-specific binding of the Orai1 antibody by its variable region (not its Fc part) to other Orai homologues (Orai2, Orai3), which partially share the peptide epitope recognized by the anti-Orai1 antibody. We also would like to kindly point out other lines of evidence indicating that Orai1 is indeed deleted in B cells of Orai1^fl/fl^ Mb1^cre/+^ mice: (a) Our qPCR data show almost complete absence of Orai1 mRNA in their B cells. (b) Our SOCE data show strongly reduced SOCE in their B cells. (c) Cre-mediated deletion of Orai1^fl/fl^ alleles was shown to be very efficient in other cell types including T cells (Kaufmann/Feske 2016, PMID 26673135) and neuronal progenitor cells (Somasundaram/Prakriya, PMID 24990931).

2) Related to the above point, the strong residual NFAT activation in Orai1 KO cells (surprising!), and lack of effect on B cell proliferation (which contradicts Gwack et al., 2008) seems consistent with the persistence of functional Orai1 protein at the cell surface.

The original study by Gwack et al. first reported on global Orai1^-/-^ mice generated through targeted Orai1 gene deletion. B cells isolated from these Orai1^-/-^ mice showed a robust (~70-75%) decrease in SOCE (see Figure 2C and Figure 4A) that agrees with data reported in our manuscript with B cells from Orai1^fl/fl^ Mb1^cre/+^ mice. Taking this and our response to the above comment into consideration, we conclude that while Orai1 expression in our B cells is not zero as is the case for data by Gwack et al., it is dramatically reduced in B cells from Orai1^fl/fl^ Mb1^cre/+^ mice. Although the NFAT activation is only partially inhibited in naive Orai1 KO B cells (Figure 7D, E), it correlates well the incomplete suppression of SOCE. Further, when one considers activated B cells (Figure 7F, G), NFAT dephosphorylation is dramatically inhibited. Gwack et al. showed a partial inhibition of proliferation in B cells from Orai1^-/-^ mice. While we see a small inhibition of proliferation in Orai1 KO B cells, we see more reduction in viability of these cells. Based on our Orai1 antibody staining, SOCE measurements and qPCR data (the latter showing ~97% decrease of *Orai1* mRNA in B cells from Orai1^fl/fl^ Mb1^cre/+^ mice), we can only speculate that in the B cells from Orai1^-/-^ mice in Gwack et al., Orai1 is knocked out from early B cell development in mice that are on mixed ICR background (otherwise mice would die) whereas in our Orai1^fl/fl^ Mb1^cre/+^ mice, Orai1 is deleted later in B cell development in mice that are on a pure C57Bl/6 background. Another potential explanation could be the complete deletion of Orai1 gene in the studies of Gwack et al. vs the 97% reduction in mRNA expression we observe in our B cells from Orai1^fl/fl^ Mb1^cre/+^ mice. Nevertheless, our results on B cell proliferation and NFAT dephosphorylation fully agree with B cell proliferation data from B cell specific STIM1/2 double KO mice (Matsumoto et al. 2011: PMID: 21530328 and Berry et al. 2020, PMID: 32563861).

3) The paper seems to imply that CRAC channels in B cells are heteromultimers of Orai1/Orai3 proteins. Without electrophysiological analysis, this claim realistically cannot be made and the study needs to demonstrate this point more definitively than simply looking at increases or decreases in cellular calcium concentrations.

We apologize if we seemed to imply that CRAC channels in B cells are heteromultimers of Orai1/Orai3. This was not our intention. Our statements were intentionally vague regarding his issue, something that reviewers noticed. As suggested, we have now included whole cell patch clamp measurements from primary B cells acutely isolated from mice (Figure 6) showing that Orai1 KO and Orai3 KO have additive effects on CRAC currents with Orai3 KO causing a small but significant enhancement of CRAC currents. While these data (along with the faster rate of SOCE inhibition by 2-APB in Orai3 KO B cells; Figure 5B, D) provide some evidence that Orai1 and Orai3 potentially heteromultimerize, they do not rule out the presence of homohexamers of Orai1 and Orai3. We have tempered our conclusions to reflect these possibilities.

4) The difference in phenotypes between Orai1 and Orai1/3 DKO B cells needs to be presented side by side to allow direct comparison. This should be done for Ca oscillations, NFAT activation and kinetics, proliferation, OCR, and respiration. See reviewer 2's comments on this matter. Experiments pertaining to Orai1 KO B cells in Figures 3 and 4 should also include those from Orai1/3 DKO B cells.

We appreciate the convenience of showing data from Orai1 and Orai1/3 DKO B cells side by side to allow comparison. We have opted to present side by side data only when experiments were performed side by side. We have the Orai1 and Orai1/3 DKO B cell data side by side for SOCE, NFAT activation, proliferation, OCR, respiration, and whole cell patch clamp (Figures5-9 and Figure 5—figure supplement 1; Figure 7—figure supplement 1). The only time where only Orai1 data (ca^2+^ oscillations and B cell development) is shown is in Figures3-4 as pointed out by the reviewer. These experiments were performed much earlier when we were still breeding mice to generate double Orai1/3 B cell knockout. Unfortunately, we do not have the means to evaluate B cell development in Orai1/3 DKO B cells, as we no longer have the double Orai1/3 KO mice due to our recent move to the University of Pittsburgh. However, please see our response to comment 6 below.

5) The functional significance of the study in its current form is very limited. Additional functional analysis to include cytokine production or antibody production is needed to increase the significance of the study.

We are grateful for this suggestion. In this revision, we now provide a heat map of the B cell cytokine and cytokine receptor profile (after IgM stimulation) from the RNA-seq results (Figure 11—figure supplement1). Furthermore, we have performed additional experiments to address this important issue using an in vivo mouse infection model. We reasoned that considering the contribution of Orai1 and Orai3 to B cell NFAT activation and metabolism, we hypothesized that deletion of *Orai1* and *Orai3* would compromise immune responses, including antibody production in response to influenza A virus (IAV) infection. These data, which are now reported in Figure 11 show that B cell-specific deletion of *Orai1* and *Orai3* does not significantly impair immune responses to influenza virus infection. We have discussed the implication of these findings in the manuscript.

6) B cell development. Check whether B cell development is affected in Orai1/3 Mb1cre KO mice. As noted by reviewer 2, Orai3 KO alone is unlikely to affect B cell development, however, does co-deletion of Orai1 and Orai3 affect B cell development or mature B cell populations in the spleen and peritoneum (Figure 4)?

Unfortunately, we cannot perform these experiments as we no longer have the double Orai1/3 KO mice due to our recent move to the University of Pittsburgh. However, we would like to respectfully argue that based on our data with Influenza virus infection (Figure 11) and on data from mice with double STIM1/2 KO mice (Matsumoto et al. 2011: PMID: 21530328 and Berry et al. 2020, PMID: 32563861), it is highly unlikely that B cell development would be affected in Orai1/3 KO mice.

7) What is the reason for reduced respiration in the Orai1/3 KO B cells which seems more than that observed in Orai1 KO cells? Use RNA-seq to examine this question.

This is an important observation. We have included GSEA analysis on previously performed RNA-seq comparing control B cells to Orai1 KO B cells and control B cells to Orai1/3 KO B cells. As suggested, we now include these data in the manuscript (Figure 9—figure supplement 1) that show that only Orai1/3 KO (and not Orai1 KO) B cells have defects in OXPHOS genes and MYC targets genes. These data indicate that Orai3 function complements that of Orai1 and could be explained by the patch clamp data where deletions of Orai1 and Orai3 have additive effects on CRAC currents. Alternatively, it is also possible that Orai3 might regulate ca^2+^-independent gene programs supporting OXPHOS. Future studies are needed to shed light on this issue.

Reviewer #1 (Recommendations for the authors):In this study, Emrich and coworkers examine the consequences of ablating Orai3 and Orai1 on store-operated calcium entry and B cell functions. It is known that SOCE is important for T cell function including T cell activation, proliferation, and cytokine synthesis. But less is known about Orai channel contributions to B cell function. Using conditional deletion of Orai1 and Orai3 in B cells, the authors examine several functional endpoints including B cell proliferation, development, and metabolism. These endpoints are diminished by Orai1 deletion. Deletion of Orai3 evokes no phenotype for most endpoints. The results indicating a lack of developmental effects on B cells with Orai1 deletion are not surprising and replicate previous work on these topics both in B cells and in T cells (relevant papers from Feske lab). Orai1 deletion also prevents NFAT activation, a well-known consequence of Orai1 inhibition in many cells. However, in several cases, the double Orai1/Orai3 knockout shows phenotypes that are slightly larger than those seen in Orai1-alone deficient B cells, suggesting that Orai3 plays a modest role in residual calcium entry and cellular functions.While the experiments are carefully conducted, I think the paper exaggerates the role of Orai3 in B cell function. Many statements such as "Orai3 and Orai1 coordinate for efficient B cell proliferation" are not meaningful and vague and overstate the role of Orai3. In fact, Orai1 or Orai3 deletion has no effect on B cell proliferation. The lack of effect of Orai1 deletion is somewhat surprising and contradicts earlier findings (Gwack et al., 2008) indicating that Orai1 is critical for B cell proliferation and this contradiction is not explained. The double Orai3/Orai1 KO evokes a modest reduction of B cell proliferation but it is unclear whether this is a specific effect or downstream of more general suppression of metabolism since it is well known that rapidly proliferating cells require a high level of metabolism and glycolysis which is suppressed in the double KO. Perhaps the most interesting part of the paper is that the double KO shows a greater suppression of oxygen consumption and maximal respiration than Orai1 KO cells alone. But no effect is seen with Orai3 KO ablation and none on basal respiration even though deletion of Orai1 alone evokes the strongest effects. Thus, statements indicating that Orai1/Orai3 channels are crucial for metabolic reprogramming do not easily follow from the data shown, which seem to indicate very minor contributions of Orai3 for these endpoints.

We thank reviewer 1 for their critical reading of our manuscript. In this revision, we have added several new experiments, including RNA-seq data showing the OXPHOS genes are only altered in Orai1/3 KO B cells, and not in Orai1 KO B cells. We have also altered our interpretations of the data to reflect the evidence presented, by removing vague or exaggerated statements. We have also discussed our results on B cell proliferation considering the findings by Gwack et al. 2008 (please see response to essential comment # 2 above).

Overall, the role of Orai3 is greatly over-emphasized and is not supported by the results. One could easily conclude based on the results that Orai3 is largely dispensable/redundant for the B cell responses that were studied here. But it is also worth noting that only a small subset of physiologically relevant B cell responses was assessed and notably cytokine and antibody production were not studied at all. These issues reduce the significance of the study.

We agree with the reviewer that for the parameters analyzed herein, Orai3 is largely dispensable. Orai3 contribution, to OXPHOS in particular, becomes apparent only within the context of double Orai1/Orai3 knockout. We now show in vivo data with B cell Orai1/3 double KO mice in a model of Influenza infection.

Additional comments:1) What is the effect of deleting Orai3 on NFAT1 activation? A lack of effect on cell proliferation in vitro is not surprising and is reminiscent of the lack of effects on proliferation in T cells with non-function Orai1 channels (McCarl, Feske et al., JI 2010).

Data in Figure 7—figure supplement 1 shows that deletion of Orai3 alone has no effect on NFAT1 dephosphorylation, which is expected.

2) Are Orai channels in a complex with Orai1 and Orai3 subunits in the same channel or are they expressed separately to form distinct channels in B cells? 2-APB appears to inhibit the residual SOCE in Orai3 deficient cells but not in Orai1 deficient cells. What are the implications of this result? The 2-APB experiment is uninterpreted in terms of the molecular composition of the SOCE channels.

We thank the reviewer for this comment. This is a very important question that is hard to answer within the context of native channel expression. Unlike the case where a molecular complex ceases to function when one of its subunits are absent, each Orai isoform is capable of mediating CRAC currents either in a heterocomplex or on its own. The 2-APB data showing faster rate of SOCE inhibition by 2-APB in Orai3 KO B cells compared to wildtype B cells (Figure 5B, D) suggest that Orai3 and Orai1 form a heterocomplex. We now present whole cell recordings of CRAC currents (Figure 6), which also suggest Orai1/3 heteromultimerization. However, these data do not exclude that at least a portion of Orai1 and Orai3 proteins are functional as homohexamers. We have discussed these possibilities in the text.

3) In order to really assess the contributions of Orai3 to B cell physiological responses, it is important to directly compare the outcomes side-by-side for the different endpoints. Thus, the metrics for NFAT activation, proliferation, OCR, respiration, etc. should be directly compared as in Figure 2.

Please see our response to essential comments #4 above.

4) Figure 6: the fact that Orai1 deletion in primary B cells causes a substantial decrease in NFAT1 activation but has no effects on Ca oscillations (Figure 2/3) implies that the Ca oscillations have no direct relevance for NFAT activation in B cells. This result seems to contradict previous literature on this issue and should be discussed. Is NFAT1 activation more dependent on the plateau Ca responses instead of oscillations?

Thank you for raising this important point. We have recently shown using combinations of Orai and STIM CRISPR KO in HEK293 cells that regardless of the nature of the ca^2+^ signal (plateau vs Oscillatory) NFAT activation depends on the ca^2+^ signal generated by native Orai1 (Yoast et al. 2020; PMID: 32415068). Specifically, ca^2+^ oscillations were intact in Orai1 KO cells (oscillations in Orai1 KO cells were mediated by Orai2/3) but NFAT nuclear translocation was abrogated. We now discuss this issue in the manuscript.

5) There is a large discrepancy between mRNA and protein levels in Figure 3. What is the true efficiency of deletion?

Please see new data in Figure 3 and please refer to our response to essential comment #1.

6) Two key functions of B cells are antibody production and secretion of cytokines that fosters T cell activation. It is hard to assess the importance of Orai channels for B cell physiology without at least some measurement of these functions. As such, the narrow focus on the study on B cell proliferation and metabolism falls short of truly understanding the role of SOCE in B cell physiology.

To address this comment, we now include in vivo data in Figure 11 and we provide a discussion of the meaning of these in vivo results considering the defects in B cell NFAT activation and metabolism in Orai1/3 KO B cells.

Reviewer #2 (Recommendations for the authors):This manuscript examines the roles of Orai channels, the predominant ca^2+^ channels in T cells, in the development and activation of B cells. This manuscript evaluates the detailed role of the members of the Orai family, Orai1, Orai2, and Orai3 channels, in ca^2+^ signaling, proliferation, and metabolism in B cells. The authors find a positive contribution of Orai1 and Orai3 in B cell activation and metabolic functions using knockout mouse studies. This study has significant novelty and impact.

We thank reviewer 2 for their positive comments on our manuscript.

Overall, the manuscript has novelty in terms of examining the role of the Orai channels in B lymphocytes, which is currently understudied. While the role of Orai1 in B cell activation and proliferation has been described before, that of Orai3 is not studied. The major weakness of the current work is that the authors need to emphasize experiments showing comparative studies between Orai1 KO, Orai3 KO, and Orai1/3 DKO B cells. Ideally, these results must be compared to Orai1/2 DKO B cells. To strengthen their conclusions, I have the following recommendations:

We thank reviewer 2 for their critical reading of our manuscript. While we agree with reviewer 2 assessment, it would take months or years for us to generate Orai1/2 DKO B cells, or ideally the Orai1/2/3 B cell triple KO mice. We hope the reviewer agrees that this should be considered under a separate study.

1. To emphasize the role of Orai3 in B cells, the authors should check whether B cell development is affected in Orai1/3 Mb1cre KO mice. Orai3 KO alone is unlikely to affect B cell development, however, does co-deletion of Orai1 and Orai3 affect B cell development or mature B cell populations in the spleen and peritoneum (Figure 4)?

Please see our response to essential comment #6.

2. The difference in phenotypes between Orai1 and Orai1/3 DKO B cells (which is the novelty of this paper) needs to be presented side by side. Experiments pertaining to Orai1 KO B cells in Figures 3 and 4 should also include those from Orai1/3 DKO B cells. For example, does the loss of both Orai1 and Orai3 affect ca^2+^ oscillations in DKO primary B cells (Figure 4), similar to their observation with cell line in Figure 3? The NFAT immunoblots from Figure 6 supplement should move to the main Figure 6. And it will be useful to compare NFAT translocation kinetics shown in Figure 6A among Orai1 KO and Orai1/3 DKO B cells.

Please see our response to essential comment #4. We have side by side comparisons each time this was possible. Sadly, we are unable to perform two experiments with Orai1/3 double KO B cells requested by the reviewer, (i) B cell development and (ii) ca^2+^ oscillations, due to the current unavailability of live Orai1/3 KO mice (it would take us months to bring up those colonies). We show that NFAT nuclear translocation in Orai1/3 DKO B cells is equivalent to that in Orai1 KO B cells based on Westerns of NFAT dephosphorylation (Figure S3). Although we cannot test the NFAT translocation kinetics in Orai1/3 DKO B cells by ImageStream for the same reason listed above, we do not expect this to differ from that in Orai1 KO B cells.

3. What is the physiological output of reduced proliferation in Orai1/3 DKO B cells seen in Figure 7? As shown in the STIM1/2 DKO B cell paper, the authors can check IL-10 production by all 4 genotypes by ELISA to get a quantitative comparison between Orai1 and Orai1/3 DKO cells. The authors should also show a heat map of the B cell cytokine profile (after IgM stimulation) from the RNA-seq results (Figure 7D and E).

We thank the reviewer for this suggestion. We know include the data of the B cell cytokine profile after anti-IgM stimulation (Figure 11—figure supplement 1).

4. In Figure 8, are the mitochondria numbers affected in Orai1/3 DKO (Figure 8D)? What is the cause of reduced respiration (more reduction than that observed in Orai1 KO cells) in the Orai1/3 KO B cells? Using RNA-seq data can the authors uncover the mechanisms linking loss of mitochondrial metabolism to reduced SOCE via Orai1/3?

Due to limitations in obtaining large numbers of Orai1/3 KO mice during breeding, we have analyzed mitochondrial mass in one preliminary experiment that showed no obvious defect in mitochondria number of Orai1/3 DKO B cells. Hence, we did not pursue this avenue further. Per reviewer request, we now present RNA-seq data showing unique defects in OXPHOS genes in B cells from Orai1/3 DKO mice.

5. Ideally, the metabolomic analysis in Figure 9 should be compared between Orai1 KO and Orai1/3 DKO B cells.

We sincerely apologize for not including these data, while we performed RNA-seq from B cells of both Orai1 KO and Orai1/3 DKO mice, metabolomic analysis was performed only in B cells from Orai1/3 DKO mice.

Reviewer #3 (Recommendations for the authors):Orai channels mediate calcium influx which is essential for immune cell activation. While the role of Orai1 and Orai2 are well established in T lymphocytes, the particular Orai isoforms that function in B cells are not well established. This study uses CRISPR-mediated gene deletion in the A20 B-cell line and primary B cells in mice to examine the functional roles of Orai1, 2, and 3. The authors show evidence that Orai1 and Orai3 are critical components of store-operated Ca channels in B cells and help drive downstream events in B cell activation. New findings include:Orai1 and Orai3 are upregulated and Orai2 is downregulated during B cell activation by different stimuli in vitro, including anti-IgM, anti-IgM + anti-CD40, and LPS. The significance of this result is undetermined, as the impact of Orai regulation on B cell responses and Ca signal generation was not studied.Based on the effects of CRISPR-mediated deletion, Orai1 and Orai3 both contribute to the magnitude of store-operated Ca entry evoked by full ER Ca depletion in A20 B cells and primary B cells. The combination of Orai1 and Orai3 was critical for the generation of sustained Ca oscillations after BCR stimulation, while deletion of one or the other channel had little effect. However, surface staining of the Orai1 knockout B cells with polyclonal anti-Orai1 antibodies indicated that despite complete suppression of mRNA, surface protein levels were ~75% of wild-type cells. The persistence of Orai1 protein weakens many of the conclusions about the functions of Orai1 based on the knockout cells, e.g. that Orai1 is dispensable for B cell differentiation in vivo, proliferation, and survival after stimulation in vitro, and has only a modest effect on NFAT activation in vitro.The double knockout of Orai1 and Orai3 has greater effects on proliferation and survival than Orai1 knockout alone, and greatly inhibits metabolic reprogramming of B cells after in vitro stimulation. However, due to the persistence of Orai1 protein the magnitude of these effects may underestimate the true contribution of Orai1.

We thank the reviewer for the accurate representation of our findings. Please see our response to essential comment 1. Regarding two specific critiques of Reviewer 3, we would like to respectfully point out that “*CRISPR-mediated deletion of Orai1 in mouse B cells”* is not correct, but that Orai1 was deleted by conditional gene targeting. The papers by (Kaufmann/Feske 2016, PMID 26673135) and (Somasundaram/Prakriya, PMID 24990931) have shown deletion of Orai1 in T cells and neuronal progenitor cells in these mice. Regarding the comment “*The characteristics of this antibody are not described, and there are questions about its specificity”,* we have specified in the Methods (under flow cytometry section) the origin of the anti-Orai1 antibody (Gwack 2008, PMID 18591248 and McCarl 2009, PMID 20004786).

1. Orai1 and Orai3 are shown to be upregulated, while Orai2 is downregulated after stimulation of B cells in vitro (Figure 1), and the effects are dependent on the type of stimulating agent. These same stimulation conditions are used later in experiments on NFAT activation, proliferation and survival, and metabolic reprogramming. However, an essential link between upregulation and functional effects is how are the Ca signals affected? SOCE (with TG) and oscillations (with anti-IgG) should be measured.

We thank the reviewer for this suggestion. We have explored the link between Orai1 and Orai3 transcript upregulation and SOCE, but only in the condition of 48hr stimulation with anti-IgM+anti-CD40. Please compare Figure 5B, C to Figure 5—figure supplement 1. We do not see any apparent increase in SOCE in activated B cells. However, since activated B cells become lymphoblasts with a bigger volume, we propose that the upregulation of Orais is likely meant to maintain SOCE, commensurate with the enhanced volume of activated B cells.

2. A serious problem that affects many of the conclusions of the paper is that CRISPR-mediated deletion of Orai1 in mouse B cells causes only a ~25% decrease in Orai1 protein as detected by the polyclonal antibody. The characteristics of this antibody are not described, and there are questions about its specificity. Has it been validated against cells known to lack Orai1? Was it directly labeled or was a secondary Ab used for FACS? Surface labeling should also be assessed in the A20 Orai1 KO lines.Assuming the antibody is specific, the persistence of 75% of the surface Orai1 undermines or changes the interpretations of many of the experiments in the paper:a) Cannot say that Orai1 is dispensable for ca^2+^ oscillations (line 206-7), B cell development (Figure 4), B cell proliferation and survival (Figure 7), or increased mitochondrial mass in activated B cells (Figure 8).b) The enhancement of SOCE in Orai2 KO cells does not actually indicate that residual SOCE in Orai1/Orai3 DKO is due to Orai2 (Figure 5; lines 251-2). An alternative explanation is that the residual SOCE is due to Orai1, and the very low SOCE seen is from Orai2 suppressing the already reduced function of Orai1.c) Cannot conclude that Orai3 is an "essential" component of the native SOCE pathway in B cells if Orai1 was not totally eliminated in the Orai1 KO (and therefore had a minimal effect) (line 413).It is also worth noting that because Orai1 is the only isoform that supports NFAT activation, the persistence of NFAT translocation in the Orai1 KO cells is another sign that Orai1 protein is still present and functional at significant levels (Figure 6). Moreover, the persistence of residual Orai1 may explain why the phenotype of the Orai1/Orai3 DKO is not as severe as the STIM1/STIM2 DKO studied by other groups (line 446-8).

Please see our response to essential comment 1.

3. At several points in the paper, the authors imply that the CRAC channel in B cells is a heteromultimer of Orai1 and Orai3 (e.g., lines 259-60). Orai3 contributes to SOCE, but no arguments are made to rule out the alternative that Orai1 and Orai3 make separate channels. Pharmacology, kinetic profiling (CDI), or permeation properties of CRAC currents in whole-cell recordings may be an approach to distinguish between these possibilities. Otherwise, it remains an open question.

We agree with the reviewer that the oligomeric state of native CRAC channels in B cells (or any other cell type for that matter) remains an open question, and we have clearly stated this in the manuscript. While our attempts to study CDI have been unsuccessful due the small nature of these native currents, we provide whole-cell CRAC measurements in primary B cells from different Orai KO mice as requested by the reviewer. Based on the increased CRAC current size in B cells from Orai3 KO mice, we propose that Orai1 and Orai3 likely heteromultimerize. However, we concede that this does not rule out the existence of homohexameric channels and that these recordings do not represent in themselves proof of heteromultimerization. More studies beyond the scope of this manuscript are needed to resolve the CRAC channel composition in native cells.

[Editors’ note: what follows is the authors’ response to the second round of review.]

The manuscript has been improved but there are some remaining key issues that need to be addressed in the reviewers' comments. The essential points are:1) The patch-clamp data are not credible and the currents do not show the classic inward rectification that is the hallmark of CRAC. Given that no other features are shown, these data should be removed as they are not critical for the study.

Thank you! We agree. These data have been removed.

2) There continue to be claims indicating that Orai3 and Orai1 form heteromeric channels. There is no evidence in the study to support this conclusion and the results can be fully explained considering that Orai1 and Orai3 form functional but separate channels. Since the conclusions don't depend on this finding, these claims should be removed.

Thank you! We have revised the manuscript so there are no claims in the conclusions as to the composition of CRAC channels in B cells. We apologize, our statements were not meant as conclusions but as speculation on the different possibilities. This is obviously a complex question that is not easy to address. The native CRAC channel in a single B cell is likely dynamic and complex with both homomers and different heteromeric assemblies.

3) Reviewers had a problem with language that make claims about Orai3 being essential (or "critical") for CRAC channel function, or other things such as proliferation (in the Title, Abstract, Discussion, and Results headers). These appear to be left over from the previous version and were not revised. Since the SOCE measurements clearly show that Orai3 is not critical or essential for CRAC channel activity nor "essential" for function including proliferation the language should be changed to "involved" in or something analogous.

The manuscript has been revised accordingly. We changed the title, abstract and text to reflect this. In the Abstract we specified that Orai3 alone is not essential: “The combined loss of Orai1 and Orai3, but not Orai3 alone, impairs SOCE, proliferation and survival….” In the results subheadings and throughout the text, we emphasize that only the combined deletion of Orai1 and Orai3 has an effect on different cellular parameters. Statements such as “Orai3 is essential for SOCE” were replaced by “Orai3 is involved in SOCE” or “Orai3 and Orai1 mediate SOCE”

4) Discuss why the Orai3 cKO cells have a respiration defect as SOCE measurements do not show any change in SOCE (Figure 5).

We now discuss this issue in the manuscript’s discussion. Please note that although Orai3 KO alone has no measurable effect on SOCE in naïve B cells, the data with activated B cells show a partial contribution of Orai3 to B cell SOCE (Figure. 5—figure supplement 1). These data are consistent with the bioenergetics data, which are from B cells activated for 24hrs with anti-IgM (naïve B cell are metabolically quiescent).

5) Kudos to the authors for evaluating several in vivo metrics of B cells lacking both Orai1 and Orai3 for B cell cytokine production, IgG antibodies, receptors, and overall immunity for infection (Figure 11). The lack of any effects is extremely surprising. This raises a key concern regarding the in vivo significance of Orai1 and Orai3 for B cell function. In particular, what do all the in vitro data mean if knocking out Orai1 and Orai3 has no effect on B cell development or function? This should be tackled in the Discussion.

We agree that the lack of any effects of B cell double knockout of Orai1 and Orai3 on humoral immunity to Influenza A infection is extremely surprising. However, these are the results we obtained and no matter how disappointing, they reflect reality. We believe that scientific reality should never be referred to as concerning or disappointing. Otherwise, we set our trainees on the path of “magical realism”, where “exciting” results “must be obtained” if they wish their papers accepted for publication. At the very least, our results offer a contrast with the function of SOCE in T cells, offer further evidence that the immune defects of CRAC channelopathies in humans are mostly T cell driven and emphasize the evolutionary hierarchy of SOCE between lymphocyte subsets, which is important to know from an academic standpoint and might be useful for future targeting of CRAC in a T cell-driven vs a B cell driven disease. We now provide in the discussion some ideas on why the results in vivo do not reflect the in vitro requirement of SOCE for NFAT induction, and metabolic reprogramming (please see end of the discussion).

Reviewer #1 (Recommendations for the authors):This revised manuscript by Emrich et al. has addressed most of the concerns brought up in the previous round of reviews. The authors have done a nice job and appropriately revised the text and discussion, but a couple of key points remain. Please see the detailed comments below.– The new patch-clamp data in Figure 6 are confusing; the currents do not show the classic inward rectification of CRAC currents, and the ramps shown in A-C in fact seem to have the wrong curvature. Since these data do not support the idea of heteromeric Orai1/Orai3 channels and are of unclear origin, I suggest removing these data as they are not needed to support the main conclusions.

Thank you! Yes, it has not escaped our attention that the I/V relationships of these currents are indeed not typical. Therefore, we did not include these recordings in the first version (R0) of the manuscript. Although we think these recordings contain a contribution from CRAC currents, the currents are very small and most likely contaminated. In any case, we agree that these recordings should be removed from the manuscript as they do not offer more clarity or insight.

– Supplementary Figure 10: An interesting result here is that upregulation of most polar metabolites co-stimulated with anti-IgM + anti-CD40 is similar to that of control cells stimulated with IgM alone and much larger than the levels seen in WT cells exposed to the GSK compound. This difference means that the effects of the GSK compound are not equivalent to that of the compound. This should be discussed in light of the fact that the DKO eliminates SOCE almost entirely.

Thank you for this observation. We have added an explanation to the discussion. We recently characterized GSK-7975A for its equal inhibitory effect on CRAC currents mediated by each Orai isoform (Orai isoforms were individually co-expressed with STIM1 in Orai1/2/3 triple KO HEK cells), compared to differential effects on Orai isoforms with other common SOCE inhibitors like Synta66 (PMID: 32896813). While the more pronounced inhibition of the upregulation of metabolites by GSK-7975A in B cells compared to the inhibition observed in Orai1/Orai3 double KO B cells could be an off-target effect of the drug, a more likely explanation is that Orai2 contributes to this difference by mediating the residual ca^2+^ influx in Orai1/Orai3 double KO B cells.

– On page 6, the heading "Both Orai3 and Orai1 are essential components of SOCE in B cells". The word essential here is incorrect as the data shows that removing Orai3 has no effects on SOCE (Figure 5B, C, and Figure 6E). Please remove the word "essential" in this heading.

Thank you! The word “essential” was removed here, from the title, the abstract and throughout the text.

Similarly, in the abstract, the sentence "We show that Orai3 and Orai1 are essential components of native CRAC channels in B cells …" should be modified to remove the word essential for the reason above.

Done. Please see response to essential comments 2 and 3.

– Finally, the title remains unchanged from the previous version. The title should be amended since the data clearly shows that Orai3 is not essential for CRAC channel function. Please revise.

Done. Please see response to essential comments 2 and 3.

Reviewer #3 (Recommendations for the authors):Comments on authors' rebuttal:1. The effects of Orai protein up- or down-regulation in response to stimulation in vitro need to be related to their effects on NFAT activation, proliferation and survival, and metabolic reprogramming. The authors compare SOCE in naïve vs. stimulated cells after full store depletion (TG), but a comparison of Ca oscillations evoked by IgM stimulation is the relevant one and should be done since these are presumably the kind of Ca signals that are involved in regulating downstream behavior under more physiological conditions.

We respectfully disagree with reviewer 3 assessment. Our view on oscillations has evolved based on models where each STIM and Orai isoforms were deleted individually and in combination. The real correlation is between the magnitude of SOCE and NFAT nuclear translocation and not between ca^2+^ oscillations and NFAT translocation (please see our response to the second comment 3 below). We found this to be the case for HEK 293 cells (PMID: 32415068 and PMID: 33657364), B cells (the current study) and T cells (work currently in progress).

2. The updated data on Orai1 expression in the Orai1fl/fl Mb1cre/+ cells makes more sense than the original dataset. It is still not totally clear why there is only a 70% reduction in Orai1 staining. The argument that the antibody for Orai1 crossreacts with Orai2 and Orai3 is not documented here, but the fact that qPCR data show an almost complete absence of Orai1 mRNA is sufficient here to conclude that Orai1 protein is effectively suppressed.

Thank you!

3. The new patch clamp data are problematic (see comment #2 below) and cannot be used to support the idea that CRAC channels are heteromers of Orai1 and Orai3 as the authors state. One cannot have it both ways – to conclude that there are heteromers, but concede that the evidence does not prove this.

Please see our response to essential comments #1 and 2 (also, please see below)

Comments on the revised manuscript:1. A major conclusion of the paper is that Orai3 and Orai1 are "essential" or "crucial" components of native CRAC channels in B cells. This is based on findings that deletion of either protein alone partially reduces SOCE, while the double KO almost completely eliminates it. The authors argue at several points in the paper that the CRAC channel may be a heteromultimer of Orai1 and Orai3, but the data fall short of demonstrating the molecular composition of CRAC channels. In regard to Figure 5 (and Figure S2), "The slower rate of SOCE inhibition by 2-APB in wildtype B cells compared to Orai1 knockout B cells suggests that the native CRAC channel in B cells is a heteromer of Orai1 and Orai3." I disagree – the data could just as well be explained by separate populations of channels containing Orai1 or Orai3. In that case, 2-APB would inhibit Orai1 but enhance Ca influx through Orai3 (Yamashita et al., JBC 2011); thus, knocking out Orai3 would be expected to increase the inhibitory effect of 2-APB, as seen in Figure 5B. Likewise, the fact that double KO is required to inhibit oscillations could be explained simply by oscillations not being sensitive to a partial loss of SOCE, and only cease when SOCE is largely eliminated. The subunit composition of the channel could be complex, with multiple populations of homomers and heteromers, and unfortunately, these experiments do not answer this question.

Please see our response to essential comments #1, 2 and 3. We cleansed the paper from the words “essential” and “crucial” and from any reference to the oligomeric state. The patch clamp data was removed from the manuscript as requested. In the first round of review (R0), we were asked to interpret the 2-APB data, which we did in the R1 version. Yes, we agree that based on the known potentiation of overexpressed Orai3 by 2-APB, the data are compatible with both homomeric and heteromeric scenarios. We would kindly caution the reviewer against extrapolating the 2-APB potentiation of overexpressed Orai3 to physiological settings. Although we do not understand it yet, we have extensive experience in many primary cell types lacking combinations of Orai isoforms (through CRISPR/Cas9 KO) where potentiation by 2-APB is not always apparent with native levels of Orai3 expression, the case of B cells herein included. In our defense, our interpretation of the 2-APB data seemed reasonable since the remaining ca^2+^ signal in Orai1 KO cells is not potentiated by 2-APB. Regardless, we deleted those statements because we don’t think the 2-APB data, or the patch clamp data, even testing 2-APB on currents (we tried) can offer definitive answers to this complex question. We agree it is better to remove any reference to the oligomeric state of these channels, which we did.

2. The patch-clamp recordings of whole-cell CRAC current are not convincing. The I/V plots are not inwardly rectifying as would be expected from normal CRAC currents, and there is no mention of correction for leak current. This throws into question the conclusion that Orai3 is "a moderate negative regulator of CRAC channel activity," especially as the KO had no effect on SOCE of naïve cells, and actually decreased SOCE in pre-activated cells or A20 B cells. A careful analysis of current characteristics in the various wild-type and knockout B cells might provide evidence for heteromultimers, but these data do not.

Please see our response to essential comments #1 and 2. All these statements were deleted. While we agree with the reviewer statement that “A careful analysis of current characteristics in the various wild-type and knockout B cells might provide evidence for heteromultimers”, reliable CRAC current recordings from primary B cells (as opposed to T cells) is not an easy task to achieve at least in our hands, as documented by our own shortcomings with those recordings.

3. The data from the Orai1 KO cells do not support the conclusion that NFAT activation depends specifically on native Orai1. Figure S3A shows that in the Orai1 KO cells, IgM and TG still dephosphorylate NFAT, just not as much as in control – this is to be expected, as SOCE is partially reduced by O1 KO. If Orai1 were the only isoform that supports NFAT activation, IgM or TG would not dephosphorylate NFAT in the KO cells.

We apologize. We should have been more careful in wording these statements. A look at our study in HEK293 cells (PMID**:** 32415068) shows that while Orai1 is the major contributor to NFAT nuclear translocation, Orai1 KO, Orai1/2 double KO and Orai1/3 double KO cells show some level of NFAT induction when compared to lack of NFAT nuclear translocation in Orai1/2/3 triple KO cells, suggesting that Orai2 and Orai3 can alone or together support moderate levels of NFAT induction. What is striking though is that the oscillations in Orai1 KO cells are indistinguishable from those of wildtype cells while NFAT induction and SOCE (in response to thapsigargin) are significantly inhibited in HEK293 (above reference) and in B cells (Figure 6 and Figure 6—Supplement 1 of the current study). In the current study, and as pointed out by reviewer 3, NFAT dephosphorylation is partially inhibited in naïve Orai1 KO B cells (more so in activated B cells), suggesting that Orai3 and/or Orai2 can mediate the remaining NFAT dephosphorylation. We have changed the statements in the manuscript to reflect this.

4. The authors state that Orai1 and Orai3 are critical for primary B cell proliferation and survival (Figure 8). This conclusion needs to be toned down, as KO of either protein alone had little or no effect, and even the double KO had only a partial effect on proliferation. At any rate, the relevance of measuring viability effects after 72 hrs in vitro seems questionable, given that these conditions differ significantly from the environment of the cells in vivo (where there are no discernable functional effects of the DKO).

We have toned down these statements and specifically stated that the effect on proliferation is observed only with the combined deletion of Orai1 and Orai3 and can be partially rescued with ani-CD40 co-stimulation.

5. It is surprising (and disappointing!) that the deletion of Orai1 and Orai3 in B cells did not affect humoral immunity to influenza A infection in mice. In my view, this result seriously undermines the significance of the paper. If Orai channels are not required for B cell development or function in vivo, it is not clear how the in vitro results of the many different assays in this paper relate to the humoral immune response.

Indeed, this result is surprising, but we prefer a disappointing truth over an exciting artifact or falsehood as we much prefer, at the behest of reviewers, to admit our ignorance than publish our fantasies. Perhaps our manuscript deserves to be rejected because we did not obtain an “exciting” result, perhaps not. Perhaps the in vivo function of SOCE in B cells could be revealed with another in vivo model such as an autoimmunity model, perhaps not. We are truly at loss as to why we do not see any effects in the double Orai1/Orai3 knockout. In this revision, we offer a discussion of the discrepancy between the results in vivo and those in vitro and some ideas on what might explain this discrepancy.